# Understanding and Improving Feature Learning for Out-of-Distribution Generalization

**Yongqiang Chen**[1]*, **Wei Huang**[2]*, **Kaiwen Zhou**[1]*
[1]The Chinese University of Hong Kong    [2]RIKEN AIP
{yqchen,kwzhou,jcheng}@cse.cuhk.edu.hk   wei.huang.vr@riken.jp

**Yatao Bian**[3], **Bo Han**[4], **James Cheng**[1]
[3]Tencent AI Lab    [4]Hong Kong Baptist University
yatao.bian@gmail.com   bhanml@comp.hkbu.edu.hk

## Abstract

A common explanation for the failure of out-of-distribution (OOD) generalization is that the model trained with empirical risk minimization (ERM) learns spurious features instead of invariant features. However, several recent studies challenged this explanation and found that deep networks may have already learned sufficiently good features for OOD generalization. Despite the contradictions at first glance, we theoretically show that ERM essentially learns *both* spurious and invariant features, while ERM tends to learn spurious features faster if the spurious correlation is stronger. Moreover, when fed the ERM learned features to the OOD objectives, the invariant feature learning quality significantly affects the final OOD performance, as OOD objectives rarely learn new features. Therefore, ERM feature learning can be a *bottleneck* to OOD generalization. To alleviate the reliance, we propose **F**eature **A**ugmented **T**raining  (FeAT), to enforce the model to learn richer features ready for OOD generalization. FeAT iteratively augments the model to learn new features while retaining the already learned features. In each round, the retention and augmentation operations are performed on different subsets of the training data that capture distinct features. Extensive experiments show that FeAT effectively learns richer features thus boosting the performance of various OOD objectives[1].

## 1   Introduction

Understanding feature learning in neural networks is crucial to understanding how they generalize to different data distributions [2, 11, 12, 62, 67, 70]. Deep networks trained with empirical risk minimization (ERM) learn highly predictive features that generalize surprisingly well to in-distribution (ID) data [24, 75]. However, ERM also tends to learn *spurious* features or shortcuts such as image backgrounds [7, 19, 23, 84] whose correlations with labels do not hold in the out-of-distribution (OOD) data, and suffers from serious performance degeneration [39]. Therefore, it is widely believed that the reason for the OOD failures of deep networks is that ERM fails to learn the desired features that have *invariant* correlations with labels across different distributions [7].

However, several recent works find that ERM-trained models have *already learned sufficiently good features* that are able to generalize to OOD data [33, 38, 63]. In addition, when optimizing various penalty terms [1, 15, 40, 41, 53, 55, 57, 61, 69, 76, 85] that aim to regularize ERM to capture the invariant features (termed as OOD objectives), there also exists a curious phenomenon that the

---

*Equal Contribution. Work done during Yongqiang's internship at Tencent AI Lab.
[1]Code is available at https://github.com/LFhase/FeAT.

37th Conference on Neural Information Processing Systems (NeurIPS 2023).

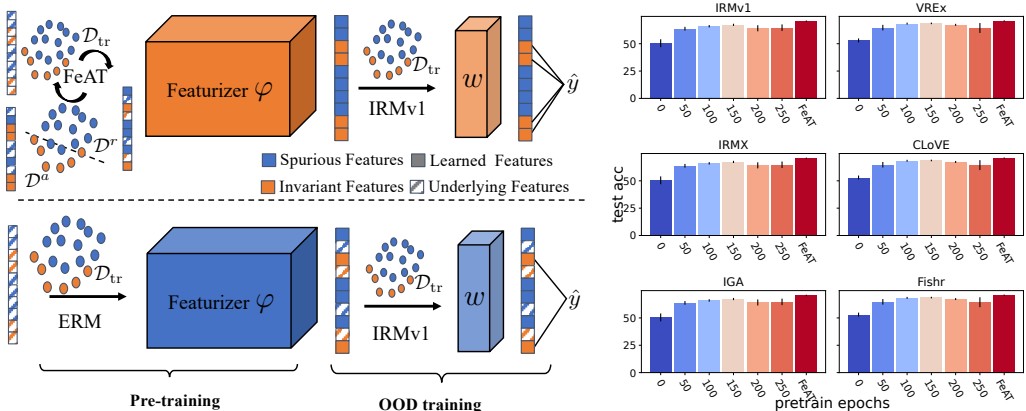

Figure 1: *(a) An illustration of FeAT (top row) compared to ERM (bottom row).* Different colors in samples denote the respective dominant features. As the original data is dominated by spurious features (blue), ERM tends to learn more spurious features but limited invariant features (orange). Thus the OOD training with IRMv1 can only leverage limited invariant features and achieve limited performance. In contrast, iteratively, FeAT divides $\mathcal{D}_{tr}$ into augmentation $D^a$ and retention sets $D^r$ that contain features not learned and already learned by the current model at the round, respectively. In each round, FeAT augments the model with new features contained in the growing augmentation sets while retaining the already learned features contained in the retention sets, which will lead the model to learn richer features for OOD training and obtain a better OOD performance. Then FeAT augments the model with new features while retaining the already learned features, which leads to richer features for OOD training and better OOD performance. *(b) OOD Performance vs. the number of ERM pre-training epochs in* COLOREDMNIST-*025.* The performance of various OOD objectives largely relies on the quality of ERM-learned features. When there exist underlying useful features poorly learned by ERM, the OOD performance will be limited. In contrast, FeAT learns richer features with 2 rounds (or 300 epochs) and improves the OOD performance.

performance of OOD objectives largely relies on the pre-training with ERM before applying the OOD objectives [16, 83]. As shown in Fig. 1(b), the number of ERM pre-training epochs *has a large influence* on the final OOD performance. These seemingly contradicting phenomena raise a challenging research question:

*What features are learned by ERM and OOD objectives, respectively, and how do the learned features generalize to in-distribution and out-of-distribution data?*

To answer the question, we conduct a theoretical investigation of feature learning in a two-layer CNN network, when trained with ERM and a widely used OOD objective, IRMv1 [4], respectively. We use a variation of the data models proposed in [2, 12], and include features with different correlation degrees to the labels to simulate invariant and spurious features [36].

First, we find that ERM essentially learns *both* spurious features and invariant features (Theorem 4.1). The degrees of spurious and invariant feature learning are mostly controlled by their correlation strengths with labels. Moreover, merely training with IRMv1 *cannot learn new* features (Theorem 4.2). Therefore, the *quality* of ERM feature learning affects the final OOD performance significantly. Hence, as the number of ERM pre-training epochs increases, the model learns invariant features better and thus the final OOD performance will increase (Fig. 1). However, when ERM does not capture *all* useful features for OOD generalization, i.e., there exist some useful features that are poorly learned by ERM, the model can hardly learn these features during OOD training and the OOD performance will be limited. Given a limited number of pre-training steps, it could often happen due to low invariant correlation strength, the feature learning biases of ERM [67], or the model architectures [28]. Consequently, ERM feature learning can be a *bottleneck* to OOD generalization [60].

To remedy the issue, we propose **F**eature **A**ugmented **T**raining (FeAT), an iterative strategy to enforce the model to learn richer features. As shown in Fig. 1(a), in each round, FeAT separates the train set into two subsets according to whether the underlying features in each set are already learned (Retention set $\mathcal{D}^r$) or not (Augmentation set $\mathcal{D}^a$), by examining whether the model yields correct ($\mathcal{D}^r$) or incorrect ($\mathcal{D}^a$) predictions for samples from the subsets, respectively. Intuitively, $\mathcal{D}^a$ and $\mathcal{D}^r$ will contain distinct features that are separated in different rounds. Then, FeAT performs distributionally

robust optimization (DRO) [50, 83] on all subsets, which *augments* the model to learn new features by minimizing the maximal ERM losses on all $\mathcal{D}^a$ and *retains* the already learned features by minimizing ERM losses on all $\mathcal{D}^r$. Along with the growth of the augmentation and retention sets, FeAT is able to learn richer features for OOD training and obtain a better OOD performance. FeAT terminates when the model cannot learn any new predictive features (Algorithm 1).

We conduct extensive experiments on both COLOREDMNIST [4, 16] and 6 datasets from the challenging benchmark, WILDS [39], and show that FeAT effectively learns richer features and thus consistently improves the OOD performance when applied to various OOD objectives (Sec. 6).

## 2 Related Work

We discuss the most related work to ours and leave more details in Appendix C.

**On Feature Learning and Generalization.** Understanding feature learning in deep networks is crucial to understanding their generalization [2, 11, 12, 22, 32, 62, 70]. Beyond the empirical probing [21, 26, 28, 65], Allen-Zhu and Li [2] proposed a new theoretical framework for analyzing the feature learning process of deep networks, which has been widely adopted to study various deep learning phenomena [12, 32, 78, 86]. However, how the learned features from ID data can generalize to OOD data remains elusive. The only exceptions are [68] and [42]. Kumar et al. [42] find fine-tuning can distort the pre-trained features while fine-tuning can be considered as a special case in our framework. Shen et al. [68] focus on how data augmentation helps promote good but hard-to-learn features and improve OOD generalization. Deng et al. [20] finds neural networks tend to learn spurious features under imbalanced groups. In contrast, we study the direct effects of ERM and OOD objectives to feature learning and provide a theoretical explanation for the curious phenomenon [33, 63]. To the best of our knowledge, we are the *first* to analyze the feature learning of ERM and OOD objectives and their interactions in the general OOD generalization setting.

**Rich Feature Learning.** Recently many OOD objectives have been proposed to regularize ERM such that the model can focus on learning invariant features [4, 41, 55, 57, 76]. However, the final OOD performance has a large dependence on the number of ERM pre-training epochs [16, 83]. To remedy the issue, Zhang et al. [83] proposed Bonsai to construct rich feature representations as network initialization for OOD training. Although both Bonsai and FeAT perform DRO on grouped subsets, Bonsai rely on multiple initializations of the whole network to capture diverse features from the subsets, and complicated ensembling of the features, which requires more training epochs for convergence. In contrast, FeAT relieves the requirements via direct augmentation-retention on the grouped subsets, and thus obtains better performance. More crucially, although rich feature learning algorithms such as Bonsai and weight averaging [5, 59] have gained some successes, explanations about the reliance of OOD performance on ERM pre-training and why rich feature learning mitigates the issue remain elusive. In addition to a new rich feature learning algorithm, our work provides theoretical explanations for the success of rich feature learning in OOD generalization.

## 3 Preliminaries and Problem Definition

**Notations.** We use old-faced letters for vectors and matrices otherwise for scalar; $\|\cdot\|_2$ to denote the Euclidean norm of a vector or the spectral norm of a matrix, while $\|\cdot\|_F$ for the Frobenius norm of a matrix. $\mathbf{I}_d$ refers to the identity matrix in $\mathbb{R}^{d\times d}$. Full details are deferred to Appendix A.

Our data model $\mathcal{D} = \{\mathbf{x}_i, y_i\}_{i=1}^n$ is adapted from [2, 12] and further characterizes each data point $\mathbf{x}_i$ as invariant and spurious feature patches from the two-bit model [16, 36].

**Definition 3.1.** $\mathcal{D} = \{\mathcal{D}_e\}_{e\in\mathcal{E}_{\text{all}}}$ is composed of multiple subsets $\mathcal{D}_e$ from different environments $e \in \mathcal{E}_{\text{all}}$, where each $\mathcal{D}_e = \{(\mathbf{x}_i^e, y_i^e)\}_{i=1}^{n_e}$ is composed of i.i.d. samples $(\mathbf{x}_i^e, y_i^e) \sim \mathbb{P}^e$. Each data $(\mathbf{x}^e, y^e) \in \mathcal{D}_e$ with $\mathbf{x}^e \in \mathbb{R}^{2d}$ and $y^e \in \{-1, 1\}$ is generated as follows:

(a) Sample $y^e \in \{-1, 1\}$ uniformly;

(b) Given $y^e$, each input $\mathbf{x}^e = [\mathbf{x}_1^e, \mathbf{x}_2^e]$ contains a feature patch $\mathbf{x}_1$ and a noise patch $\mathbf{x}_2$, that are sampled as:
$$\mathbf{x}_1 = y \cdot \text{Rad}(\alpha) \cdot \mathbf{v}_1 + y \cdot \text{Rad}(\beta) \cdot \mathbf{v}_2 \quad \mathbf{x}_2 = \boldsymbol{\xi}$$
where $\text{Rad}(\delta)$ is a random variable taking value $-1$ with probability $\delta$ and $+1$ with probability $1-\delta$, $\mathbf{v}_1 = [1, 0, \ldots 0]^\top$ and $\mathbf{v}_2 = [0, 1, 0, \ldots 0]^\top$.

(c) A noise vector $\boldsymbol{\xi}$ is generated from the Gaussian distribution $\mathcal{N}(\mathbf{0}, \sigma_p^2 \cdot (\mathbf{I}_d - \mathbf{v}_1 \mathbf{v}_1^\top - \mathbf{v}_2 \mathbf{v}_2^\top))$

Definition 3.1 is inspired by the structure of image data in image classification with CNN [2], where the inputs consist of different patches, some of the patches consist of features that are related to the class label of the image, and the others are noises that are irrelevant to the label. In particular, $\mathbf{v}_1$ and $\mathbf{v}_2$ are feature vectors that simulate the invariant and spurious features, respectively. Although our data model focuses on two feature vectors, the discussion and results can be further generalized to multiple invariant and spurious features with fine-grained characteristics [68]. Following previous works [12], we assume that the noise patch is generated from the Gaussian distribution such that the noise vector is orthogonal to the signal vector $\mathbf{v}$. Each environment is denoted as $\mathcal{E}_\alpha = \{(\alpha, \beta_e) : 0 < \beta_e < 1\}$, where $\mathbf{v}_1$ is the invariant feature as $\alpha$ is fixed while $\mathbf{v}_2$ is the spurious feature as $\beta_e$ varies across $e$.

**CNN model.** We consider training a two-layer convolutional neural network with a hidden layer width of $m$. The filters are applied to $\mathbf{x}_1$, $\mathbf{x}_2$, respectively,[2] and the second layer parameters of the network are fixed as $\frac{1}{m}$ and $-\frac{1}{m}$, respectively. Then the network can be written as $f(\mathbf{W}, \mathbf{x}) = F_{+1}(\mathbf{W}_{+1}, \mathbf{x}) - F_{-1}(\mathbf{W}_{-1}, \mathbf{x})$, where $F_{+1}(\mathbf{W}_{+1}, \mathbf{x})$ and $F_{-1}(\mathbf{W}_{-1}, \mathbf{x})$ are defined as follows:

$$F_j(\mathbf{W}_j, \mathbf{x}) = \frac{1}{m} \sum_{r=1}^m \left[ \psi(\mathbf{w}_{j,r}^\top \mathbf{x}_1) + \psi(\mathbf{w}_{j,r}^\top \mathbf{x}_2) \right], \tag{1}$$

where $\psi(x)$ is the activation function. We assume that all network weights are initialized as $\mathcal{N}(0, \sigma_0^2)$.

**ERM objective.** We train the CNN model by minimizing the empirical cross-entropy loss function:

$$L(\mathbf{W}) = \sum_{e \in \mathcal{E}_{\mathrm{tr}}} \frac{1}{n_e} \sum_{i=1}^{n_e} \ell(y_i^e \cdot f(\mathbf{W}, \mathbf{x}_i^e)), \tag{2}$$

where $\ell(z) = \log(1 + \exp(-z))$ and $\{\mathcal{D}_e\}_{e \in \mathcal{E}_{\mathrm{tr}}} = \{\{\mathbf{x}_i^e, y_i^e\}_{i=1}^{n_e}\}_{e \in \mathcal{E}_{\mathrm{tr}}}$ is the trainset with $\sum_{e \in \mathcal{E}_{\mathrm{tr}}} n_e = n$.

**OOD objective.** The goal of OOD generalization is, given the data from training environments $\{\mathcal{D}_e\}_{e \in \mathcal{E}_{\mathrm{tr}}}$, to find a predictor $f : \mathcal{X} \to \mathcal{Y}$ that generalizes well to all (unseen) environments, or minimizes $\max_{e \in \mathcal{E}_{\mathrm{all}}} L_e(f)$, where $L_e$ is the empirical risk under environment $e$. The predictor $f = w \circ \varphi$ is usually composed of a featurizer $\varphi : \mathcal{X} \to \mathcal{Z}$ that learns to extract useful features, and a classifier $w : \mathcal{Z} \to \mathcal{Y}$ that makes predictions from the extracted features.

Since we are interested in cases where the OOD objective succeeds in learning the invariant features. In the discussion below, without loss of generality, we study one of the most widely discussed OOD objective, IRMv1 objective, from IRM framework [4], and the data model where IRMv1 succeeds. Specifically, the IRM framework approaches OOD generalization by finding an invariant representation $\varphi$, such that there exists a classifier acting on $\varphi$ that is simultaneously optimal in $\mathcal{E}_{\mathrm{tr}}$. Hence, IRM leads to a challenging bi-level optimization problem as

$$\min_{w, \varphi} \sum_{e \in \mathcal{E}_{\mathrm{tr}}} L_e(w \circ \varphi), \text{s.t. } w \in \underset{\bar{w}: \mathcal{Z} \to \mathcal{Y}}{\arg\min} L_e(\bar{w} \circ \varphi), \forall e \in \mathcal{E}_{\mathrm{tr}}. \tag{3}$$

Due to the optimization difficulty of Eq. (3), Arjovsky et al. [4] relax Eq. (3) into IRMv1 as follows:

$$\min_\varphi \sum_{e \in \mathcal{E}_{\mathrm{tr}}} L_e(\varphi) + \lambda |\nabla_{w|w=1} L_e(w \cdot \varphi)|^2. \tag{4}$$

Given the convolutional neural network (Eq. 1) and logistic loss (Eq. 2), IRMv1 can be written as

$$L_{\mathrm{IRMv1}}(\mathbf{W}) = \sum_{e \in \mathcal{E}_{\mathrm{tr}}} \frac{1}{n_e} \sum_{i=1}^{n_e} \ell\left(y_i^e \cdot f(\mathbf{W}, \mathbf{x}_i^e)\right) + \sum_{e \in \mathcal{E}_{\mathrm{tr}}} \frac{\lambda}{n_e^2} \left(\sum_{i=1}^{n_e} \ell_i'^e \cdot y_i^e \cdot f(\mathbf{W}, \mathbf{x}_i^e)\right)^2, \tag{5}$$

where $\ell_i'^e = \ell'(y_i^e \cdot f(\mathbf{W}, \mathbf{x}_i^e)) = -\frac{\exp(-y_i^e \cdot f(\mathbf{W}, \mathbf{x}_i^e))}{1 + \exp(-y_i^e \cdot f(\mathbf{W}, \mathbf{x}_i^e))}$. Due to the complexity of IRMv1, in the analysis below, we introduce $C_{\mathrm{IRMv1}}^e$ for the ease of expressions. Specifically, we define $C_{\mathrm{IRMv1}}^e$ as

$$C_{\mathrm{IRMv1}}^e \triangleq \frac{1}{n_e} \sum_{i=1}^{n_e} \ell'\left(y_i^e \hat{y}_i^e\right) \cdot y_i^e \hat{y}_i^e,$$

where $\hat{y}_i^e \triangleq f(\mathbf{W}, \mathbf{x}_i^e)$ is the logit of sample $\mathbf{x}_i$ from environment $e$. The convergence of $C_{\mathrm{IRMv1}}$ indicates the convergence of IRMv1 penalty. The following lemma will be useful in our analysis.

---

[2]When the environment $e$ is not explicitly considered, we will omit it for clarity.

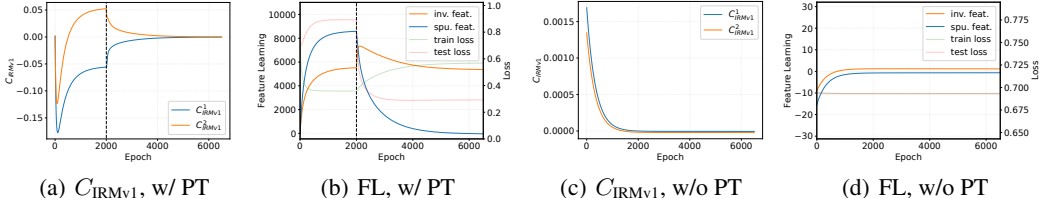

Figure 2: The convergences of $C_{IRMv1}$ and feature learning coefficients (FL) with or without ERM pre-training (PT). The invariant and spurious feature learning terms are the mean of $\langle \mathbf{w}_{j,r}, j\mathbf{v}_1 \rangle$ and $\langle \mathbf{w}_{j,r}, j\mathbf{v}_2 \rangle$ for $j \in \{\pm 1\}, r \in [m]$, respectively. The training environments are $\mathcal{E}_{tr} = \{(0.25, 0.1), (0.25, 0.2)\}$. The black dashed line indicates the end of pre-training. More details are given in Appendix D.1.

**Lemma 3.2.** *(Cao et al. [12]) Let* $\mathbf{w}_{j,r}(t)$[3] *for* $j \in \{+1, -1\}$ *and* $r \in \{1, 2, \ldots, m\}$ *be the convolution filters of the CNN at $t$-th iteration of gradient descent. Then there exists unique coefficients* $\gamma_{j,r}^{inv}(t), \gamma_{j,r}^{spu}(t) \geq 0$ *and* $\rho_{j,r,i}(t)$ *such that,*

$$\mathbf{w}_{j,r}(t) = \mathbf{w}_{j,r}(0) + j \cdot \gamma_{j,r}^{inv}(t) \cdot \mathbf{v}_1 + j \cdot \gamma_{j,r}^{spu}(t) \cdot \mathbf{v}_2 + \sum_{i=1}^{n} \rho_{j,r,i}(t) \cdot \|\boldsymbol{\xi}_i\|_2^{-2} \cdot \boldsymbol{\xi}_i. \tag{6}$$

We refer Eq. (6) as the *signal-noise decomposition* of $\mathbf{w}_{j,r}(t)$ [12]. We add normalization factor $\|\boldsymbol{\xi}_i\|_2^{-2}$ in the definition so that $\rho_{j,r}^{(t)} \approx \langle \mathbf{w}_{j,r}^{(t)}, \boldsymbol{\xi}_i \rangle$. Note that $\|\mathbf{v}_1\|_2 = \|\mathbf{v}_2\|_2 = 1$, the corresponding normalization factors are thus neglected. Furthermore, $\gamma_{j,r}^{inv} \approx \langle \mathbf{w}_{j,r}, \mathbf{v}_1 \rangle$ and $\gamma_{j,r}^{spu} \approx \langle \mathbf{w}_{j,r}, \mathbf{v}_2 \rangle$ respectively denote the degrees of invariant and spurious feature learning.

# 4 Theoretical Understanding of Feature Learning in OOD Generalization

## 4.1 ERM Feature Learning

With the setup in Sec. 3, we first study the feature learning of the ERM objective. We consider a two training environments setup $\mathcal{E}_{tr} = \{(\alpha, \beta_1), (\alpha, \beta_2)\}$ where the signal of invariant feature is weaker than the average of spurious signals (i.e., $\alpha > \frac{\beta_1 + \beta_2}{2}$), which corresponds to Figure 2. For a precise characterization of the training dynamic, we adopted a minimal setup where $\psi(x) = x$ in Figure 2(a) and the following theorem, which already captures the key phenomenon in ERM feature learning. We study ERM feature learning with *non-linear* activations in Appendix D.2.3.

**Theorem 4.1.** *(Informal) For* $\rho > 0$*, let* $\underline{n} \triangleq \min_{e \in \mathcal{E}_{tr}} n_e$*. Suppose that we run $T$ iterations of GD for the ERM objective. With sufficiently large $\underline{n}$ and $\psi(x) = x$, assuming that (i) $\alpha, \beta_1, \beta_2 < \frac{1}{2}$, and (ii) $\alpha > \frac{\beta_1 + \beta_2}{2}$, with properly chosen $\sigma_0^2$ and $\sigma_p^2$, there exists a constant $\eta$, such that with probability at least $1 - 2\rho$, both invariant and spurious features are converging and the increment of the spurious feature is larger than that of the invariant feature at any iteration $t \in \{0, \ldots, T-1\}$ (the detailed quantitative result of this gap can be found at (15) in Appendix D.2).*

As the formal statement of Theorem 4.1 is too complicated and lengthy, we leave it and its proof in Appendix D.2, while giving an informal but more intuitive version here. Theorem 4.1 states that ERM training learns both invariant feature and spurious feature at the same time, and if the average of spurious signals is stronger, the coefficient of spurious feature learning will dominate that of invariant feature learning in the whole training process, corresponding to Figure 2(b). We establish the proof based on inspecting a novel recursive equation, which might be of independent interest. Note that Theorem 4.1 can be directly generalized to handle any number of environments.

Speaking of implications, Theorem 4.1 provides answers to the seemingly contradicting phenomena that ERM fails in OOD generalization [7, 19] but still learns the invariant features [33, 38, 63]. In fact, ERM fails since it learns the spurious features more quickly, when spurious correlations are stronger than invariant correlations. Nevertheless, invariant feature learning also happens, even when the spurious correlations are strong, so long as the invariant feature has a non-trivial correlation strength with the labels. Therefore, simply re-training a classifier based on a subset of unbiased data on top of the ERM-trained featurizer achieves impressive OOD generalization performance [33, 38, 63].

---

[3]We use $\mathbf{w}_{j,r}(t)$, $\mathbf{w}_{j,r}^{(t)}$ and $\mathbf{w}_{j,r}^t$ interchangeably.

Theorem 4.1 also provides an explanation for the ID-OOD performance correlations when fine-tuning or training neural networks (especially large pre-trained models like CLIP [56], GPT [10]) [46, 71, 79, 80]. We provide a detailed discussion in Appendix C.

## 4.2 IRM Feature Learning

Although Theorem 4.1 states that ERM learns both invariant and spurious features, the following questions remain unanswered: (1) whether IRMv1 learns new features or simply amplifies the already learned ERM features, and (2) how the quality of the ERM-learned features affects the feature learning when IRMv1 is incorporated. We first study IRMv1 training from scratch (w/o pre-training).

**Theorem 4.2.** *Consider training a CNN model (1) with data model (3.1), define* $\mathbf{c}(t) \triangleq \left[ C_{IRMv1}^1(\mathbf{W}, t), C_{IRMv1}^2(\mathbf{W}, t), \cdots, C_{IRMv1}^{|\mathcal{E}_{tr}|}(\mathbf{W}, t) \right]$, *and* $\lambda_0 = \lambda_{\min}(\mathbf{H}^\infty)$, *where* $\mathbf{H}_{e,e'}^\infty \triangleq \frac{1}{2mn_en_{e'}} \sum_{i=1}^{n_e} \psi'(\langle \mathbf{w}_{j,r}(0), \mathbf{x}_{1,i}^e \rangle) \mathbf{x}_{1,i}^{e\top} \sum_{i'=1}^{n_{e'}} \psi'(\langle \mathbf{w}_{j,r}(0), \mathbf{x}_{1,i'}^{e'} \rangle) \mathbf{x}_{1,i'}^{e'}$. *Suppose that dimension* $d = \Omega(\log(m/\delta))$, *network width* $m = \Omega(1/\delta)$, *regularization factor* $\lambda \geq 1/(\sigma_0|\mathcal{E}_{tr}|^{3/2})$, *noise variance* $\sigma_p = O(d^{-2})$, *weight initial scale* $\sigma_0 = O(\frac{|\mathcal{E}_{tr}|^{7/2}\beta^3 L}{d^{1/2}m^2\lambda_0^2\log(1/\epsilon)})$, *then with probability at least* $1 - \delta$, *after training time* $T = \Omega\left(\frac{\log(1/\epsilon)}{\eta\lambda\lambda_0}\right)$, *we have* $\|\mathbf{c}(T)\|_2 \leq \epsilon$, $\gamma_{j,r}^{inv}(T) = o(1)$, $\gamma_{j,r}^{spu}(T) = o(1)$.

The proof is given in Appendix D.3. We highlight that Theorem 4.2 allows any number of training environments, which indicates a fundamental limitation of pure IRMv1 training. Intuitively, Theorem 4.2 implies that, when a heavy regularization of IRMv1 is applied, the model will not learn any features, corresponding to Figure 2(d). Instead, IRMv1 suppresses any feature learning, even at the beginning of the training. Then, what would happen when given a properly pre-trained network?

After ERM pre-training, according to Theorem 4.1, we have $|\langle \mathbf{w}_{j,r}, \mathbf{v}_1 \rangle| = \Omega(1)$, $|\langle \mathbf{w}_{j,r}, \mathbf{v}_2 \rangle| = \Omega(1)$, and $|\langle \mathbf{w}_{j,r}, \boldsymbol{\xi} \rangle| = O(\sigma_0\sigma_p\sqrt{d})$. Then, we have the following hold.

**Proposition 4.3.** *Given the same setting as Theorem 4.2, suppose that* $\psi(x) = x$, $\gamma_{j,r}^{inv}(t_1) = \gamma_{j,r}^{inv}(t_1 - 1)$, *and* $\gamma_{j,r}^{spu}(t_1) = \gamma_{j,r}^{spu}(t_1 - 1)$ *at the end of ERM pre-train* $t_1$, $\delta > 0$, *and* $n > C\log(1/\delta)$, *with* $C$ *being a positive constant, then with a high probability at least* $1 - \delta$, *we have* $\sum_e C_{IRMv1}^e(t_1) = 0$, $\gamma_{j,r}^{inv}(t_1 + 1) > \gamma_{j,r}^{inv}(t_1)$, *and* $\gamma_{j,r}^{spu}(t_1 + 1) < \gamma_{j,r}^{spu}(t_1)$.

The proof is given in Appendix D.4, which takes converged feature learning terms from Theorem 4.1 as the inputs. Proposition 4.3 demonstrates that with sufficient ERM pre-training, IRMv1 can enhance the learning of invariant features while suppressing the learning of spurious features, which is verified in Figure 2(b) and 2(a). Thus, when given the initialization with better learned invariant features, i.e., longer ERM pre-training epochs, IRMv1 improves the invariant feature better. Proposition 4.3 explains why the final OOD performance highly depends on the ERM pre-training [16, 83].

## 4.3 Limitations of ERM Feature Learning

Combining results from both Sec. 4.1 and Sec. 4.2, we know that the invariant features will be learned during ERM pre-training and discovered during OOD training. However, given poorly learned invariant features, can IRMv1 still improve it? In practice, there often exist some invariant features that are not properly learned by ERM. For example, in our data model Def. 3.1 when the invariant correlation is much weaker than the spurious correlation, given a limited number of training steps, the spurious feature learning can dominate the invariant feature learning. Besides, when considering other factors such as the simplicity bias of ERM [67] or the inductive biases of the network architecture [28], it is more likely that there exist invariant features that are not properly learned [60]. Then we have:

**Corollary 4.4.** *Consider training the CNN with the data generated from Def. 3.1, suppose that* $\psi(x) = x$, $\gamma_{j,r}^{inv}(t_1) = o(1)$, *and* $\gamma_{j,r}^{spu}(t_1) = \Theta(1)$ *at the end of ERM pre-training* $t_1$. *Suppose that* $\delta > 0$, *and* $n > C\log(1/\delta)$, *with* $C$ *being a positive constant, then with a high probability at least* $1 - \delta$, *we have* $\gamma_{j,r}^{inv}(t_1 + 1) < \gamma_{j,r}^{inv}(t_1)$.

Corollary 4.4 shows that IRMv1 requires sufficiently well-learned features for OOD generalization. It is also consistent with the experimental results in Fig. 2(b), 2(c), and Fig. 1, where all the OOD objectives only achieve a performance comparable to random guesses.

# 5 Feature Augmented Training

## 5.1 Rich Features for OOD Generalization

The results in Sec. 4 imply the necessity of learning all potentially useful features during the pre-training stage for OOD generalization. Otherwise, the OOD training is less likely to enhance the poorly learned features. It also explains the success of learning diverse and rich features by weight averaging [5, 59] and rich feature construction (or Bonsai) [83], and other approaches [58, 81].

Despite the empirical success, however, the learning of rich features in both Bonsai and weight averaging is unstable and expensive. On the one hand, they may discard previously learned useful features or fail to explore all the desired features as it is hard to evaluate the quality of the intermediate learned features. On the other hand, they also need multiple initializations and training of the whole networks with different random seeds to encourage the diversity of feature learning, which brings more instability and computational overhead, especially when applied to large and deep networks.

## 5.2 The FeAT Algorithm

To overcome the limitations of previous rich feature learning algorithms, we propose **F**eature **A**ugmented **T**raining (FeAT), that directly augment the feature learning in an iterative manner.

Intuitively, the potentially useful features presented in the training data are features that have non-trivial correlations with labels, or using the respective feature to predict the labels is able to achieve a *non-trivial training performance*. Moreover, the invariance principle assumes that the training data comes from different environments [4], which implies that each set of features can only dominate the correlations with labels in a *subset* of data. Therefore, it is possible to differentiate the distinct sets of useful features entangled in the training data into different subsets, where ERM can effectively learn the dominant features presented in the corresponding subset as shown in Theorem 4.1.

The intuition naturally motivates an iterative rich feature learning algorithm, i.e., FeAT, that identifies the subsets containing distinct features and explores to learn new features in multiple rounds. The details of FeAT are given in Algorithm 1,

---

**Algorithm 1** FeAT: **F**eature **A**ugmented **T**raining

1: **Input:** Training data $\mathcal{D}_{\mathrm{tr}}$; the maximum augmentation rounds $K$; predictor $f := w \circ \varphi$; length of inner training epochs $t$; termination threshold $p$;
2: Initialize groups $G^a \leftarrow \mathcal{D}_{\mathrm{tr}}, G^r \leftarrow \{\}$;
3: **for** $k \in [1, \ldots, K]$ **do**
4:     Randomly initialize $w_k$;
5:     **for** $j \in [1, \ldots, t]$ **do**
6:         Obtain $\ell_{\mathrm{FeAT}}$ with $G$ via Eq. 7;
7:         Update $w_k, \varphi$ with $\ell_{\mathrm{FeAT}}$;
8:     **end for**
9:     `// Early Stop if` $f_k = w_k \circ \varphi$ `fails to find new features.`
10:     **if** Training accuracy of $f_k$ is smaller than $p$ **then**
11:         Set $K = k - 1$ and terminate the loop;
12:     **end if**
13:     Split $\mathcal{D}_{\mathrm{tr}}$ into groups $\mathcal{D}_k^a, \mathcal{D}_k^r$ according to whether $f_k$ classifies the examples in $\mathcal{D}_{\mathrm{tr}}$ correctly or not;
14:     Update groups $G^a \leftarrow G^a \cup \{\mathcal{D}_k^a\}, G^r \leftarrow G^r \cup \{\mathcal{D}_k^r\}$;
15: **end for**
16: Synthesize the final classifier $w \leftarrow \frac{1}{K} \sum_{i=1}^{K} w_i$;
17: **return** $f = w \circ \varphi$;

---

where we are given a randomly initialized or pre-trained model $f = w \circ \varphi$ that consists of a featurizer $\varphi$ and a classifier $w$. In round $k$, FeAT first identifies the subset that contains the already learned features by collecting the samples where $f$ yields the correct prediction, denoted as $G_k^r$, and the subset of samples that contains the features that have not been learned, denoted as $G_k^a$.

At the $k$-th round, given the grouped subsets $G = \{G^r, G^a\}$ with $2k - 1$ groups, where $G^a = \{\mathcal{D}_i^a\}_{i=0}^{k-1}$ is the grouped sets for new feature augmentation, and $G^r = \{\mathcal{D}_i^r\}_{i=1}^{k-1}$ is the grouped sets for already learned feature retention (notice that $\mathcal{D}_0^r$ is the empty set), where $\mathcal{D}_i^a$ and $\mathcal{D}_i^r$ are the corresponding augmentation and retention set elicited at $i$-th round. FeAT performs distributionally robust optimization (DRO) [50, 83] on $G^a$ to explore new features that have not been learned in previous rounds. Meanwhile, FeAT also needs to *retain* the already learned features by minimizing the empirical risk at $G^r$, for which we store and use the historical classifiers $w_i$ with the current featurizer to evaluate the feature retention degree. Then, the FeAT objective at round $k$ is

$$\ell_{\mathrm{FeAT}} = \max_{\mathcal{D}_i^a \in G^a} \ell_{\mathcal{D}_i^a}(w_k \circ \varphi) + \lambda \cdot \sum_{\mathcal{D}_i^r \in G^r} \ell_{\mathcal{D}_i^r}(w_i \circ \varphi), \tag{7}$$

Table 1: OOD performance on COLOREDMNIST datasets initialized with different representations.

| | COLOREDMNIST-025 | | | | COLOREDMNIST-01 | | | |
|---|---|---|---|---|---|---|---|---|
| | ERM-NF | ERM | BONSAI | FEAT | ERM-NF | ERM | BONSAI | FEAT |
| ERM | 17.14 (±0.73) | 12.40 (±0.32) | 11.21 (±0.49) | **17.27** (±2.55) | 73.06 (±0.71) | 73.75 (±0.49) | 70.95 (±0.93) | **76.05** (±1.45) |
| IRMv1 | 67.29 (±0.99) | 59.81 (±4.46) | 70.28 (±0.72) | **70.57** (±0.68) | 76.89 (±3.25) | 73.84 (±0.56) | 76.71 (±4.10) | **82.33** (±1.77) |
| V-REX | 68.62 (±0.73) | 65.96 (±1.29) | 70.31 (±0.66) | **70.82** (±0.59) | 83.52 (±2.52) | 81.20 (±3.27) | 82.61 (±1.76) | **84.70** (±0.69) |
| IRMX | 67.00 (±1.95) | 64.05 (±0.88) | 70.46 (±0.42) | **70.78** (±0.61) | 81.61 (±1.98) | 75.97 (±0.88) | 80.28 (±1.62) | **84.34** (±0.97) |
| IB-IRM | 56.09 (±2.04) | 59.81 (±4.46) | 70.28 (±0.72) | **70.57** (±0.68) | 75.81 (±0.63) | 73.84 (±0.56) | 76.71 (±4.10) | **82.33** (±1.77) |
| CLOVE | 58.67 (±7.69) | 65.78 (±0.00) | 65.57 (±3.02) | **65.78** (±2.68) | 75.66 (±10.6) | 74.73 (±0.36) | 72.73 (±1.18) | **75.12** (±1.08) |
| IGA | 51.22 (±3.67) | 62.43 (±3.06) | **70.17** (±0.89) | 67.11 (±3.40) | 74.20 (±2.45) | 73.74 (±0.48) | 74.72 (±3.60) | **83.46** (±2.17) |
| FISHR | 69.38 (±0.39) | 67.74 (±0.90) | 68.75 (±1.10) | **70.56** (±0.97) | 77.29 (±1.61) | 82.23 (±1.35) | 84.19 (±0.66) | **84.26** (±0.93) |
| ORACLE | 71.97 (±0.34) | | | | 86.55 (±0.27) | | | |

where $\ell_{\mathcal{D}_i}(w \circ \varphi)$ refers to the empirical risk of $w \circ \varphi$ evaluated at the subset $\mathcal{D}_i$, and $\{w_i | 1 \leq i \leq k-1\}$ are the historical classifiers trained in round $i$. The final classifier is the average of all historical classifiers as they already capitalize all the learned features in each round.

**Relations with previous rich feature learning algorithms.** Compared with previous rich feature learning algorithms, FeAT *directly* trades off the exploration of new features and the retention of the already learned features. Although Bonsai also adopts DRO to explore new features, the isolation of new feature exploration and already learned feature synthesis makes the feature learning in Bonsai more unstable. In other words, Bonsai can not evaluate the intermediate feature learning results due to the *indirect* feature exploration and synthesis. Consequently, Bonsai can not control when to stop the new feature exploration, and thus may fail to explore all of the desired features or discard important features. Besides, the multiple re-initializations and re-training of the whole network in Bonsai could also lead to suboptimal performance and meanwhile require more computational overhead.

**Practical implementations.** Algorithm 1 requires to store $2K-1$ subsets and a larger memory cost in training the network, which may cause additional storage burden when $\varphi$ contains a massive amount of parameters [39]. Hence, we propose a lightweight variant of FeAT (denoted as iFeAT) which only retains the latest subsets and historical classifiers ($\mathcal{D}_{k-1}^a, \mathcal{D}_{k-1}^r, w_{k-1}$ at the $k$-th round). Throughout the whole experiment, we will use iFeAT and find that iFeAT already achieves state-of-the-art. More details are given in Appendix E.

As iFeAT stores only the latest augmentation and retention subsets, inspecting the training performance for termination check (line 10 of Algorithm 1) may not be suitable. However, one can still inspect the performance in either an OOD validation set to check the quality of the intermediate feature representations, or the retention set to check whether learning new features leads to a severe contradiction of the already learned features (FeAT should terminate if so).

Compared to ERM, the additional computational and memory overhead introduced in FeAT mainly lie in the FeAT training and partitioning. At each training step, FeAT needs $(k-1)$ additional forward and backward propagation, the same as Bonsai, while FeAT only needs $\min(1, k-1)$ additional propagation. Besides, Bonsai additionally require another round of training with $(K-1)$ additional propagation, given $K$ total rounds. More details are given in Appendix F.4.

## 6 Empirical Study

We conduct extensive experiments on COLOREDMNIST [16] and WILDS [39] to verify the effectiveness of FeAT in learning richer features than ERM and the state-of-the-art algorithm Bonsai [83].

**Proof-of-concept study on COLOREDMNIST.** We first conduct a proof-of-concept study using COLOREDMNIST [16] and examine the feature learning performance of FeAT under various conditions. We consider both the original COLOREDMNIST with $\mathcal{E}_{tr} = \{(0.25, 0.1), (0.25, 0.2)\}$ (denoted as COLOREDMNIST-025), where spurious features are better correlated with labels, and the modified COLOREDMNIST (denoted as COLOREDMNIST-01) with $\mathcal{E}_{tr} = \{(0.1, 0.2), (0.1, 0.25)\}$, where invariant features are better correlated with labels. We compare the OOD performance of the features learned by FeAT, with that of ERM and the state-of-the-art rich feature learning algorithm Bonsai [83]. Based on the features learned by ERM, Bonsai, and FeAT, we adopt various state-of-the-art OOD objectives including IRMv1 [4], VREx [41], IRMX [16], IB-IRM [1], CLOvE [76], IGA [40] and Fishr [57] for OOD training, in order to evaluate the practical quality of the learned features. The feature representations are frozen once initialized for the OOD training as fine-tuning the featurizer can distort the pre-trained features [43]. We also compare FeAT with the common training approach that uses unfrozen ERM features, denoted as ERM-NF. For Bonsai, we trained 2

Table 2: OOD generalization performances on WILDS benchmark.

| INIT. | METHOD | CAMELYON17 Avg. acc. (%) | CIVILCOMMENTS Worst acc. (%) | FMOW Worst acc. (%) | IWILDCAM Macro F1 | AMAZON 10-th per. acc. (%) | RXRX1 Avg. acc. (%) |
|---|---|---|---|---|---|---|---|
| ERM | DFR† | 95.14 (±1.96) | **77.34** (±0.50) | 41.96 (±1.90) | 23.15 (±0.24) | 48.00 (±0.00) | - |
| ERM | DFR-s† | - | 82.24 (±0.13) | 56.17 (±0.62) | 52.44 (±0.34) | - | - |
| Bonsai | DFR† | 95.17 (±0.18) | 77.07 (±0.85) | 43.26 (±0.82) | 21.36 (±0.41) | 46.67 (±0.00) | - |
| Bonsai | DFR-s† | - | 81.26 (±1.86) | 58.58 (±1.17) | 50.85 (±0.18) | - | - |
| FeAT | DFR† | **95.28** (±0.19) | **77.34** (±0.59) | **43.54** (±1.26) | **23.54** (±0.52) | **49.33** (±0.00) | - |
| FeAT | DFR-s† | - | 79.56 (±0.38) | 57.69 (±0.78) | 52.31 (±0.38) | - | - |
| ERM | ERM | 74.30 (±5.96) | 55.53 (±1.78) | 33.58 (±1.02) | 28.22 (±0.78) | 51.11 (±0.63) | 30.21 (±0.09) |
| ERM | GroupDRO | 76.09 (±6.46) | 69.50 (±0.15) | 33.03 (±0.52) | 28.51 (±0.58) | 52.00 (±0.00) | 29.99 (±0.13) |
| ERM | IRMv1 | 75.68 (±7.41) | 68.84 (±0.95) | 33.45 (±1.07) | 28.76 (±0.45) | 52.00 (±0.00) | 30.10 (±0.05) |
| ERM | V-REx | 71.60 (±7.88) | 69.03 (±1.08) | 33.06 (±0.46) | 28.82 (±0.47) | 52.44 (±0.63) | 29.88 (±0.35) |
| ERM | IRMX | 73.49 (±9.33) | 68.91 (±1.19) | 33.13 (±0.86) | 28.82 (±0.47) | 52.00 (±0.00) | 30.10 (±0.05) |
| Bonsai | ERM | 73.98 (±5.30) | 63.34 (±3.49) | 31.91 (±0.51) | 28.27 (±1.05) | 48.58 (±0.56) | 24.22 (±0.44) |
| Bonsai | GroupDRO | 72.82 (±5.37) | 70.23 (±1.33) | 33.12 (±1.20) | 27.16 (±1.18) | 42.67 (±1.09) | 22.95 (±0.46) |
| Bonsai | IRMv1 | 73.59 (±6.16) | 68.39 (±2.01) | 32.51 (±1.23) | 27.60 (±1.57) | 47.11 (±0.63) | 23.35 (±0.43) |
| Bonsai | V-REx | 76.39 (±5.32) | 68.67 (±1.29) | 33.17 (±1.26) | 25.81 (±0.42) | 48.00 (±0.00) | 23.34 (±0.42) |
| Bonsai | IRMX | 64.77 (±10.1) | 69.56 (±0.95) | 32.63 (±0.75) | 27.62 (±0.66) | 46.67 (±0.00) | 23.34 (±0.40) |
| FeAT | ERM | 77.80 (±2.48) | 68.11 (±2.27) | 33.13 (±0.78) | 28.47 (±0.63) | **52.89** (±0.63) | **30.66** (±0.42) |
| FeAT | GroupDRO | **80.41** (±3.30) | **71.29** (±0.46) | 33.55 (±1.67) | 28.38 (±1.32) | 52.58 (±0.56) | 29.99 (±0.11) |
| FeAT | IRMv1 | 77.97 (±3.09) | 70.33 (±1.14) | **34.04** (±0.70) | **29.66** (±1.52) | **52.89** (±0.63) | 29.99 (±0.19) |
| FeAT | V-REx | 75.12 (±6.55) | 70.97 (±1.06) | 34.00 (±0.71) | 29.48 (±1.94) | **52.89** (±0.63) | 30.57 (±0.53) |
| FeAT | IRMX | 76.91 (±6.76) | 71.18 (±1.10) | 33.99 (±0.73) | 29.04 (±2.96) | **52.89** (±0.63) | 29.92 (±0.16) |

†DFR/DFR-s use an additional OOD dataset to evaluate invariant and spurious feature learning, respectively.

rounds following Zhang et al. [83], while for FeAT the automatic termination stopped at round 2 in COLOREDMNIST-025 and round 3 in COLOREDMNIST-01. For ERM, we pre-trained the model with the same number of overall epochs as FeAT in COLOREDMNIST-01, while early stopping at the number of epochs of 1 round in COLOREDMNIST-025 to prevent over-fitting. All methods adopted the same backbone and the same training protocol following previous works [16, 83]. More details are given in Appendix F.1.

The results are reported in Table 1. It can be found that ERM will learn insufficiently good features under both stronger spurious correlations and invariant correlations, confirming our discussion in Sec. 4.3. Besides, Bonsai learns richer features in COLOREDMNIST-025 and boosts OOD performance, but Bonsai sometimes leads to suboptimal performances in COLOREDMNIST-01, which could be caused by the unstable feature learning in Bonsai. In contrast, FeAT consistently improves the OOD performance of all OOD objectives for all the COLOREDMNIST datasets, demonstrating the advances of direct feature learning control in FeAT than Bonsai and ERM.

**Experiments on real-world benchmarks.** We also compare FeAT with ERM and Bonsai in 6 real-world OOD generalization datasets curated by Koh et al. [39] that contain complicated features and distribution shifts. The learned features are evaluated with several representative state-of-the-art OOD objectives in WILDS, including GroupDro [64], IRMv1 [4], VREx [41] as well as IRMX [16]. By default, we train ERM, Bonsai and FeAT the same number of steps, and kept the rounds of Bonsai and FeAT the same (though Bonsai still requires one more round for feature synthesis). The only exception is in RXRX1 where both Bonsai and FeAT required more steps than ERM to converge. We use the same evaluation protocol following the practice in the literature [16, 39, 69, 83] to ensure a fair comparison. More details are given in Appendix F.2.

In addition to OOD objectives, we evaluate the learned features with Deep Feature Reweighting (DFR) [38]. DFR uses an additional OOD validation set where the *spurious correlation does not hold*, to perform logistic regression based on the learned features. Intuitively, DFR can serve as a proper measure for the quality of learned invariant features [33]. When the original dataset does not provide a proper OOD validation set, e.g., CAMELYON17, we use an alternative implementation based on a random split of the training and test data to perform the invariant feature quality measure [63]. Similarly, we also report DFR-s by regression with the environment labels (when available) to evaluate the spurious feature learning of different methods. More details are given in Appendix F.2.2.

The results are presented in Table 2. Similarly, when the tasks grow more challenging and neural architectures become more complicated, the ERM learned features can have a lower quality as discussed Sec. 4.3. For example, ERM can not sufficiently learn all useful features in FMoW, while ERM can learn more spurious correlations in CivilComments. Moreover, it can also be observed the instability of Bonsai in learning richer features that Bonsai even underperforms ERM in rich feature learning and OOD generalization in multiple datasets. In con-

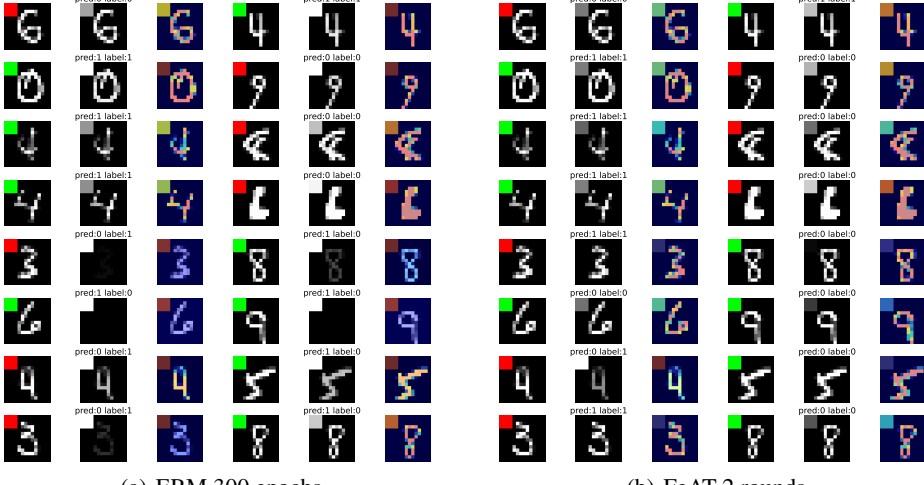

| (a) ERM 300 epochs | (b) FeAT 2 rounds |

Figure 3: GradCAM visualization on COLOREDMNIST-025, where the shortcuts are now concentrated to a colored path at the up left. Three visualizations are drawn for each sample: the original figure, the gray-colored gradcam, and the gradcam. It can be found that ERM can not properly capture the desired features while FeAT can stably capture the desired features.

trast, FeAT consistently achieves the best invariant feature learning performance across various challenging realistic datasets. Meanwhile, compared to ERM and Bonsai, FeAT also reduces over-fitting to the spurious feature learning led by spurious correlations. As a result, FeAT achieves consistent improvements when the learned features are applied to various OOD objectives.

**The termination check in FeAT.** As elaborated in Sec. 5.2, a key difference between FeAT and previous rich feature learning algorithms such as Bonsai is that FeAT is able to access the intermediate feature representa-

Table 3: Performances at different FeAT rounds.

| COLOREDMNIST-025 | ROUND-1 | ROUND-2 | ROUND-3 |
|---|---|---|---|
| TRAINING ACC. | 85.08± 0.14 | 71.87± 0.96 | 84.93± 1.26 |
| RETENTION ACC. | - | 88.11± 4.28 | 43.82± 0.59 |
| OOD ACC. | 11.08± 0.30 | 70.64± 0.62 | 10.07± 0.26 |

tions and thus can perform the automatic termination check and learn the desired features stably. To verify, we list the FeAT performances in various subsets of COLOREDMNIST-025 at different rounds in Table 3. By inspecting the retention accuracy, after FeAT learns sufficiently good features at Round 2, it is not necessary to proceed with Round 3 as it will destroy the already learned features and lead to degenerated retention and OOD performance. More details and results are given in Appendix F.1.

**Computational analysis.** We also analyze the computational and memory overhead of different methods, for which the details are given in Appendix F.4. Compared to ERM and Bonsai, iFeAT achieves the best performance without introducing too much additional overhead.

**Feature learning analysis.** We visualize the feature learning of ERM and FeAT on ColoredMNIST-025. As shown in Fig. 3, ERM can learn both invariant and spurious features to predict the label, aligned with our theory. However, ERM focuses more on spurious features and even forgets certain features with longer training epochs, which could be due to multiple reasons such as the simplicity biases of ERM. Hence predictions based on ERM learned features fail to generalize to OOD examples. In contrast, FeAT effectively captures the meaningful features for all samples and generalizes to OOD examples well. More analysis including results on WILDS benchmark can be found in Appendix F.5.

## 7 Conclusions

In this paper, we conducted a theoretical investigation of the invariant and spurious feature learning of ERM and OOD objectives. We found that ERM learns both invariant and spurious features when OOD objectives rarely learn new features. Thus, the features learned in the ERM pre-training can greatly influence the final OOD performance. Having learned the limitations of ERM pre-training, we proposed FeAT to learn all potentially useful features. Our extensive experimental results verify that FeAT significantly boosts the OOD performance when used for OOD training.

## Acknowledgements

We thank the reviewers for their valuable comments. This work was supported by the RIKEN Incentive Research Project 100847-202301062011, by CUHK direct grant 4055146. BH was supported by the NSFC Young Scientists Fund No. 62006202, NSFC General Program No. 62376235, Guangdong Basic and Applied Basic Research Foundation No. 2022A1515011652, HKBU Faculty Niche Research Areas No. RC-FNRA-IG/22-23/SCI/04, and Tencent AI Lab Rhino-Bird Gift Fund.

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

# Appendix of FeAT

## Contents

# A    Notations

We use bold-faced letters for vectors and matrices otherwise for scalar. We use $\|\cdot\|_2$ to denote the Euclidean norm of a vector or the spectral norm of a matrix, while denoting $\|\cdot\|_F$ as the Frobenius norm of a matrix. For a neural network, we denote $\psi(x)$ as the activation function. Let $\mathbf{I}_d$ be the identity matrix with a dimension of $\mathbb{R}^{d\times d}$. When comparing two sequences $\{a_n\}$ and $\{b_n\}$, we employ standard asymptotic notations such as $O(\cdot)$, $o(\cdot)$, $\Omega(\cdot)$, and $\Theta(\cdot)$ to describe their limiting behavior. Lastly, sequences of integers are denoted as $[n] = \{1, 2, \ldots, n\}$.

Table 4: Notations for key concepts involved in this paper

| Symbols | Definitions |
|---|---|
| $\mathcal{X} = \mathbb{R}^n$ | the input space |
| $\mathcal{Y} = \mathbb{R}$ | the label space |
| $\mathcal{Z} = \mathbb{R}^d$ | the latent space |
| $m$ | the hidden dimension |
| $F_j(\cdot)$ | the $j$-th filter of the CNN model |
| $\mathbf{W}_j$ | the weights of $j$-th filter of the CNN model, containing $m$ hidden units $\mathbf{w}_{j,r}$ |
| $\psi(\cdot)$ | the activation function of the CNN model |
| $\varphi$ | the featurizer $\varphi : \mathcal{X} \to \mathcal{Z}$ learns a latent representation for each input example |
| $w$ | the classifier $w : \mathcal{Z} \to \mathcal{Y}$ |
| $w_j$ | the classifier learned at $j$-th round |
| $f \in \mathcal{F}$ | the predictor $f = w \circ \varphi : \mathcal{X} \to \mathcal{Y}$ is composed of a featurizer and classifier when $w$ is linear, $f$ can be simply represented via dot product $w \cdot \varphi$ |
| $\mathcal{E}_{\text{all}}$ | the set of indices for all environments |
| $\mathcal{E}_{\text{tr}}$ | the subset of indices of training environments |
| $e$ | the index set of a specific environment |
| $\mathcal{E}_\alpha$ | the set of environments following the data model as Def. 3.1, where each is specified as $(\alpha, \beta_e)$ |
| $\mathcal{D}^e, \mathcal{D}_e$ | the dataset from environment $e$, containing $n_e$ samples $\{\mathbf{x}_i^e, y_i^e\}$ considered as i.i.d. from $\mathbb{P}^e$ |
| $\mathcal{D}$ | the overall dataset containing $n$ samples from all environments, $\mathcal{D} = \{\mathcal{D}^e\}_{e\in\mathcal{E}_{\text{all}}}$ |
| $\mathcal{D}^a$ | the augmentation set, we use $\mathcal{D}_i^a$ to denote the augmentation set separated at $i$-th round |
| $\mathcal{D}^r$ | the retention set, we use $\mathcal{D}_i^r$ to denote the retention set separated at $i$-th round |
| $G$ | $G = \{G^r, G^a\}$ with $2k - 1$ groups at round $k$, where $G^a = \{\mathcal{D}_i^a\}_{i=0}^{k-1}$ is the grouped sets, for new feature augmentation and $G^r = \{\mathcal{D}_i^r\}_{i=1}^{k-1}$ is the grouped sets for already learned feature retention |
| $L_e$ | the empirical risk calculated based on $\mathcal{D}^e$, e.g., square loss or logistic loss |
| $\ell_{\text{FeAT}}$ | the FeAT objective, including $\ell_{\mathcal{D}_i^a}$ the empirical risk at $\mathcal{D}_i^a$ and $\ell_{\mathcal{D}_i^r}$ at $\mathcal{D}_i^r$ |
| $L_{\text{IRMv1}}(\mathbf{W})$ | the IRMv1 loss |
| $\ell'^e$ | the first order derivative of $L_e$ with respect to the $i$-th sample from environment $e$ |
| $C_{\text{IRMv1}}^e$ | $C_{\text{IRMv1}}^e \triangleq \frac{1}{n_e}\sum_{i=1}^{n_e} \ell'\left(y_i^e \hat{y}_i^e\right) \cdot y_i^e \hat{y}_i^e$, a useful quantity to analyze IRMv1 dynamics |
| $\gamma_{j,r}^{inv}, \gamma_{j,r,1}$ | the invariant feature learning quantity in Eq. 6 |
| $\gamma_{j,r}^{spu}, \gamma_{j,r,2}$ | the spurious feature learning quantity in Eq. 6 |
| $\rho_{j,r,i}(t)$ | the noise feature learning quantity in Eq. 6 |

# B  Limitations and Future Directions

As a pioneering work that studies feature learning of ERM and OOD objectives and their interactions in OOD generalization, our theoretical settings are limited to studying the influence of spurious and invariant correlation strengths on spurious and invariant feature learning, based on a two-layer CNN network. In fact, the feature learning of a network can be influenced by several other factors, such as the difficulty of learning a feature and the capacity of features that a model can learn [21, 28]. Future works can be built by extending our framework to consider the influence of a broad of factors on feature learning in OOD generalization.

Moreover, as there could exist cases where certain features should not be learned, it is also promising to explore how to prevent the feature learning of undesirable features during the early stages of OOD generalization and to further relieve the optimization dilemma in OOD generalization [16], to improve the robustness against backdoor attacks [48], and its further implications to OOD generalization [45]. Besides, it is also interesting to investigate feature learning for complicated data such as graphs [32], especially under various graph distribution shifts [14, 15, 17, 34, 77].

# C  Related Work

**On Feature Learning and Generalization.** Understanding feature learning in deep networks is crucial to understanding their generalization [2, 11, 12, 22, 62, 70]. Earlier attempts are mostly about empirical probing [21, 26, 28, 65]. Elhage et al. [21], Hermann and Lampinen [28], Shah et al. [67] find that the feature learning of a network can be influenced by several other factors, such as the difficulty of learning a feature and the capacity of features that a model can learn. Although our data model focuses on the correlation perspective, different correlation strengths in fact can simulate the difficulty or the simplicity of learning a feature.

Beyond the empirical probing, Allen-Zhu and Li [2] proposed a new theoretical framework that characterizes the feature learning process of deep networks, which has been widely adopted to analyze behaviors of deep networks [12, 78, 86] However, how the learned features from ID data can be generalized to OOD data remains elusive. The only exceptions are [68] and [42]. Kumar et al. [42] find fine-tuning can distort the pre-trained features while fine-tuning can be considered as a special case in our framework. Shen et al. [68] focus on how data augmentation helps promote good but hard-to-learn features and improve OOD generalization. Deng et al. [20] studies feature learning when the group-related features are more predictive for inferring group labels. In contrast, we study the direct effects of ERM and OOD objectives to feature learning and provide a theoretical explanation to the phenomenon that ERM may have already learned good features [33, 63]. To the best of our knowledge, we are the *first* to analyze the feature learning of ERM and OOD objectives and their interactions in the general OOD generalization setting.

**On the correlation between ID and OOD performances.** The debate about feature learning and generalization under distribution shifts also extends to the ID and OOD performance correlations along with training or fine-tuning neural nets across a variety of OOD generalization tasks. Andreassen et al. [3], Miller et al. [47], Wenzel et al. [79] found that there often exists a linear dependency between ID and OOD performance under a wide range of models and distribution shifts. While Kumar et al. [42], Wortsman et al. [80] found that fine-tuning pre-trained models often leads to an increased in-distribution but decreased OOD performance. Teney et al. [74] observed cases where ID and OOD performance are inversely correlated. Chen et al. [16], Naganuma et al. [49] studied the ID and OOD performance trade-offs from the optimization perspective.

Our work provides theoretical explanations for different correlation behaviors of ID and OOD performance, as well as provides a solution for mitigating the trade-offs in optimization. Theorem 4.1 implies that, in cases where invariant features are more informative than spurious features, the higher ID performance indicates a better fit to invariant features, thus promising a higher OOD performance, aligned with observations in [3, 47, 79]. While in cases where invariant features are less informative than spurious features, the higher ID performance implies a better fit to spurious features, thus bringing a lower OOD performance [74]. Similarly, when fine-tuning a pre-trained model, if the model does not learn the features sufficiently well, ID-OOD performance will be in a positive correlation. However, when spurious correlations are present as easy-to-learn features, ERM can lead to a better fit for spurious features and distort the previously learned invariant features [42, 46, 80].

**Rich Feature Learning.** Recently many OOD objectives have been proposed to regularize ERM such that the model can focus on learning invariant features [4, 41, 55, 57, 76]. However, due to the intrinsic conflicts of ERM and OOD objectives, it often requires exhaustive hyperparameter tuning of ERM pre-training epochs and regularization weights [16, 83]. Especially, the final OOD performance has a large dependence on the number of pre-training epochs. To remedy the issue, Zhang et al. [83] proposed Bonsai to construct rich feature representations with plentiful potentially useful features such as network initialization. Although both Bonsai and FeAT perform DRO on grouped subsets, Bonsai rely on multiple initializations of the whole network to capture diverse features from the subsets, and complicated ensembling of the features, which requires much more training epochs for the convergence. In contrast, FeAT relieves the requirements by performing direct augmentation-retention on the grouped subsets, and thus obtains better performance. More crucially, although Bonsai and other rich feature learning algorithms such as weight averaging [5, 59, 82] have gained impressive successes in mitigating the dilemma, explanations about the reliance on ERM pre-training and why rich feature learning mitigates the dilemma remain elusive. Our work provides novel theoretical explanations for the success of rich feature learning algorithms for OOD generalization. Complementary to the empirical observations made by existing works, our work provides the first theoretical explanation for the feature learning of ERM and OOD objectives for OOD generalization.

Besides, there exists a rich literature on learning diverse representations for better generalization. Similar to weight average [59], Teney et al. [73] propose to train diverse models to resolve simplicity bias. Lee et al. [44] propose to learn diverse solutions for the underspecified learning problem. Nicolicioiu et al. [52] propose to regularize attention heads in transformers to learn diverse features. Chen et al. [13] propose to learn diverse classifiers for sample efficient domain adaption.

# D  Proofs for theoretical results

## D.1  Implementation details of the synthetic CNN experiments

For linear activation function $\psi(x) = x$, the logit $\hat{y}_i^e$ (which is a function of $\mathbf{W}$) of sample $i$ in the environment $e$ can be explicitly written as

$$\hat{y}_i^e = f(\mathbf{W}, \mathbf{x}_i^e) = F_{+1}(\mathbf{W}_{+1}, \mathbf{x}_i^e) - F_{-1}(\mathbf{W}_{-1}, \mathbf{x}_i^e) = \sum_{j \in \{\pm 1\}} \frac{j}{m} \sum_{r=1}^{m} \left[ \mathbf{w}_{j,r}^\top (\mathbf{x}_{i,1}^e + \mathbf{x}_{i,2}^e) \right],$$

where $\mathbf{W} \triangleq \{\mathbf{W}_{+1}, \mathbf{W}_{-1}\}$ and $\mathbf{W}_j \triangleq \begin{bmatrix} \mathbf{w}_{j,1}^\top \\ \vdots \\ \mathbf{w}_{j,m}^\top \end{bmatrix}$ for $j \in \{\pm 1\}$. We initialized all the network

weights as $\mathcal{N}(0, \sigma_0^2)$ and we set $\sigma_0 = 0.01$.

The test dataset $(\mathbf{x}, y)$ is generated through

$$\mathbf{x}_{i,1} = y_i \cdot \mathbf{v}_1 + y_i \cdot \mathrm{Rad}(1 - \beta_e) \cdot \mathbf{v}_2, \qquad \mathbf{x}_{i,2} = \boldsymbol{\xi},$$

where half of the dataset uses $\mathrm{Rad}(1 - \beta_1)$ and the other half uses $\mathrm{Rad}(1 - \beta_2)$. Here $\boldsymbol{\xi} \sim \mathcal{N}(0, \sigma_p^2 \cdot (\mathbf{I}_d - \mathbf{v}_1 \mathbf{v}_1^\top - \mathbf{v}_2 \mathbf{v}_2^\top))$ and we chose $\sigma_p = 0.01$.

From the definition of IRMv1, we take derivative wrt. the scalar 1 of the logit $1 \cdot \hat{y}_i^e$. Thus, for environment $e$, the penalty is

$$\left( \frac{1}{n_e} \sum_{i=1}^{n_e} \nabla_{w|w=1} \ell \left( y_i^e (w \cdot \hat{y}_i^e) \right) \right)^2 = \left( \frac{1}{n_e} \sum_{i=1}^{n_e} \ell' \left( y_i^e \hat{y}_i^e \right) \cdot y_i^e \hat{y}_i^e \right)^2.$$

Then, the IRMv1 objective is (we set $n_1 = n_2 = 2500$ in the simulation)

$$L_{\mathrm{IRMv1}}(\mathbf{W}) = \sum_{e \in \mathcal{E}_{tr}} \frac{1}{n_e} \sum_{i=1}^{n_e} \ell \left( y_i^e \hat{y}_i^e \right) + \lambda \sum_{e \in \mathcal{E}_{tr}} \left( \frac{1}{n_e} \sum_{i=1}^{n_e} \ell' \left( y_i^e \hat{y}_i^e \right) \cdot y_i^e \hat{y}_i^e \right)^2.$$

We used constant stepsize GD to minimize $L_{\mathrm{IRMv1}}(\mathbf{W})$, and we chose $\lambda = 10^8$ (heavy regularization setup).

Let $C_{\text{IRMv1}}^e \triangleq \frac{1}{n_e} \sum_{i=1}^{n_e} \ell'\left(y_i^e \hat{y}_i^e\right) \cdot y_i^e \hat{y}_i^e$. The gradient of $L_{\text{IRMv1}}(\mathbf{W})$ with respect to each $\mathbf{w}_{j,r}$ can be explicitly written as

$$\nabla_{\mathbf{w}_{j,r}} L_{\text{IRMv1}}(\mathbf{W})$$

$$= \sum_{e \in \mathcal{E}_{tr}} \frac{1}{n_e} \sum_{i=1}^{n_e} \ell'\left(y_i^e \hat{y}_i^e\right) \cdot y_i^e \cdot \frac{j}{m}(\mathbf{x}_{i,1}^e + \mathbf{x}_{i,2}^e)$$

$$+ 2\lambda \sum_{e \in \mathcal{E}_{tr}} \frac{C_{\text{IRMv1}}^e}{n_e} \sum_{i=1}^{n_e} \left( \ell''\left(y_i^e \hat{y}_i^e\right) \cdot \hat{y}_i^e \cdot \frac{j}{m}(\mathbf{x}_{i,1}^e + \mathbf{x}_{i,2}^e) + \ell'\left(y_i^e \hat{y}_i^e\right) \cdot y_i^e \cdot \frac{j}{m}(\mathbf{x}_{i,1}^e + \mathbf{x}_{i,2}^e) \right)$$

$$= \sum_{e \in \mathcal{E}_{tr}} \frac{j}{n_e m} \sum_{i=1}^{n_e} \ell'\left(y_i^e \hat{y}_i^e\right) \cdot y_i^e \cdot (\mathbf{x}_{i,1}^e + \mathbf{x}_{i,2}^e)$$

$$+ 2\lambda \sum_{e \in \mathcal{E}_{tr}} \frac{j C_{\text{IRMv1}}^e}{n_e m} \sum_{i=1}^{n_e} \ell''\left(y_i^e \hat{y}_i^e\right) \cdot \hat{y}_i^e \cdot (\mathbf{x}_{i,1}^e + \mathbf{x}_{i,2}^e)$$

$$+ 2\lambda \sum_{e \in \mathcal{E}_{tr}} \frac{j C_{\text{IRMv1}}^e}{n_e m} \sum_{i=1}^{n_e} \ell'\left(y_i^e \hat{y}_i^e\right) \cdot y_i^e \cdot (\mathbf{x}_{i,1}^e + \mathbf{x}_{i,2}^e)$$

$$= \sum_{e \in \mathcal{E}_{tr}} \frac{j(1 + 2\lambda C_{\text{IRMv1}}^e)}{n_e m} \sum_{i=1}^{n_e} \ell'\left(y_i^e \hat{y}_i^e\right) \cdot y_i^e \cdot (\mathbf{x}_{i,1}^e + \mathbf{x}_{i,2}^e)$$

$$+ 2\lambda \sum_{e \in \mathcal{E}_{tr}} \frac{j C_{\text{IRMv1}}^e}{n_e m} \sum_{i=1}^{n_e} \ell''\left(y_i^e \hat{y}_i^e\right) \cdot \hat{y}_i^e \cdot (\mathbf{x}_{i,1}^e + \mathbf{x}_{i,2}^e).$$

Observe that $C_{\text{IRMv1}}^e$ is in fact the scalar gradient $C_{\text{IRMv1}}^e = \nabla_{w|w=1} L_{\text{ERM}}^e(\mathbf{W})$ that we want to force zero, whose effect can be understood as a dynamic re-weighting of the ERM gradient. Due to its importance in the analysis and interpretation of IRMv1, we tracked $C_{\text{IRMv1}}^e$ in our simulations.

The invariant and spurious feature learning terms that we tracked are the mean of $\langle \mathbf{w}_{j,r}, j\mathbf{v}_1 \rangle$ and $\langle \mathbf{w}_{j,r}, j\mathbf{v}_2 \rangle$ for $j \in \{\pm 1\}, r \in [m]$, respectively.

## D.2 Proof for Theorem 4.1

**Theorem D.1** (Formal statement of Theorem 4.1). *For $\rho > 0$, denote $\underline{n} \triangleq \min_{e \in \mathcal{E}_{tr}} n_e$, $n \triangleq \sum_{e \in \mathcal{E}_{tr}} n_e$, $\epsilon_C \triangleq \sqrt{\frac{2 \log (16/\rho)}{\underline{n}}}$ and $\delta \triangleq \exp\{O(\underline{n}^{-1})\} - 1$. Define the feature learning terms $\Lambda_{j,r}^t \triangleq \langle \mathbf{w}_{j,r}^t, j\mathbf{v}_1 \rangle$ and $\Gamma_{j,r}^t \triangleq \langle \mathbf{w}_{j,r}^t, j\mathbf{v}_2 \rangle$ for $j \in \{\pm 1\}, r \in [m]$. Suppose we run $T$ iterations of GD for the ERM objective. With sufficiently large $\underline{n}$ and $\psi(x) = x$, assuming that*

$$\alpha, \beta_1, \beta_2 < \frac{1 - \epsilon_C - \delta(\frac{1}{4} + \frac{\epsilon_C}{2})}{2} \quad (\alpha, \beta_1, \beta_2 \text{ are sufficiently smaller than } \frac{1}{2}),$$

$$\alpha > \frac{\beta_1 + \beta_2}{2} + \epsilon_C + \frac{\delta(1 + \epsilon_C)}{2} \quad (\alpha \text{ is sufficiently larger than } \frac{\beta_1 + \beta_2}{2}),$$

*and choosing*

$$\sigma_0^2 = O\left(\underline{n}^{-2} \log^{-1}(m/\rho)\right),$$

$$\sigma_p^2 = O\left(\min\left\{d^{-1/2}\log^{-1/2}(nm/\rho), T^{-1}\eta^{-1}m\left(d + n\sqrt{d\log(n^2/\rho)}\right)^{-1}\right\}\right),$$

*there exists a constant $\eta$, such that for any $j \in \{\pm 1\}, r \in [m]$, with probability at least $1 - 2\rho$, $\Lambda_{j,r}^t$ and $\Gamma_{j,r}^t$ are converging and the increment of the spurious feature $\Gamma_{j,r}^{t+1} - \Gamma_{j,r}^t$ is larger than that of the invariant feature $\Lambda_{j,r}^{t+1} - \Lambda_{j,r}^t$ at any iteration $t \in \{0, \ldots, T-1\}$.*

*Proof of Theorem D.1.* We begin with checking the feature learning terms in the ERM stage using constant stepsize GD: $\mathbf{W}^{t+1} = \mathbf{W}^t - \eta \cdot \nabla_{\mathbf{W}} L_{\text{IRMv1}}(\mathbf{W}^t)$. Note that with $\psi(x) = x$ the update rule for each $\mathbf{w}_{j,r}, \forall j \in \{+1, -1\}, r \in [m]$ can be written as

$$\mathbf{w}_{j,r}^{t+1} = \mathbf{w}_{j,r}^t - \frac{j\eta}{m} \sum_{e \in \mathcal{E}_{tr}} \frac{1}{n_e} \sum_{i=1}^{n_e} \ell'\left(y_i^e \hat{y}_i^e\right) \cdot y_i^e \cdot (\mathbf{x}_{i,1}^e + \mathbf{x}_{i,2}^e)$$

$$= \mathbf{w}_{j,r}^t - \frac{j\eta}{m} \sum_{e \in \mathcal{E}_{tr}} \frac{1}{n_e} \sum_{i=1}^{n_e} \ell'\left(y_i^e \hat{y}_i^e\right) \cdot (\text{Rad}(\alpha)_i \cdot \mathbf{v}_1 + \text{Rad}(\beta_e)_i \cdot \mathbf{v}_2 + y_i^e \boldsymbol{\xi}_i^e).$$

Define the quantities of interest (the feature learning terms): $\Lambda_{j,r}^t \triangleq \langle \mathbf{w}_{j,r}^t, j\mathbf{v}_1 \rangle, \Gamma_{j,r}^t \triangleq \langle \mathbf{w}_{j,r}^t, j\mathbf{v}_2 \rangle, \Xi_{j,r,i}^{t,e} \triangleq \langle \mathbf{w}_{j,r}^t, j\boldsymbol{\xi}_i^e \rangle$. From our data generating procedure (Definition 3.1), we know that the first two coordinates of $\boldsymbol{\xi}_i^e$ are zero. Thus, we can write down the update rule for each feature learning term as follows.

$$\Lambda_{j,r}^{t+1} = \Lambda_{j,r}^t - \frac{\eta}{m} \sum_{e \in \mathcal{E}_{tr}} \frac{1}{n_e} \sum_{i=1}^{n_e} \ell'\left(y_i^e \hat{y}_i^e\right) \cdot \text{Rad}(\alpha)_i,$$

$$\Gamma_{j,r}^{t+1} = \Gamma_{j,r}^t - \frac{\eta}{m} \sum_{e \in \mathcal{E}_{tr}} \frac{1}{n_e} \sum_{i=1}^{n_e} \ell'\left(y_i^e \hat{y}_i^e\right) \cdot \text{Rad}(\beta_e)_i,$$

$$\Xi_{j,r,i'}^{t+1,e'} = \Xi_{j,r,i'}^{t,e'} - \frac{\eta}{m} \sum_{e \in \mathcal{E}_{tr}} \frac{1}{n_e} \sum_{i=1}^{n_e} \ell'\left(y_i^e \hat{y}_i^e\right) \cdot y_i^e \cdot \langle \boldsymbol{\xi}_i^e, \boldsymbol{\xi}_{i'}^{e'} \rangle.$$

More explicitly, we can write

$$\Lambda_{j,r}^{t+1} = \Lambda_{j,r}^t + \frac{\eta}{m} \sum_{e \in \mathcal{E}_{tr}} \frac{1}{n_e} \sum_{i=1}^{n_e} \frac{\text{Rad}(\alpha)_i}{1 + \exp\{y_i^e \hat{y}_i^e\}}, \tag{8}$$

$$\Gamma_{j,r}^{t+1} = \Gamma_{j,r}^t + \frac{\eta}{m} \sum_{e \in \mathcal{E}_{tr}} \frac{1}{n_e} \sum_{i=1}^{n_e} \frac{\text{Rad}(\beta_e)_i}{1 + \exp\{y_i^e \hat{y}_i^e\}}, \tag{9}$$

$$\Xi_{j,r,i'}^{t+1,e'} = \Xi_{j,r,i'}^{t,e'} + \frac{\eta}{m} \sum_{e \in \mathcal{E}_{tr}} \frac{1}{n_e} \sum_{i=1}^{n_e} \frac{y_i^e \cdot \langle \boldsymbol{\xi}_i^e, \boldsymbol{\xi}_{i'}^{e'} \rangle}{1 + \exp\{y_i^e \hat{y}_i^e\}}. \tag{10}$$

Notice that the updates (8), (9) for $\Lambda_{j,r}, \Gamma_{j,r}$ are independent of $j, r$. Denoting

$$\Delta_\Lambda^t \triangleq \frac{1}{m} \sum_{e \in \mathcal{E}_{tr}} \frac{1}{n_e} \sum_{i=1}^{n_e} \frac{\text{Rad}(\alpha)_i}{1 + \exp\{y_i^e \hat{y}_i^e\}},$$

$$\Delta_\Gamma^t \triangleq \frac{1}{m} \sum_{e \in \mathcal{E}_{tr}} \frac{1}{n_e} \sum_{i=1}^{n_e} \frac{\text{Rad}(\beta_e)_i}{1 + \exp\{y_i^e \hat{y}_i^e\}},$$

we can conclude that for any $j \in \{+1, -1\}, r \in [m]$,

$$\Lambda_{j,r}^{t+1} = \Lambda_{j,r}^t + \eta \cdot \Delta_\Lambda^t = \eta \cdot \sum_{k=0}^t \Delta_\Lambda^k + \Lambda_{j,r}^0,$$

$$\Gamma_{j,r}^{t+1} = \Gamma_{j,r}^t + \eta \cdot \Delta_\Gamma^t = \eta \cdot \sum_{k=0}^t \Delta_\Gamma^k + \Gamma_{j,r}^0.$$

(11)

Then, we write the logit $\hat{y}_i^e$ as

$$\hat{y}_i^e = \sum_{j \in \{\pm 1\}} \frac{j}{m} \sum_{r=1}^m \left[ \langle \mathbf{w}_{j,r}^t, y_i^e \cdot \text{Rad}(\alpha)_i \cdot \mathbf{v}_1 + y_i^e \cdot \text{Rad}(\beta_e)_i \cdot \mathbf{v}_2 + \mathbf{x}_{i,2}^e \rangle \right]$$

$$= \sum_{j \in \{\pm 1\}} \frac{j}{m} \sum_{r=1}^m \left[ j y_i^e \cdot \text{Rad}(\alpha)_i \cdot \Lambda_{j,r}^t + j y_i^e \cdot \text{Rad}(\beta_e)_i \cdot \Gamma_{j,r}^t + j \cdot \Xi_{j,r,i}^{t,e} \right]$$

$$= \sum_{j \in \{\pm 1\}} \frac{1}{m} \sum_{r=1}^m \left[ y_i^e \cdot \text{Rad}(\alpha)_i \cdot \Lambda_{j,r}^t + y_i^e \cdot \text{Rad}(\beta_e)_i \cdot \Gamma_{j,r}^t + \Xi_{j,r,i}^{t,e} \right]$$

$$= y_i^e \cdot \text{Rad}(\alpha)_i \cdot \sum_{j \in \{\pm 1\}} \sum_{r=1}^m \frac{\Lambda_{j,r}^t}{m} + y_i^e \cdot \text{Rad}(\beta_e)_i \cdot \sum_{j \in \{\pm 1\}} \sum_{r=1}^m \frac{\Gamma_{j,r}^t}{m} + \sum_{j \in \{\pm 1\}} \sum_{r=1}^m \frac{\Xi_{j,r,i}^{t,e}}{m}$$

$$= y_i^e \cdot \text{Rad}(\alpha)_i \cdot 2\eta \cdot \sum_{k=0}^{t-1} \Delta_\Lambda^k + y_i^e \cdot \text{Rad}(\beta_e)_i \cdot 2\eta \cdot \sum_{k=0}^{t-1} \Delta_\Gamma^k$$

$$+ y_i^e \cdot \text{Rad}(\alpha)_i \cdot \sum_{j \in \{\pm 1\}} \sum_{r=1}^m \frac{\Lambda_{j,r}^0}{m} + y_i^e \cdot \text{Rad}(\beta_e)_i \cdot \sum_{j \in \{\pm 1\}} \sum_{r=1}^m \frac{\Gamma_{j,r}^0}{m} + \sum_{j \in \{\pm 1\}} \sum_{r=1}^m \frac{\Xi_{j,r,i}^{t,e}}{m}.$$

Denoting $\mathbb{Q}_i^e \triangleq \text{Rad}(\alpha)_i \sum_{j \in \{\pm 1\}} \sum_{r=1}^m \frac{\Lambda_{j,r}^0}{m} + \text{Rad}(\beta_e)_i \sum_{j \in \{\pm 1\}} \sum_{r=1}^m \frac{\Gamma_{j,r}^0}{m} + y_i^e \cdot \sum_{j \in \{\pm 1\}} \sum_{r=1}^m \frac{\Xi_{j,r,i}^{t,e}}{m}$, we have

$$\hat{y}_i^e = y_i^e \cdot \left( \text{Rad}(\alpha)_i \cdot 2\eta \cdot \sum_{k=0}^{t-1} \Delta_\Lambda^k + \text{Rad}(\beta_e)_i \cdot 2\eta \cdot \sum_{k=0}^{t-1} \Delta_\Gamma^k + \mathbb{Q}_i^e \right),$$

We need the following concentration lemma to control the scale of $\mathbb{Q}_i^e$, whose proof is given in Appendix D.2.1.

**Lemma D.2.** *Denote* $\underline{n} \triangleq \min_{e \in \mathcal{E}_{tr}} n_e, n \triangleq \sum_{e \in \mathcal{E}_{tr}} n_e$. *For* $\rho > 0$, *if*

$$\sigma_0^2 = O\left(\underline{n}^{-2} \log^{-1}(m/\rho)\right),$$

$$\sigma_p^2 = O\left(\min\left\{ d^{-1/2} \log^{-1/2}(nm/\rho), T^{-1}\eta^{-1}m \left( d + n\sqrt{d \log(n^2/\rho)} \right)^{-1} \right\}\right),$$

*then with probability at least* $1 - \rho$, *for any* $e \in \mathcal{E}_{tr}, i \in [n_e]$, *it holds that* $|\mathbb{Q}_i^e| = O(\underline{n}^{-1})$.

Then $\Delta_\Lambda^t$ and $\Delta_\Gamma^t$ can be explicitly written as

$$\Delta_\Lambda^t =$$
$$\sum_{e \in \mathcal{E}_{tr}} \frac{1}{n_e m} \sum_{i=1}^{n_e} \frac{\text{Rad}(\alpha)_i}{1 + \exp\left\{\text{Rad}(\alpha)_i \cdot 2\eta \cdot \sum_{k=0}^{t-1} \Delta_\Lambda^k\right\} \cdot \exp\left\{\text{Rad}(\beta_e)_i \cdot 2\eta \cdot \sum_{k=0}^{t-1} \Delta_\Gamma^k\right\} \cdot \exp\left\{\mathbb{Q}_i^e\right\}},$$
$$\Delta_\Gamma^t =$$
$$\sum_{e \in \mathcal{E}_{tr}} \frac{1}{n_e m} \sum_{i=1}^{n_e} \frac{\text{Rad}(\beta_e)_i}{1 + \exp\left\{\text{Rad}(\alpha)_i \cdot 2\eta \cdot \sum_{k=0}^{t-1} \Delta_\Lambda^k\right\} \cdot \exp\left\{\text{Rad}(\beta_e)_i \cdot 2\eta \cdot \sum_{k=0}^{t-1} \Delta_\Gamma^k\right\} \cdot \exp\left\{\mathbb{Q}_i^e\right\}}.$$

We are going to analyze the convergences of two sequences $\{\Delta_\Gamma^t + \Delta_\Lambda^t\}$ and $\{\Delta_\Gamma^t - \Delta_\Lambda^t\}$. Notice that

$$\Delta_\Gamma^t + \Delta_\Lambda^t =$$
$$\sum_{e \in \mathcal{E}_{tr}} \frac{1}{n_e m} \sum_{i=1}^{n_e} \frac{\text{Rad}(\beta_e)_i + \text{Rad}(\alpha)_i}{1 + \exp\left\{\text{Rad}(\alpha)_i \cdot 2\eta \cdot \sum_{k=0}^{t-1} \Delta_\Lambda^k\right\} \cdot \exp\left\{\text{Rad}(\beta_e)_i \cdot 2\eta \cdot \sum_{k=0}^{t-1} \Delta_\Gamma^k\right\} \cdot \exp\left\{\mathbb{Q}_i^e\right\}},$$
$$\Delta_\Gamma^t - \Delta_\Lambda^t =$$
$$\sum_{e \in \mathcal{E}_{tr}} \frac{1}{n_e m} \sum_{i=1}^{n_e} \frac{\text{Rad}(\beta_e)_i - \text{Rad}(\alpha)_i}{1 + \exp\left\{\text{Rad}(\alpha)_i \cdot 2\eta \cdot \sum_{k=0}^{t-1} \Delta_\Lambda^k\right\} \cdot \exp\left\{\text{Rad}(\beta_e)_i \cdot 2\eta \cdot \sum_{k=0}^{t-1} \Delta_\Gamma^k\right\} \cdot \exp\left\{\mathbb{Q}_i^e\right\}}.$$

We can further write these two terms as

$$\Delta_\Gamma^t + \Delta_\Lambda^t = \sum_{e \in \mathcal{E}_{tr}} \frac{2}{n_e m} \sum_{\substack{i \in [n_e] \\ \text{Rad}(\beta_e)_i = +1 \\ \text{Rad}(\alpha)_i = +1}} \frac{1}{1 + \exp\left\{2\eta \cdot \sum_{k=0}^{t-1} \left(\Delta_\Gamma^k + \Delta_\Lambda^k\right)\right\} \cdot \exp\left\{\mathbb{Q}_i^e\right\}}$$
$$- \sum_{e \in \mathcal{E}_{tr}} \frac{2}{n_e m} \sum_{\substack{i \in [n_e] \\ \text{Rad}(\beta_e)_i = -1 \\ \text{Rad}(\alpha)_i = -1}} \frac{1}{1 + \exp\left\{-2\eta \cdot \sum_{k=0}^{t-1} \left(\Delta_\Gamma^k + \Delta_\Lambda^k\right)\right\} \cdot \exp\left\{\mathbb{Q}_i^e\right\}},$$
$$\Delta_\Gamma^t - \Delta_\Lambda^t = \sum_{e \in \mathcal{E}_{tr}} \frac{2}{n_e m} \sum_{\substack{i \in [n_e] \\ \text{Rad}(\beta_e)_i = +1 \\ \text{Rad}(\alpha)_i = -1}} \frac{1}{1 + \exp\left\{2\eta \cdot \sum_{k=0}^{t-1} \left(\Delta_\Gamma^k - \Delta_\Lambda^k\right)\right\} \cdot \exp\left\{\mathbb{Q}_i^e\right\}}$$
$$- \sum_{e \in \mathcal{E}_{tr}} \frac{2}{n_e m} \sum_{\substack{i \in [n_e] \\ \text{Rad}(\beta_e)_i = -1 \\ \text{Rad}(\alpha)_i = +1}} \frac{1}{1 + \exp\left\{-2\eta \cdot \sum_{k=0}^{t-1} \left(\Delta_\Gamma^k - \Delta_\Lambda^k\right)\right\} \cdot \exp\left\{\mathbb{Q}_i^e\right\}}.$$

According to Lemma D.2, for all $e \in \mathcal{E}_{tr}, i \in [n_e], \rho > 0$, letting $\delta \triangleq \exp\{O(\underline{n}^{-1})\} - 1$, we have $1 + \delta \geq \exp\left\{\mathbb{Q}_i^e\right\} \geq (1 + \delta)^{-1}$ with probability at least $1 - \rho$. Let $C_{j\ell}^e \triangleq |\{i \mid \text{Rad}(\alpha)_i = j, \text{Rad}(\beta_e)_i = \ell, i \in \mathcal{E}_e\}|$ for any $j \in \{\pm 1\}, \ell \in \{\pm 1\}, e \in \mathcal{E}_{tr}$, and then define $\overline{C}_{j\ell} \triangleq \sum_{e \in \mathcal{E}_{tr}} \frac{C_{j\ell}^e}{n_e}$.

We can upper bound and formulate $\Delta_\Gamma^t + \Delta_\Lambda^t$ and $\Delta_\Gamma^t - \Delta_\Lambda^t$ as

$$\Delta_\Gamma^t + \Delta_\Lambda^t \leq$$

$$\frac{2}{m}\left(\frac{\overline{C}_{+1+1}}{1 + \exp\left\{2\eta \cdot \sum_{k=0}^{t-1}\left(\Delta_\Gamma^k + \Delta_\Lambda^k\right)\right\} \cdot (1+\delta)^{-1}} - \frac{\overline{C}_{-1-1}}{1 + \exp\left\{-2\eta \cdot \sum_{k=0}^{t-1}\left(\Delta_\Gamma^k + \Delta_\Lambda^k\right)\right\} \cdot (1+\delta)}\right)$$

$$= \frac{2}{m} \cdot \frac{\overline{C}_{+1+1}(1+\delta) - \overline{C}_{-1-1} \cdot \exp\left\{2\eta \cdot \sum_{k=0}^{t-1}\left(\Delta_\Gamma^k + \Delta_\Lambda^k\right)\right\}}{1 + \delta + \exp\left\{2\eta \cdot \sum_{k=0}^{t-1}\left(\Delta_\Gamma^k + \Delta_\Lambda^k\right)\right\}}, \tag{12}$$

$$\Delta_\Gamma^t - \Delta_\Lambda^t \leq$$

$$\frac{2}{m}\left(\frac{\overline{C}_{-1+1}}{1 + \exp\left\{2\eta \cdot \sum_{k=0}^{t-1}\left(\Delta_\Gamma^k - \Delta_\Lambda^k\right)\right\} \cdot (1+\delta)^{-1}} - \frac{\overline{C}_{+1-1}}{1 + \exp\left\{-2\eta \cdot \sum_{k=0}^{t-1}\left(\Delta_\Gamma^k - \Delta_\Lambda^k\right)\right\} \cdot (1+\delta)}\right)$$

$$= \frac{2}{m} \cdot \frac{\overline{C}_{-1+1}(1+\delta) - \overline{C}_{+1-1} \cdot \exp\left\{2\eta \cdot \sum_{k=0}^{t-1}\left(\Delta_\Gamma^k - \Delta_\Lambda^k\right)\right\}}{1 + \delta + \exp\left\{2\eta \cdot \sum_{k=0}^{t-1}\left(\Delta_\Gamma^k - \Delta_\Lambda^k\right)\right\}}. \tag{13}$$

Based on similar arguments, we can also establish lower bounds for these two terms,

$$\Delta_\Gamma^t + \Delta_\Lambda^t \geq \frac{2}{m} \cdot \frac{\overline{C}_{+1+1} - \overline{C}_{-1-1}(1+\delta) \cdot \exp\left\{2\eta \cdot \sum_{k=0}^{t-1}\left(\Delta_\Gamma^k + \Delta_\Lambda^k\right)\right\}}{1 + \exp\left\{2\eta \cdot \sum_{k=0}^{t-1}\left(\Delta_\Gamma^k + \Delta_\Lambda^k\right)\right\} \cdot (1+\delta)}, \tag{14}$$

$$\Delta_\Gamma^t - \Delta_\Lambda^t \geq \frac{2}{m} \cdot \frac{\overline{C}_{-1+1} - \overline{C}_{+1-1}(1+\delta) \cdot \exp\left\{2\eta \cdot \sum_{k=0}^{t-1}\left(\Delta_\Gamma^k - \Delta_\Lambda^k\right)\right\}}{1 + \exp\left\{2\eta \cdot \sum_{k=0}^{t-1}\left(\Delta_\Gamma^k - \Delta_\Lambda^k\right)\right\} \cdot (1+\delta)}. \tag{15}$$

The upper and lower bounds (12), (13), (14) and (15) imply that the convergences of $\{\Delta_\Gamma^t + \Delta_\Lambda^t\}$ and $\{\Delta_\Gamma^t - \Delta_\Lambda^t\}$ are determined by recursive equations of the form $\mathcal{Q}^t = \frac{C_1 - C_2 \cdot \exp\left\{\eta \sum_{k=0}^{t-1} \mathcal{Q}^k\right\}}{1 + C_3 \cdot \exp\left\{\eta \sum_{k=0}^{t-1} \mathcal{Q}^k\right\}}$. We first establish that with suitably chosen $\eta$, the sequences $\{\Delta_\Gamma^t + \Delta_\Lambda^t\}$ and $\{\Delta_\Gamma^t - \Delta_\Lambda^t\}$ are guaranteed to be positive. Observed that for the $\mathcal{Q}^t$-type recursive equation, the sign of $\mathcal{Q}^0$ is independent of $\eta$, and only determined by the constants $C_1, C_2, C_3$. At iteration 0, (14) and (15) give

$$\Delta_\Gamma^0 + \Delta_\Lambda^0 \geq \frac{2}{m} \cdot \frac{\overline{C}_{+1+1} - \overline{C}_{-1-1}(1+\delta)}{2+\delta}, \tag{16}$$

$$\Delta_\Gamma^0 - \Delta_\Lambda^0 \geq \frac{2}{m} \cdot \frac{\overline{C}_{-1+1} - \overline{C}_{+1-1}(1+\delta)}{2+\delta}. \tag{17}$$

To proceed, we need the following concentration lemma to control the deviations of the constants $\overline{C}_{+1+1}, \overline{C}_{+1-1}, \overline{C}_{-1+1}$ and $\overline{C}_{-1-1}$ from their expectations, whose proof is given in Appendix D.2.2.

**Lemma D.3.** *For $\rho > 0$, considering two environments and denoting $\epsilon_C \triangleq \sqrt{\frac{2\log(16/\rho)}{n}}$, with probability at least $1 - \rho$, we have*

$$\begin{aligned}
\left|\overline{C}_{+1+1} - (1-\alpha)(2 - \beta_1 - \beta_2)\right| &\leq \epsilon_C, \\
\left|\overline{C}_{+1-1} - (1-\alpha)(\beta_1 + \beta_2)\right| &\leq \epsilon_C, \\
\left|\overline{C}_{-1+1} - \alpha(2 - \beta_1 - \beta_2)\right| &\leq \epsilon_C, \\
\left|\overline{C}_{-1-1} - \alpha(\beta_1 + \beta_2)\right| &\leq \epsilon_C.
\end{aligned} \tag{18}$$

Using Lemma D.3, with probability at least $1 - \rho$, the constants $\overline{C}_{+1+1}, \overline{C}_{+1-1}, \overline{C}_{-1+1}$ and $\overline{C}_{-1-1}$ are close to their expectations.

Based on our assumptions that

$$\alpha, \beta_1, \beta_2 < \frac{1 - \epsilon_C - \delta(\frac{1}{4} + \frac{\epsilon_C}{2})}{2} \quad (\alpha, \beta_1, \beta_2 \text{ are sufficiently smaller than } \frac{1}{2}),$$

$$\alpha > \frac{\beta_1 + \beta_2}{2} + \epsilon_C + \frac{\delta(1 + \epsilon_C)}{2} \quad (\alpha \text{ is sufficiently larger than } \frac{\beta_1 + \beta_2}{2}),$$

it can be verified that with probability at least $1 - 2\rho$, $\Delta_\Gamma^0 + \Delta_\Lambda^0 > 0$, $\Delta_\Gamma^0 - \Delta_\Lambda^0 > 0$.

Then, at iteration 1, from (14) and (15), we see that as long as we require

$$\eta < \min \left\{ \frac{1}{2(\Delta_\Gamma^0 + \Delta_\Lambda^0)} \log \frac{\overline{C}_{+1+1}}{\overline{C}_{-1-1}(1 + \delta)}, \frac{1}{2(\Delta_\Gamma^0 - \Delta_\Lambda^0)} \log \frac{\overline{C}_{-1+1}}{\overline{C}_{+1-1}(1 + \delta)} \right\},$$

it holds that $\Delta_\Gamma^1 + \Delta_\Lambda^1 > 0$, $\Delta_\Gamma^1 - \Delta_\Lambda^1 > 0$. By recursively applying this argument, we see the requirement for $\eta$ to ensure that $\Delta_\Gamma^t + \Delta_\Lambda^t > 0$ and $\Delta_\Gamma^t - \Delta_\Lambda^t > 0$ for any $t \in \{0, \ldots, T\}$ is

$$\eta < \min \left\{ \frac{1}{2 \sum_{k=0}^{T-1} (\Delta_\Gamma^k + \Delta_\Lambda^k)} \log \frac{\overline{C}_{+1+1}}{\overline{C}_{-1-1}(1 + \delta)}, \frac{1}{2 \sum_{k=0}^{T-1} (\Delta_\Gamma^k - \Delta_\Lambda^k)} \log \frac{\overline{C}_{-1+1}}{\overline{C}_{+1-1}(1 + \delta)} \right\}.$$

(19)

In other words, for the $\mathcal{Q}^t$-type recursive equation, as long as $\mathcal{Q}^0 \geq 0$, there always exists a sufficiently small $\eta$ to guarantee that the whole sequence $\{\mathcal{Q}^t\}$ is positive. From now on, we will focus on the case where the two sequences $\{\Delta_\Gamma^t + \Delta_\Lambda^t\}$ and $\{\Delta_\Gamma^t - \Delta_\Lambda^t\}$ decrease to an $\epsilon_\Delta > 0$ error, i.e., $\min_{t \in \{0,\ldots,T\}} \{\Delta_\Gamma^t + \Delta_\Lambda^t, \Delta_\Gamma^t - \Delta_\Lambda^t\} = \epsilon_\Delta$.

Then, we show that the two sequences $\{\Delta_\Gamma^t + \Delta_\Lambda^t\}$ and $\{\Delta_\Gamma^t - \Delta_\Lambda^t\}$ decrease monotonically, which thus leads to a more refined upper bound for $\eta$ at (19). Inspect the upper bounds (12), (13) at iteration $t + 1$, which can be written as

$$\Delta_\Gamma^{t+1} + \Delta_\Lambda^{t+1} \leq$$

$$\frac{2}{m} \cdot \frac{\overline{C}_{+1+1} - \overline{C}_{-1-1} \cdot \exp\left\{2\eta \cdot \sum_{k=0}^{t-1}(\Delta_\Gamma^k + \Delta_\Lambda^k)\right\} \cdot \exp\left\{2\eta \cdot (\Delta_\Gamma^t + \Delta_\Lambda^t)\right\}(1 + \delta)^{-1}}{1 + \exp\left\{2\eta \cdot \sum_{k=0}^{t-1}(\Delta_\Gamma^k + \Delta_\Lambda^k)\right\} \cdot \exp\left\{2\eta \cdot (\Delta_\Gamma^t + \Delta_\Lambda^t)\right\}(1 + \delta)^{-1}} \triangleq \spadesuit^{t+1},$$

$$\Delta_\Gamma^{t+1} - \Delta_\Lambda^{t+1} \leq$$

$$\frac{2}{m} \cdot \frac{\overline{C}_{-1+1} - \overline{C}_{+1-1} \cdot \exp\left\{2\eta \cdot \sum_{k=0}^{t-1}(\Delta_\Gamma^k - \Delta_\Lambda^k)\right\} \cdot \exp\left\{2\eta \cdot (\Delta_\Gamma^t - \Delta_\Lambda^t)\right\}(1 + \delta)^{-1}}{1 + \exp\left\{2\eta \cdot \sum_{k=0}^{t-1}(\Delta_\Gamma^k - \Delta_\Lambda^k)\right\} \cdot \exp\left\{2\eta \cdot (\Delta_\Gamma^t - \Delta_\Lambda^t)\right\}(1 + \delta)^{-1}} \triangleq \clubsuit^{t+1}.$$

Requiring that $\eta > \max\left\{\frac{1}{\Delta_\Gamma^t + \Delta_\Lambda^t} \log(1 + \delta), \frac{1}{\Delta_\Gamma^t - \Delta_\Lambda^t} \log(1 + \delta)\right\}, \forall t \in \{0, \ldots, T\} \Rightarrow \eta > \epsilon_\Delta^{-1} \log(1 + \delta)$, we have

$$\spadesuit^{t+1} < \frac{2}{m} \cdot \frac{\overline{C}_{+1+1} - \overline{C}_{-1-1} \cdot \exp\left\{2\eta \cdot \sum_{k=0}^{t-1}(\Delta_\Gamma^k + \Delta_\Lambda^k)\right\} \cdot \exp\left\{2\eta \cdot (\Delta_\Gamma^t + \Delta_\Lambda^t)\right\}(1 + \delta)^{-1}}{1 + \exp\left\{2\eta \cdot \sum_{k=0}^{t-1}(\Delta_\Gamma^k + \Delta_\Lambda^k)\right\} \cdot (1 + \delta)}$$

$$< \Delta_\Gamma^t + \Delta_\Lambda^t,$$

$$\clubsuit^{t+1} < \frac{2}{m} \cdot \frac{\overline{C}_{-1+1} - \overline{C}_{+1-1} \cdot \exp\left\{2\eta \cdot \sum_{k=0}^{t-1}(\Delta_\Gamma^k - \Delta_\Lambda^k)\right\} \cdot \exp\left\{2\eta \cdot (\Delta_\Gamma^t - \Delta_\Lambda^t)\right\}(1 + \delta)^{-1}}{1 + \exp\left\{2\eta \cdot \sum_{k=0}^{t-1}(\Delta_\Gamma^k - \Delta_\Lambda^k)\right\} \cdot (1 + \delta)}$$

$$< \Delta_\Gamma^t - \Delta_\Lambda^t,$$

where the last inequalities use the lower bounds (14) and (15).

Based on the above discussion and (19), we can now clarify the requirements of $\eta$ for the sequences $\{\Delta_\Gamma^t + \Delta_\Lambda^t\}$ and $\{\Delta_\Gamma^t - \Delta_\Lambda^t\}$ to be positive and monotonically decreasing:

$$\epsilon_\Delta^{-1} \log(1 + \delta) < \eta < \min \left\{ \frac{m(2 + \delta)}{4T(\overline{C}_{+1+1}(1 + \delta) - \overline{C}_{-1-1})} \log \frac{\overline{C}_{+1+1}}{\overline{C}_{-1-1}(1 + \delta)}, \right.$$

$$\left. \frac{m(2 + \delta)}{4T(\overline{C}_{-1+1}(1 + \delta) - \overline{C}_{+1-1})} \log \frac{\overline{C}_{-1+1}}{\overline{C}_{+1-1}(1 + \delta)} \right\},$$

(20)

which uses the upper bounds (12) and (13) at iteration 0. The constants $\overline{C}_{+1+1}, \overline{C}_{+1-1}, \overline{C}_{-1+1}$ and $\overline{C}_{-1-1}$ can be substituted using the concentration bounds at (18) to generate an upper bound for $\eta$ that only involves $\alpha, \beta_1, \beta_2, m, \delta, T, \epsilon_C$. Here we omit the precise upper bound for clarity. Note that

the left hand side of (20) approaches 0 if $\delta \to 0$, which means that there exists a constant choice of $\eta$ in (20) if $\underline{n}$ is sufficiently large in Lemma D.2 and D.3.

To conclude, in view of (11), the convergences of the sequences $\{\Delta_\Gamma^t + \Delta_\Lambda^t\}$ and $\{\Delta_\Gamma^t - \Delta_\Lambda^t\}$ imply that $\Lambda_{j,r}^t$ and $\Gamma_{j,r}^t$ are converging, and the positive sequence $\{\Delta_\Gamma^t - \Delta_\Lambda^t\}$ indicates that the increment of the spurious feature $\Gamma_{j,r}^{t+1} - \Gamma_{j,r}^t$ is larger than that of the invariant feature $\Lambda_{j,r}^{t+1} - \Lambda_{j,r}^t$ at any iteration $t \in \{0, \dots, T-1\}$. $\qquad\square$

### D.2.1 Proof of Lemma D.2

First, we recall some concentration inequalities for sub-Gaussian random variables. Since $\boldsymbol{\xi}_i^e \sim \mathcal{N}(0, \sigma_p^2 \cdot (\mathbf{I}_d - \mathbf{v}_1 \mathbf{v}_1^\top - \mathbf{v}_2 \mathbf{v}_2^\top))$, for $(i', e') \neq (i, e)$, using Bernstein's inequality for sub-exponential random variables, we have for sufficiently small $a \geq 0$,

$$\Pr\left\{|\langle \boldsymbol{\xi}_i^e, \boldsymbol{\xi}_{i'}^{e'} \rangle| \geq a\right\} \leq 2\exp\left\{-\frac{a^2}{4\sigma_p^4(d-2)}\right\},$$

$$\Pr\left\{\left|\|\boldsymbol{\xi}_i^e\|_2^2 - \sigma_p^2(d-2)\right| \geq a\right\} \leq 2\exp\left\{-\frac{a^2}{512\sigma_p^4(d-2)}\right\}.$$

Moreover, for $\xi_r \sim \mathcal{N}(0, \sigma_0^2)$ (indicating the initial weights $\mathbf{w}_{j,r}^0$), the standard Gaussian tail gives

$$\Pr\left\{\left|\frac{1}{m}\sum_{r=1}^m \xi_r\right| \geq a\right\} \leq 2\exp\left\{-\frac{ma^2}{2\sigma_0^2}\right\}.$$

Denote $n \triangleq \sum_{e \in \mathcal{E}_{tr}} n_e, \underline{n} \triangleq \min_{e \in \mathcal{E}_{tr}} n_e$, by properly choosing $a$ for each tail bound and applying a union bound, we can conclude that for $\rho > 0$, with probability at least $1 - \rho$, it holds that $\forall i, e, i', e', r$,

$$|\langle \boldsymbol{\xi}_i^e, \boldsymbol{\xi}_{i'}^{e'} \rangle| \leq 2\sigma_p^2 \sqrt{(d-2)\log\frac{8n^2}{\rho}}, \quad \|\boldsymbol{\xi}_i^e\|_2^2 \leq \sigma_p^2(d-2) + 16\sigma_p^2\sqrt{2(d-2)\log\frac{8n}{\rho}},$$

$$\left|\frac{1}{m}\sum_{r=1}^m \xi_r\right| \leq \sigma_0\sqrt{\frac{2}{m}\log\frac{32m}{\rho}}, \qquad |\langle \boldsymbol{\xi}_r, \boldsymbol{\xi}_{i'}^{e'} \rangle| \leq 2\sigma_p\sigma_0\sqrt{(d-2)\log\frac{16nm}{\rho}}.$$

We start with bound the growth of $\Xi_{j,r,i}^{t,e}$. By bounding the update rule (10), with probability at least $1 - \rho$, we have

$$\left|\Xi_{j,r,i'}^{t+1,e'}\right| \leq \left|\Xi_{j,r,i'}^{t,e'}\right| + \frac{\eta}{m}\sum_{e \in \mathcal{E}_{tr}} \frac{1}{n_e}\sum_{i=1}^{n_e} \frac{1}{1 + \exp\{y_i^e \hat{y}_i^e\}} \cdot |\langle \boldsymbol{\xi}_i^e, \boldsymbol{\xi}_{i'}^{e'} \rangle|$$

$$\leq \left|\Xi_{j,r,i'}^{t,e'}\right| + \frac{\eta}{m}\sum_{e \in \mathcal{E}_{tr}} \frac{1}{n_e}\sum_{i=1}^{n_e} |\langle \boldsymbol{\xi}_i^e, \boldsymbol{\xi}_{i'}^{e'} \rangle|$$

$$= \left|\Xi_{j,r,i'}^{0,e'}\right| + (t+1) \cdot \frac{\eta}{m}\sum_{e \in \mathcal{E}_{tr}} \frac{1}{n_e}\sum_{i=1}^{n_e} |\langle \boldsymbol{\xi}_i^e, \boldsymbol{\xi}_{i'}^{e'} \rangle|$$

$$= |\langle \boldsymbol{\xi}_r, \boldsymbol{\xi}_{i'}^{e'} \rangle| + (t+1) \cdot \left(\frac{\eta}{mn_{e'}}\|\boldsymbol{\xi}_{i'}^{e'}\|_2^2 + \sum_{(i,e) \neq (i',e')} \frac{\eta}{mn_e}|\langle \boldsymbol{\xi}_i^e, \boldsymbol{\xi}_{i'}^{e'} \rangle|\right)$$

$$\leq 2\sigma_p\sigma_0\sqrt{(d-2)\log\frac{16nm}{\rho}}$$

$$+ \frac{T\eta\sigma_p^2}{m\underline{n}}\left((d-2) + 16\sqrt{2(d-2)\log\frac{8n}{\rho}} + 2n\sqrt{(d-2)\log\frac{8n^2}{\rho}}\right).$$

Then, we can bound $|\mathbb{Q}_i^e|$ as

$$|\mathbb{Q}_i^e| \leq 2 \cdot \left| \frac{1}{m} \sum_{r=1}^m \xi_r \right| + 2 \cdot \left| \frac{1}{m} \sum_{r=1}^m \xi_r \right| + \frac{2}{m} \sum_{r=1}^m |\Xi_{j,r,i}^{t,e}|$$

$$\leq 4\sigma_0 \sqrt{\frac{2}{m} \log \frac{32m}{\rho}} + 4\sigma_p \sigma_0 \sqrt{(d-2) \log \frac{16nm}{\rho}}$$

$$+ \frac{2T\eta\sigma_p^2}{m\underline{n}} \left( (d-2) + 16\sqrt{2(d-2)\log \frac{8n}{\rho}} + 2n\sqrt{(d-2)\log \frac{8n^2}{\rho}} \right).$$

Thus, with sufficient small $\sigma_0, \sigma_p$, i.e.,

$$\sigma_0^2 = O\left( \underline{n}^{-2} \log^{-1}(m/\rho) \right),$$

$$\sigma_p^2 = O\left( \min\left\{ d^{-1/2} \log^{-1/2}(nm/\rho), T^{-1}\eta^{-1}m \left( d + n\sqrt{d\log(n^2/\rho)} \right)^{-1} \right\} \right),$$

we ensured that $|\mathbb{Q}_i^e| = O(\underline{n}^{-1})$.

### D.2.2 Proof of Lemma D.3

For $e \in \mathcal{E}_{tr}$, using Hoeffding's inequality, it holds that

$$\Pr\left\{ \left| \frac{1}{n_e} \sum_{i=1}^{n_e} \mathbf{1}_{\{\text{Rad}(\alpha)_i=+1, \text{Rad}(\beta_e)_i=+1\}} - (1-\alpha)(1-\beta_e) \right| \geq a \right\} \leq 2\exp\{-2a^2 n_e\}.$$

Considering two environments, using a union bound, we can conclude that

$$\Pr\left\{ \left| \overline{C}_{+1+1} - (1-\alpha)(2-\beta_1-\beta_2) \right| \leq a \right\} \geq 1 - 4\exp\left\{ -\frac{a^2 n}{2} \right\}.$$

Thus, for $\rho > 0$, with probability at least $1 - \frac{\rho}{4}$, we can conclude that

$$\left| \overline{C}_{+1+1} - (1-\alpha)(2-\beta_1-\beta_2) \right| \leq \sqrt{\frac{2\log(16/\rho)}{\underline{n}}}.$$

Using the above arguments for other constants $\overline{C}_{+1-1}, \overline{C}_{-1+1}$ and $\overline{C}_{-1-1}$, and applying a union bound, we obtain the desired results.

### D.2.3 ERM Feature Learning with Non-Linear Activation Functions

It was numerically observed that in the early stage of (stochastic) GD training, the learning dynamics of neural networks can be mimicked by training a simple linear model [35]. Hu et al. [30] rigorously proved this phenomenon for training two-layer neural network with $\ell_2$ loss function in the Neural Tangent Kernel (NTK) region. We briefly summarize their results here: For a two-layer fully-connected neural network (with fixed second layer $\{v_r\}$):

$$f_{FC}(\mathbf{W}, \mathbf{x}) \triangleq \frac{1}{\sqrt{m}} \sum_{r=1}^m v_r \psi\left( \mathbf{w}_r^\top \mathbf{x}/\sqrt{d} \right), \tag{21}$$

considering the $\ell_2$ training loss $\ell_2(\hat{y}, y) \triangleq \frac{1}{2}(\hat{y} - y)^2$ and the ERM objective $L_{\text{ERM}}(\mathbf{W}) = \frac{1}{n} \sum_{i=1}^n \ell_2\left( f_{FC}(\mathbf{W}, \mathbf{x}_i), y_i \right)$, when using GD: $\mathbf{W}^{t+1} = \mathbf{W}^t - \eta \cdot \nabla L_{\text{ERM}}(\mathbf{W}^t)$ to minimize the ERM objective, the following holds.

**Theorem D.4** (Theorem 3.2 of [30]). *Let $\alpha_{nl} \in (0, \frac{1}{4})$ be a fixed constant, and $\psi(\cdot)$ be a smooth (with bounded first and second derivatives) or piece-wise linear activation function. Suppose that $n$ and $m$ satisfy $n = \Omega(d^{1+\alpha_{nl}})$ and $m = \Omega(d^{1+\alpha_{nl}})$. Suppose that $\eta \ll d$. Then there exists a universal constant $c > 0$ such that with high probability, for all $t = O(\frac{d}{\eta} \log d)$ simultaneously, the learned neural network $f_{FC}^t$ and the linear model $f_{lin}^t$ (defined below) at iteration $t$ are close on average on the training data:*

$$\frac{1}{n} \sum_{i=1}^n \left( f_{FC}^t(\mathbf{x}_i) - f_{lin}^t(\mathbf{x}_i) \right)^2 = O(d^{-\Omega(\alpha_{nl})}).$$

The linear model $f_{lin}(\boldsymbol{\beta}, \mathbf{x}) = \boldsymbol{\beta}^\top \boldsymbol{R}(\mathbf{x})$ is a linear function of the transformed data $\boldsymbol{R}(\mathbf{x}) = \frac{1}{\sqrt{d}} \begin{bmatrix} \zeta \mathbf{x} \\ \nu \end{bmatrix}$, where $\zeta$ and $\nu$ are constants related to $\psi'$ and the dataset distribution (see (5) in [30] for formal definitions).

We show that we can relate our data model to the dataset setup in [30], and thus by analyzing the feature learning terms for the linear model $f_{lin}(\boldsymbol{\beta}, \mathbf{x})$ similar to the analysis[4] in Appendix D.2, we obtain similar results as in Theorem D.1 in the early stage of GD training, but with an error of $O(d^{-\Omega(\alpha_{nl})})$.

Recall that our CNN model is $f(\mathbf{W}, \mathbf{x}) = F_{+1}(\mathbf{W}_{+1}, \mathbf{x}) - F_{-1}(\mathbf{W}_{-1}, \mathbf{x})$, where $F_{+1}(\mathbf{W}_{+1}, \mathbf{x})$ and $F_{-1}(\mathbf{W}_{-1}, \mathbf{x})$ are defined as follows:

$$F_j(\mathbf{W}_j, \mathbf{x}) = \frac{1}{m} \sum_{r=1}^{m} \left[ \psi(\mathbf{w}_{j,r}^\top \mathbf{x}_1) + \psi(\mathbf{w}_{j,r}^\top \mathbf{x}_2) \right], j \in \{\pm 1\}.$$

We can cast this CNN model into an instance of the two-layer fully connected neural network defined at (21) by specifying the values of $\{v_r = \pm \frac{1}{\sqrt{m}}\}$ and transforming the dataset as $\left\{ \sqrt{d} \begin{bmatrix} y \cdot \mathrm{Rad}(\alpha) \cdot \mathbf{v}_1 + y \cdot \mathrm{Rad}(\beta) \cdot \mathbf{v}_2 \\ 0 \end{bmatrix}, \sqrt{d} \boldsymbol{\xi} \right\}$. Then by tuning the norms of $\mathbf{v}_1, \mathbf{v}_2$ and $\boldsymbol{\xi}$, we obtain a dataset that satisfies the input assumptions in [30]. Note that this cast drops the shared variable of our CNN model and thus might lead to a slightly different training dynamic. To fix such gap, we can leverage Proposition 6.4.1 in [29] for the early stage behavior of training a CNN model.

Based on the above ideas, to formalize the convergence results of the feature learning terms in the non-linear case, it remains to re-derive the analysis in Appendix D.2 based on $\ell_2$ loss function, which follows a similar line of proofs and has a simpler dynamic.

---

[4]Note that when $\psi(x) = x$, our CNN model can be viewed as a linear model with re-parameterized weight matrices. Thus, the discussion in Appendix D.2 can be viewed as studying the feature learning terms for a linear model with logistic loss function.

## D.3 Proof for Theorem 4.2

**Theorem D.5** (Restatement of Theorem 4.2). *Consider training a CNN model with the same data as in Theorem 4.1, define*

$$\mathbf{c}(t) \triangleq \left[ C_{IRMv1}^1(\mathbf{W}, t), C_{IRMv1}^2(\mathbf{W}, t), \cdots, C_{IRMv1}^{|\mathcal{E}_{tr}|}(\mathbf{W}, t) \right],$$

*and $\lambda_0 = \lambda_{\min}(\mathbf{H}^\infty)$, where we define*

$$\mathbf{H}_{e,e'}^\infty \triangleq \frac{1}{2mn_e n_{e'}} \sum_{i=1}^{n_e} \psi'(\langle \mathbf{w}_{j,r}(0), \mathbf{x}_{1,i}^e \rangle) \mathbf{x}_{1,i}^{e\top} \sum_{i'=1}^{n_{e'}} \psi'(\langle \mathbf{w}_{j,r}(0), \mathbf{x}_{1,i'}^{e'} \rangle) \mathbf{x}_{1,i'}^{e'}.$$

*Suppose that activation function is smooth, $\psi'(0) \le \beta$, $|\psi'(x) - \psi'(x')| < \beta|x - x'|$ and Lipschitz $|\psi(0)| < L$, $|\psi(x) - \psi(x')| < L|x - x'|$. Assume that dimension $d = \Omega(\log(m/\delta))$, network width $m = \Omega(1/\delta)$, regularization factor $\lambda \ge 1/(\sigma_0\sqrt{|\mathcal{E}_{tr}|}^3)$, noise variance $\sigma_p = O(d^{-2})$, weight initial scale $\sigma_0 = O(\frac{|\mathcal{E}_{tr}|^{7/2}\beta^3 L}{d^{1/2}m^2\lambda_0^2\log(1/\epsilon)})$, then with probability at least $1 - \delta$, after training time $T = \Omega\left(\frac{\log(1/\epsilon)}{\eta\lambda\lambda_0}\right)$, we have:*

$$\|\mathbf{c}(T)\|_2 \le \epsilon, \quad \gamma_{j,r}^{inv}(T) = o(1), \quad \gamma_{j,r}^{spu}(T) = o(1).$$

Before proving Theorem D.5, we first provide some useful lemmas as follows:

**Lemma D.6** ([12]). *Suppose that $\delta > 0$ and $d = \Omega(\log(4n/\delta))$. Then with probability at least $1 - \delta$,*

$$\sigma_p^2 d/2 \le \|\boldsymbol{\xi}_i\|_2^2 \le 3\sigma_p^2 d/2$$

*for all $i, i' \in [n]$.*

**Lemma D.7** ([12]). *Suppose that $d \ge \Omega(\log(mn/\delta))$, $m = \Omega(\log(1/\delta))$. Then with probability at least $1 - \delta$,*

$$|\langle \mathbf{w}_{j,r}^{(0)}, \mathbf{v}_1 \rangle| \le \sqrt{2\log(8m/\delta)} \cdot \sigma_0 \|\mathbf{v}_1\|_2,$$
$$|\langle \mathbf{w}_{j,r}^{(0)}, \mathbf{v}_2 \rangle| \le \sqrt{2\log(8m/\delta)} \cdot \sigma_0 \|\mathbf{v}_2\|_2,$$
$$|\langle \mathbf{w}_{j,r}^{(0)}, \boldsymbol{\xi}_i \rangle| \le 2\sqrt{\log(8mn/\delta)} \cdot \sigma_0 \sigma_p \sqrt{d}$$

*for all $r \in [m]$, $j \in \{\pm 1\}$ and $i \in [n]$.*

**Lemma D.8.** *Suppose that $\delta > 0$ and $d = \Omega(\log(4m/\delta))$. Then with probability at least $1 - \delta$, for all $r \in [m]$ and $j \in \{-1, 1\}$, we have*

$$\sigma_0^2 d/2 \le \|\mathbf{w}_{j,r}(0)\|_2^2 \le 3\sigma_0^2 d/2.$$

*Proof of Lemma D.8.* By Bernstein's inequality, with probability at least $1 - \delta/(2m)$ we have

$$\left|\|\mathbf{w}_{j,r}(0)\|_2^2 - \sigma_0^2 d\right| = O(\sigma_0^2 \cdot \sqrt{d\log(4m/\delta)}).$$

Therefore, as long as $d = \Omega(\log(4m/\delta))$, we have

$$\sigma_0^2 d/2 \le \|\mathbf{w}_{j,r}(0)\|_2^2 \le 3\sigma_0^2 d/2.$$

$\square$

*Proof of Theorem D.5.* The proof is by induction method. First we show the gradient flow of weights by IRMv1 objective function (5):

$$\frac{d\mathbf{w}_{j,r}(t)}{dt} = -\eta \cdot \nabla_{\mathbf{w}_{j,r}} L_{\mathrm{IRMv1}}(\mathbf{W}(t))$$

$$= -\frac{\eta}{nm} \sum_{e \in \mathcal{E}_{\mathrm{tr}}} \sum_{i=1}^{n_e} \ell_i'(t) \psi'(\langle \mathbf{w}_{j,r}(t), y_i^e \mathbf{v}_i^e \rangle) \cdot j\mathbf{v}_i^e - \frac{\eta}{nm} \sum_{e \in \mathcal{E}_{\mathrm{tr}}} \sum_{i=1}^{n_e} \ell_i'(t) \psi'(\langle \mathbf{w}_{j,r}(t), \boldsymbol{\xi}_i \rangle) \cdot jy_i^e \boldsymbol{\xi}_i$$

$$- \frac{2\eta\lambda}{nm} \sum_{e \in \mathcal{E}_{\mathrm{tr}}} C_{\mathrm{IRMv1}}^e \sum_{i=1}^{n_e} \ell_i'' \hat{y}_i^e \psi'(\langle \mathbf{w}_{j,r}(t), y_i^e \mathbf{v}_i^e \rangle) jy_i^e \mathbf{v}_i^e - \frac{2\eta\lambda}{nm} \sum_{e \in \mathcal{E}_{\mathrm{tr}}} C_{\mathrm{IRMv1}}^e \sum_{i=1}^{n_e} \ell_i'' \hat{y}_i^e \psi'(\langle \mathbf{w}_{j,r}(t), \boldsymbol{\xi}_i \rangle) j\boldsymbol{\xi}_i$$

$$- \frac{2\eta\lambda}{nm} \sum_{e \in \mathcal{E}_{\mathrm{tr}}} C_{\mathrm{IRMv1}}^e \sum_{i=1}^{n_e} \ell_i'(t) \psi'(\langle \mathbf{w}_{j,r}(t), y_i^e \mathbf{v}_i^e \rangle) j\mathbf{v}_i^e - \frac{2\eta\lambda}{nm} \sum_{e \in \mathcal{E}_{\mathrm{tr}}} C_{\mathrm{IRMv1}}^e \sum_{i=1}^{n_e} \ell_i'(t) \psi'(\langle \mathbf{w}_{j,r}(t), \boldsymbol{\xi}_i \rangle) jy_i^e \boldsymbol{\xi}_i$$

$$= -\frac{\eta}{nm} \sum_{e \in \mathcal{E}_{\mathrm{tr}}} (1 + 2\lambda C_{\mathrm{IRMv1}}^e(t)) \sum_{i=1}^{n_e} \ell_i'(t) \psi'(\langle \mathbf{w}_{j,r}(t), y_i^e \mathbf{v}_i^e \rangle) \cdot j\mathbf{v}_i^e$$

$$- \frac{\eta}{nm} \sum_{e \in \mathcal{E}_{\mathrm{tr}}} (1 + 2\lambda C_{\mathrm{IRMv1}}^e(t)) \sum_{i=1}^{n_e} \ell_i'(t) \psi'(\langle \mathbf{w}_{j,r}(t), \boldsymbol{\xi}_i \rangle) \cdot jy_i^e \boldsymbol{\xi}_i$$

$$- \frac{2\eta\lambda}{nm} \sum_{e \in \mathcal{E}_{\mathrm{tr}}} C_{\mathrm{IRMv1}}^e \sum_{i=1}^{n_e} \ell_i'' \hat{y}_i^e \psi'(\langle \mathbf{w}_{j,r}(t), y_i^e \mathbf{v}_i^e \rangle) jy_i^e \mathbf{v}_i^e - \frac{2\eta\lambda}{nm} \sum_{e \in \mathcal{E}_{\mathrm{tr}}} C_{\mathrm{IRMv1}}^e \sum_{i=1}^{n_e} \ell_i'' \hat{y}_i^e \psi'(\langle \mathbf{w}_{j,r}(t), \boldsymbol{\xi}_i \rangle) j\boldsymbol{\xi}_i,$$

where $C_{\mathrm{IRMv1}}^e = \frac{1}{n_e} \sum_{i=1}^{n_e} \ell_i'^e \hat{y}_i^e y_i^e$ and $\mathbf{v}_i^e = \mathrm{Rad}(\alpha)_i \cdot \mathbf{v}_1 + \mathrm{Rad}(\beta_e)_i \cdot \mathbf{v}_2$. Note that $\ell''$ has the opposite sign to $\ell'$.

Then we look at the dynamics of $C_{\mathrm{IRMv1}}^e(t)$ according to the gradient flow update rule:

$$\frac{dC_{\mathrm{IRMv1}}^e(\mathbf{W}, t)}{dt} = \sum_{j=\pm 1} \sum_{r=1}^{m} \left\langle \frac{\partial C_{\mathrm{IRMv1}}^e(\mathbf{W}, t)}{\partial \mathbf{w}_{j,r}(t)}, \frac{d\mathbf{w}_{j,r}(t)}{dt} \right\rangle$$

$$= \sum_{e'} 2\lambda C_{\mathrm{IRMv1}}^{e'}(\mathbf{W}, t) \sum_{j} \sum_{r=1}^{m} \left\langle \frac{\partial C_{\mathrm{IRMv1}}^e(\mathbf{W}, t)}{\partial \mathbf{w}_{j,r}(t)}, \frac{\partial C_{\mathrm{IRMv1}}^{e'}(\mathbf{W}, t)}{\partial \mathbf{w}_{j,r}(t)} \right\rangle$$

$$+ \sum_{j=\pm 1} \sum_{r=1}^{m} \left\langle \frac{\partial C_{\mathrm{IRMv1}}^e(\mathbf{W}, t)}{\partial \mathbf{w}_{j,r}(t)}, \frac{\partial L(\mathbf{W}, t)}{\partial \mathbf{w}_{j,r}(t)} \right\rangle$$

$$= 2\lambda \sum_{e'} C_{\mathrm{IRMv1}}^{e'}(\mathbf{W}, t) \cdot \mathbf{H}_{e,e'}(t) + \mathbf{g}_e(t),$$

where we define $\mathbf{H}_{e,e'}(t) = \sum_{j} \sum_{r=1}^{m} \left\langle \frac{\partial C_{\mathrm{IRMv1}}^e(\mathbf{W}, t)}{\partial \mathbf{w}_{j,r}(t)}, \frac{\partial C_{\mathrm{IRMv1}}^{e'}(\mathbf{W}, t)}{\partial \mathbf{w}_{j,r}(t)} \right\rangle$ and $\mathbf{g}_e(t) = \sum_{j=\pm 1} \sum_{r=1}^{m} \left\langle \frac{\partial C_{\mathrm{IRMv1}}^e(\mathbf{W}, t)}{\partial \mathbf{w}_{j,r}(t)}, \frac{\partial L(\mathbf{W}, t)}{\partial \mathbf{w}_{j,r}(t)} \right\rangle$. Thus $\mathbf{H}(t)$ is an $|\mathcal{E}_{\mathrm{tr}}| \times |\mathcal{E}_{\mathrm{tr}}|$ matrix. We can write the dynamics of $\mathbf{c}(t) = \left[ C_{\mathrm{IRMv1}}^1(\mathbf{W}, t), C_{\mathrm{IRMv1}}^2(\mathbf{W}, t), \cdots, C_{\mathrm{IRMv1}}^{|\mathcal{E}_{\mathrm{tr}}|}(\mathbf{W}, t) \right]$ in a compact way:

$$\frac{d\mathbf{c}(t)}{dt} = 2\lambda \cdot \mathbf{H}(t)\mathbf{c}(t) + \mathbf{g}(t). \tag{22}$$

Our next step is to show $\mathbf{H}(t)$ is stable during training. To proceed with the analysis, we write down the expression for $\frac{\partial C_{\mathrm{IRMv1}}^e(\mathbf{W}, t)}{\partial \mathbf{w}_{j,r}(t)} \in \mathbb{R}^d$:

$$\frac{\partial C_{\mathrm{IRMv1}}^e(\mathbf{W}(t))}{\partial \mathbf{w}_{j,r}(t)} = \frac{1}{n_e m} \sum_{i=1}^{n_e} \ell_i'(t) \psi'(\langle \mathbf{w}_{j,r}(t), y_i^e \mathbf{v}_i^e \rangle) \cdot j\mathbf{v}_i^e + \frac{1}{n_e m} \sum_{i=1}^{n_e} \ell_i'(t) \psi'(\langle \mathbf{w}_{j,r}(t), \boldsymbol{\xi}_i \rangle) \cdot jy_i^e \boldsymbol{\xi}_i$$

$$+ \frac{1}{n_e m} \sum_{i=1}^{n_e} \ell_i'' \hat{y}_i^e \psi'(\langle \mathbf{w}_{j,r}(t), y_i^e \mathbf{v}_i^e \rangle) \cdot jy_i^e \mathbf{v}_i^e + \frac{1}{n_e m} \sum_{i=1}^{n_e} \ell_i'' \hat{y}_i^e \psi'(\langle \mathbf{w}_{j,r}(t), \boldsymbol{\xi}_i \rangle) \cdot j\boldsymbol{\xi}_i.$$

When we consider non-linear activation function $\psi(x)$, the entry of matrix $\mathbf{H}(t)$ can be computed as follows:

$$\mathbf{H}_{e,e'}(t) = \sum_j \sum_{r=1}^m \left\langle \frac{\partial C_{\text{IRMv1}}^e(\mathbf{W},t)}{\partial \mathbf{w}_{j,r}(t)}, \frac{\partial C_{\text{IRMv1}}^{e'}(\mathbf{W},t)}{\partial \mathbf{w}_{j,r}(t)} \right\rangle$$

$$= \sum_j \sum_{r=1}^m \left(\frac{1}{n_e m}\right)\left(\frac{1}{n_{e'} m}\right)\left[ \sum_{i=1}^{n_e} \ell_i'(t)\psi' j \mathbf{v}_i^{e\top} \sum_{i'=1}^{n_{e'}} \ell_{i'}'(t)\psi' j \mathbf{v}_{i'}^{e'} + \sum_{i=1}^{n_e} \psi'\ell_i''(t)\hat{y}_i^e(t) j y_i^e \mathbf{v}_i^{e\top} \sum_{i'=1}^{n_{e'}} \psi' \ell_{i'}''(t)\hat{y}_{i'}^{e'}(t) j y_{i'}^{e'} \mathbf{v}_{i'}^{e'} \right]$$

$$+ \sum_j \sum_{r=1}^m \left(\frac{1}{n_e m}\right)\left(\frac{1}{n_{e'} m}\right)\left[ \sum_{i=1}^{n_e} \psi' \ell_i''(t)\hat{y}_i^e(t) j y_i^e \mathbf{v}_i^{e\top} \sum_{i'=1}^{n_{e'}} \psi' \ell_{i'}'(t) j \mathbf{v}_{i'}^{e'} + \sum_{i=1}^{n_e} \psi' \ell_i'(t) j \mathbf{v}_i^{e\top} \sum_{i'=1}^{n_{e'}} \psi' \ell_{i'}''(t)\hat{y}_{i'}^{e'}(t) j \mathbf{v}_{i'}^{e'} \right]$$

$$+ \sum_j \sum_{r=1}^m \left(\frac{1}{n_e m}\right)\left(\frac{1}{n_{e'} m}\right)\left[ \sum_{i=1}^{n_e} \psi' \ell_i'(t) j y_i^e \boldsymbol{\xi}_i^{e\top} \sum_{i'=1}^{n_{e'}} \psi' \ell_{i'}'(t) j y_{i'}^{e'} \boldsymbol{\xi}_{i'}^{e'} + \sum_{i=1}^{n_e} \psi' \ell_i''(t)\hat{y}_i^e(t) j \boldsymbol{\xi}_i^{e\top} \sum_{i'=1}^{n_{e'}} \psi' \ell_{i'}''(t)\hat{y}_{i'}^{e'}(t) j \boldsymbol{\xi}_{i'}^{e'} \right]$$

$$+ \sum_j \sum_{r=1}^m \left(\frac{1}{n_e m}\right)\left(\frac{1}{n_{e'} m}\right)\left[ \sum_{i=1}^{n_e} \psi' \ell_i''(t)\hat{y}_i^e j \boldsymbol{\xi}_i^{e\top} \sum_{i'=1}^{n_{e'}} \psi' \ell_{i'}'(t) j y_{i'}^{e'} \boldsymbol{\xi}_{i'}^{e'} + \sum_{i=1}^{n_e} \psi' y_i^e \ell_i'(t) j \boldsymbol{\xi}_i^{e\top} \sum_{i'=1}^{n_{e'}} \psi' \ell_{i'}''(t) j \boldsymbol{\xi}_{i'}^e \hat{y}_{i'}^{e'}(t) \right]$$

$$\triangleq \mathbf{H}_{e,e'}^1(t) + \mathbf{H}_{e,e'}^2(t) + \mathbf{H}_{e,e'}^3(t) + \mathbf{H}_{e,e'}^4(t) + \mathbf{H}_{e,e'}^5(t) + \mathbf{H}_{e,e'}^6(t) + \mathbf{H}_{e,e'}^7(t) + \mathbf{H}_{e,e'}^8(t).$$

The matrix $\mathbf{H}$ is composed of eight elements. In addition, we define

$$\mathbf{H}_{e,e'}^{1,\infty} = \sum_j \sum_{r=1}^m \left(\frac{1}{n_e m}\right)\left(\frac{1}{n_{e'} m}\right)\left[ \sum_{i=1}^{n_e} -\frac{1}{2}\psi'(\langle \mathbf{w}_{j,r}(0), \mathbf{v}_i^e \rangle) j \mathbf{v}_i^{e\top} \sum_{i'=1}^{n_{e'}} -\frac{1}{2}\psi'(\langle \mathbf{w}_{j,r}(0), \mathbf{v}_{i'}^{e'} \rangle) j \mathbf{v}_{i'}^{e'} \right]$$

$$= \frac{1}{2 m n_e n_{e'}} \sum_{i=1}^{n_e} \psi'(\langle \mathbf{w}_{j,r}(0), \mathbf{v}_i^e \rangle) \mathbf{v}_i^{e\top} \sum_{i'=1}^{n_{e'}} \psi'(\langle \mathbf{w}_{j,r}(0), \mathbf{v}_{i'}^{e'} \rangle) \mathbf{v}_{i'}^{e'}.$$

Then we can show that:

$$\left| \mathbf{H}_{e,e'}^1(t) - \mathbf{H}_{e,e'}^{1,\infty} \right|$$

$$= \frac{2}{m n_e n_{e'}} \left| \sum_{i=1}^{n_e} \psi'(t)\ell_i'(t)\mathbf{v}_i^{e\top} \sum_{i'=1}^{n_{e'}} \psi'(t)\ell_{i'}'(t)\mathbf{v}_{i'}^{e'} - \sum_{i=1}^{n_e} \frac{1}{2}\psi'(0)\mathbf{v}_i^{e\top} \sum_{i'=1}^{n_{e'}} \frac{1}{2}\psi'(0)\mathbf{v}_{i'}^{e'} \right|$$

$$= \frac{2}{m n_e n_{e'}} \left| \sum_{i=1}^{n_e} \psi'(t)\ell_i'(t)\mathbf{v}_i^{e\top} \sum_{i'=1}^{n_{e'}} \psi'(t)\ell_{i'}'(t)\mathbf{v}_{i'}^{e'} - \sum_{i=1}^{n_e} \psi'(0)\ell_i'(t)\mathbf{v}_i^{e\top} \sum_{i'=1}^{n_{e'}} \psi'(0)\ell_{i'}'(t)\mathbf{v}_{i'}^{e'} \right.$$

$$\left. + \sum_{i=1}^{n_e} \psi'(0)\ell_i'(t)\mathbf{v}_i^{e\top} \sum_{i'=1}^{n_{e'}} \psi'(0)\ell_{i'}'(t)\mathbf{v}_{i'}^{e'} - \sum_{i=1}^{n_e} \frac{1}{2}\psi'(0)\mathbf{v}_i^{e\top} \sum_{i'=1}^{n_{e'}} \frac{1}{2}\psi'(0)\mathbf{v}_{i'}^{e'} \right|$$

$$\leq \frac{2}{m n_e n_{e'}} \left| \sum_{i=1}^{n_e} \psi'(t)\ell_i'(t)\mathbf{v}_i^{e\top} \sum_{i'=1}^{n_{e'}} \psi'(t)\ell_{i'}'(t)\mathbf{v}_{i'}^{e'} - \sum_{i=1}^{n_e} \psi'(0)\ell_i'(t)\mathbf{v}_i^{e\top} \sum_{i'=1}^{n_{e'}} \psi'(t)\ell_{i'}'(t)\mathbf{v}_{i'}^{e'} \right|$$

$$+ \frac{2}{m n_e n_{e'}} \left| \sum_{i=1}^{n_e} \psi'(0)\ell_i'(t)\mathbf{v}_i^{e\top} \sum_{i'=1}^{n_{e'}} \psi'(t)\ell_{i'}'(t)\mathbf{v}_{i'}^{e'} - \sum_{i=1}^{n_e} \psi'(0)\ell_i'(t)\mathbf{v}_i^{e\top} \sum_{i'=1}^{n_{e'}} \psi'(0)\ell_{i'}'(t)\mathbf{v}_{i'}^{e'} \right|$$

$$+ \frac{2}{m n_e n_{e'}} \left| \sum_{i=1}^{n_e} \psi'(0)\ell_i'(t)\mathbf{v}_i^{e\top} \sum_{i'=1}^{n_{e'}} \psi'(0)\ell_{i'}'(t)\mathbf{v}_{i'}^{e'} - \sum_{i=1}^{n_e} \psi'(0)\ell_i'\mathbf{v}_i^{e\top} \sum_{i'=1}^{n_{e'}} \psi'(0)\frac{1}{2}\mathbf{v}_{i'}^{e'} \right|$$

$$+ \frac{2}{m n_e n_{e'}} \left| \sum_{i=1}^{n_e} \psi'(0)\ell_i'(t)\mathbf{v}_i^{e\top} \sum_{i'=1}^{n_{e'}} \psi'(0)\frac{1}{2}\mathbf{v}_{i'}^{e'} - \sum_{i=1}^{n_e} \psi'(0)\frac{1}{2}\mathbf{v}_i^{e\top} \sum_{i'=1}^{n_{e'}} \psi'(0)\frac{1}{2}\mathbf{v}_{i'}^{e'} \right|$$

$$\leq \frac{2}{m n_e n_{e'}} \left| \sum_{i=1}^{n_e} (\psi'(t) - \psi'(0))\ell_i'(t)\mathbf{v}_i^{e\top} \sum_{i'=1}^{n_{e'}} \ell_{i'}'(t)\psi'(t)\mathbf{v}_{i'}^{e'} \right| + \frac{2}{m n_e n_{e'}} \left| \sum_{i=1}^{n_e} \psi'(t)\ell_i'(t)\mathbf{v}_i^{e\top} \sum_{i'=1}^{n_{e'}} (\psi'(t) - \psi'(0))\frac{1}{2}\mathbf{v}_{i'}^{e'} \right|$$

$$+ \frac{2}{m n_e n_{e'}} \left| \sum_{i=1}^{n_e} \psi'(0)\ell_i'(t)\mathbf{v}_i^{e\top} \sum_{i'=1}^{n_{e'}} \psi'(0)\left(\ell_{i'}'(t) + \frac{1}{2}\right)\mathbf{v}_{i'}^{e'} \right| + \frac{2}{m n_e n_{e'}} \left| \sum_{i=1}^{n_e} \psi'(0)\left(\ell_i'(t) + \frac{1}{2}\right)\mathbf{v}_i^{e\top} \sum_{i'=1}^{n_{e'}} \psi'(0)\frac{1}{2}\mathbf{v}_{i'}^{e'} \right|$$

$$\triangleq I_1 + I_2 + I_3 + I_4$$

where we calculate each item as follows:

$$I_1 = \frac{2}{mn_e n_{e'}} \left| \sum_{i=1}^{n_e} (\psi'(t) - \psi'(0)) \ell_i'(t) \mathbf{v}_i^{e\top} \sum_{i'=1}^{n_{e'}} \ell_{i'}'(t) \psi'(t) \mathbf{v}_{i'}^{e'} \right|$$

$$\overset{(a)}{\leq} \frac{2}{mn_e n_{e'}} \left| \sum_{i=1}^{n_e} \beta \|\mathbf{w}_{j,r}(t) - \mathbf{w}_{j,r}(0)\|_2 \|\mathbf{v}_i^e\|_2 \ell_i'(t) \mathbf{v}_i^{e\top} \sum_{i'=1}^{n_{e'}} \ell_{i'}'(t) \psi'(t) \mathbf{v}_{i'}^{e'} \right|$$

$$\overset{(b)}{\leq} \frac{2\beta^2}{mn_e n_{e'}} \sum_{i=1}^{n_e} \|\mathbf{w}_{j,r}(t) - \mathbf{w}_{j,r}(0)\|_2 \|\mathbf{v}_i^e\|_2^2 \sum_{i'=1}^{n_{e'}} \|\mathbf{w}_{j,r}(t)\|_2 \|\mathbf{v}_{i'}^{e'}\|_2^2$$

$$\overset{(c)}{\leq} \frac{32\beta^2 R(R + \frac{3}{2}\sigma_0 d)}{m},$$

where we have used $R \triangleq \|\mathbf{w}_{j,r}(t)\|_2$. Besides, inequality (a) results from applying the smoothness property of the activation function and Cauchy–Schwarz inequality; inequality (b) is by smoothness property of the activation function and Cauchy–Schwarz inequality. Besides, we have used $|\ell_i^e| \leq 1$ for all $i \in n_e$ and $e \in \mathcal{E}_{\text{all}}$; inequality (c) is by the fact that $\|\mathbf{v}_i^e\|_2 \leq 2$ for all $i \in n_e$ and $e \in \mathcal{E}_{\text{all}}$ and Lemma D.8.

Similarly, we calculate the upper bound for $I_2$ as follows:

$$I_2 = \frac{2}{mn_e n_{e'}} \left| \sum_{i=1}^{n_e} \psi'(t) \ell_i'(t) \mathbf{v}_i^{e\top} \sum_{i'=1}^{n_{e'}} (\psi'(t) - \psi'(0)) \frac{1}{2} \mathbf{v}_{i'}^{e'} \right|$$

$$\leq \frac{32\beta^2 R(R + \frac{3}{2}\sigma_0 d)}{m}.$$

Next, we give the upper bound of $I_3$:

$$I_3 = \frac{2}{mn_e n_{e'}} \left| \sum_{i=1}^{n_e} \psi'(0) \ell_i'(t) \mathbf{v}_i^{e\top} \sum_{i'=1}^{n_{e'}} \psi'(0) \left( \ell_{i'}'(t) + \frac{1}{2} \right) \mathbf{v}_{i'}^{e'} \right|$$

$$\leq \frac{2}{mn_e n_{e'}} \left| \sum_{i=1}^{n_e} \beta \|\mathbf{w}_{j,r}(0)\|_2 \|\mathbf{v}_i^e\|_2 \ell_i'(t) \mathbf{v}_i^{e\top} \sum_{i'=1}^{n_{e'}} \beta \|\mathbf{w}_{j,r}(0)\|_2 \|\mathbf{v}_{i'}^{e'}\|_2 \left( \ell_{i'}'(t) + \frac{1}{2} \right) \mathbf{v}_{i'}^{e'} \right|$$

$$\leq \frac{64\beta^2 LR(\frac{3}{2}\sigma_0 d)^2}{m},$$

where we have used $\gamma$ which is defined as follows:

$$|\hat{y}_i^e(t)| = \left| \frac{1}{m} \sum_{j} \sum_{r=1}^{m} \left[ \psi(\mathbf{w}_{j,r}^\top(t)\mathbf{x}_1) + \psi(\mathbf{w}_{j,r}^\top(t)\mathbf{x}_2) \right] \right|$$

$$\overset{(a)}{\leq} 2LR,$$

where inequality (a) is by the Lipschitz property of non-linear activation function and we have used the bound for $\ell_i'(t) + \frac{1}{2}$:

$$\left| \ell_i'(t) + \frac{1}{2} \right| = \left| -\frac{\exp(-y_i^e \cdot f(\mathbf{W}, \mathbf{x}_i, t))}{1 + \exp(-y_i^e \cdot f(\mathbf{W}, \mathbf{x}_i, t))} + \frac{1}{2} \right|$$

$$= \left| \frac{1}{2} - \frac{1}{1 + \exp(y_i^e \cdot f(\mathbf{W}, \mathbf{x}_i, t))} \right|$$

$$\leq \max \left\{ \left| \frac{1}{2} - \frac{1}{1 + \exp(2LR)} \right|, \left| \frac{1}{2} - \frac{1}{1 + \exp(-2LR)} \right| \right\}$$

$$\leq \max \left\{ \left| \frac{1}{2} - \frac{1}{2 + \frac{7}{4}2LR} \right|, \left| \frac{1}{2} - \frac{1}{2 - 2LR} \right| \right\} = \Theta(LR).$$

and we provide the bound of $\ell_i''(t) - \frac{1}{4}$:

$$
\left| \ell_i''(t) - \frac{1}{4} \right| = \left| \frac{\exp(-y_i^e \cdot f(\mathbf{W}, \mathbf{x}_i, t))}{(1 + \exp(-y_i^e \cdot f(\mathbf{W}, \mathbf{x}_i, t)))^2} - \frac{1}{4} \right|
$$

$$
= \left| \frac{1}{\exp(y_i^e \cdot f(\mathbf{W}, \mathbf{x}_i, t)) + 2 + \exp(-y_i^e \cdot f(\mathbf{W}, \mathbf{x}_i, t))} - \frac{1}{4} \right|
$$

$$
\leq \left| \frac{1}{4} - \frac{1}{2 + 2\exp((2LR)^2/2)} \right| = \Theta((LR)^2).
$$

Similarly, we give the upper bound of $I_4$:

$$
I_4 = \frac{2}{mn_e n_{e'}} \left| \sum_{i=1}^{n_e} \psi'(0) \left( \ell_i'(t) + \frac{1}{2} \right) \mathbf{v}_i^{e\top} \sum_{i'=1}^{n_{e'}} \psi'(0) \frac{1}{2} \mathbf{v}_{i'}^{e'} \right|
$$

$$
\leq \frac{2}{mn_e n_{e'}} \left| \sum_{i=1}^{n_e} \beta \|\mathbf{w}_{j,r}(0)\|_2 \|\mathbf{v}_i^e\|_2 (\ell_i'(t) + \frac{1}{2}) \mathbf{v}_i^{e\top} \sum_{i'=1}^{n_{e'}} \beta \|\mathbf{w}_{j,r}(0)\|_2 \|\mathbf{v}_{i'}^{e'}\|_2 \frac{1}{2} \mathbf{v}_{i'}^{e'} \right|
$$

$$
\leq \frac{64 \beta^2 LR \gamma (\frac{3}{2} \sigma_0 d)^2}{m}.
$$

Together, we obtain the upper bound for $\left| \mathbf{H}_{e,e'}^1(t) - \mathbf{H}_{e,e'}^{1,\infty} \right|$:

$$
\left| \mathbf{H}_{e,e'}^1(t) - \mathbf{H}_{e,e'}^{1,\infty} \right| \leq \frac{64 \beta^2 R(R + \frac{3}{2}\sigma_0 d)}{m} + \frac{128 \beta^2 LR(\frac{3}{2}\sigma_0 d)^2}{m}.
$$

Then we calculate the upper bound for the residual terms:

$$
\left| \mathbf{H}_{e,e'}^2(t) \right| = \left| \sum_j \sum_{r=1}^m \left( \frac{1}{n_e m} \right) \left( \frac{1}{n_{e'} m} \right) \sum_{i=1}^{n_e} \psi' \ell_i''(t) \hat{y}_i^e(t) j y_i^e \mathbf{v}_i^{e\top} \sum_{i'=1}^{n_{e'}} \psi' \ell_{i'}''(t) \hat{y}_{i'}^{e'}(t) j y_{i'}^{e'} \mathbf{v}_{i'}^{e'} \right|
$$

$$
= \frac{2}{mn_e n_{e'}} \left| \sum_{i=1}^{n_e} \psi'(t) \ell_i''(t) \hat{y}_i^e(t) j y_i^e \mathbf{v}_i^{e\top} \sum_{i'=1}^{n_{e'}} \psi'(t) \ell_{i'}''(t) \hat{y}_{i'}^{e'}(t) j y_{i'}^{e'} \mathbf{v}_{i'}^{e'} \right|
$$

$$
\overset{(a)}{\leq} \frac{2\beta^2}{mn_e n_{e'}} \left| \sum_{i=1}^{n_e} \|\mathbf{w}_{j,r}(t)\|_2 \|\mathbf{v}_i^e\|_2 \ell_i''(t) \hat{y}_i^e(t) \mathbf{v}_i^{e\top} \sum_{i'=1}^{n_{e'}} \|\mathbf{w}_{j,r}(t)\|_2 \|\mathbf{v}_{i'}^{e'}\|_2 \ell_{i'}''(t) \hat{y}_i^e(t) \mathbf{v}_{i'}^{e'} \right|
$$

$$
\overset{(b)}{\leq} \frac{128 \beta^2 L^2 R^4}{m},
$$

where inequality (a) is by the smoothness property of the activation function and Cauchy–Schwarz inequality, and inequality (b) is by triangle inequality and the fact that $\|\mathbf{v}_i^e\|_2 \leq 2$ for all $i \in n_e$ and $e \in \mathcal{E}_{\text{all}}$, and $|\ell_i''| \leq 1$ for all $i \in [n]$ and Lemma D.8. Similarly, we further provide the upper bound of residual terms:

$$
\left| \mathbf{H}_{e,e'}^3(t) \right| = \left| \sum_j \sum_{r=1}^m \left( \frac{1}{n_e m} \right) \left( \frac{1}{n_{e'} m} \right) \sum_{i=1}^{n_e} \psi' \ell_i''(t) \hat{y}_i^e(t) j y_i^e \mathbf{v}_i^{e\top} \sum_{i'=1}^{n_{e'}} \psi' \ell_{i'}'(t) j \mathbf{v}_{i'}^{e'} \right|
$$

$$
= \frac{2}{mn_e n_{e'}} \left| \sum_{i=1}^{n_e} \psi'(t) \ell_i''(t) \hat{y}_i^e(t) j y_i^e \mathbf{v}_i^{e\top} \sum_{i'=1}^{n_{e'}} \psi'(t) \ell_{i'}'(t) j \mathbf{v}_{i'}^{e'} \right|
$$

$$
\leq \frac{2\beta^2}{mn_e n_{e'}} \left| \sum_{i=1}^{n_e} \|\mathbf{w}_{j,r}(t)\|_2 \|\mathbf{v}_i^e\|_2 \ell_i''(t) \hat{y}_i^e(t) \mathbf{v}_i^{e\top} \sum_{i'=1}^{n_{e'}} \|\mathbf{w}_{j,r}(t)\|_2 \|\mathbf{v}_{i'}^{e'}\|_2 \ell_{i'}'(t) \mathbf{v}_{i'}^{e'} \right|
$$

$$
\leq \frac{64 \beta^2 LR^3}{m}.
$$

Similarly, we further have that:

$$
\begin{aligned}
\left|\mathbf{H}_{e,e'}^{4}(t)\right| &= \left|\sum_{j}\sum_{r=1}^{m}\left(\frac{1}{n_e m}\right)\left(\frac{1}{n_{e'}m}\right)\sum_{i=1}^{n_e}\psi'\ell_i'(t)j\mathbf{v}_i^{e\top}\sum_{i'=1}^{n_{e'}}\psi'\ell_{i'}''(t)\hat{y}_{i'}^{e'}(t)j\mathbf{v}_{i'}^{e'}\right| \\
&= \frac{2}{mn_e n_{e'}}\left|\sum_{i=1}^{n_e}\psi'(t)\ell_i'(t)j\mathbf{v}_i^{e\top}\sum_{i'=1}^{n_{e'}}\psi'(t)\ell_{i'}''(t)\hat{y}_{i'}^{e'}(t)j\mathbf{v}_{i'}^{e'}\right| \\
&\leq \frac{2\beta^2}{mn_e n_{e'}}\left|\sum_{i=1}^{n_e}\|\mathbf{w}_{j,r}(t)\|_2\|\mathbf{v}_i^e\|_2\ell_i'(t)\mathbf{v}_i^{e\top}\sum_{i'=1}^{n_{e'}}\|\mathbf{w}_{j,r}(t)\|_2\|\mathbf{v}_{i'}^{e'}\|_2\ell_{i'}''(t)\hat{y}_{i'}^{e'}(t)j\mathbf{v}_{i'}^{e'}\right| \\
&\leq \frac{64\beta^2 LR^3}{m}.
\end{aligned}
$$

Keep going on, we provide the computation results further:

$$
\begin{aligned}
\left|\mathbf{H}_{e,e'}^{5}(t)\right| &= \left|\sum_{j}\sum_{r=1}^{m}\left(\frac{1}{n_e m}\right)\left(\frac{1}{n_{e'}m}\right)\sum_{i=1}^{n_e}\psi'\ell_i'(t)jy_i^e\boldsymbol{\xi}_i^{e\top}\sum_{i'=1}^{n_{e'}}\psi'\ell_{i'}'(t)jy_{i'}^{e'}\boldsymbol{\xi}_{i'}^{e'}\right| \\
&\overset{(a)}{\leq} \frac{2\beta^2}{mn_e n_{e'}}\left|\sum_{i=1}^{n_e}\|\mathbf{w}_{j,r}(t)\|_2\|\boldsymbol{\xi}_i\|_2\ell_i'(t)\boldsymbol{\xi}_i^{e\top}\sum_{i'=1}^{n_{e'}}\|\mathbf{w}_{j,r}(t)\|_2\|\boldsymbol{\xi}_{i'}^{e'}\|_2\ell_{i'}'(t)\boldsymbol{\xi}_{i'}^{e'}\right| \\
&\overset{(b)}{\leq} \frac{2\beta^2 R^2\sigma_p^2 d}{m},
\end{aligned}
$$

where inequality (a) is by smoothness property of non-linear activation function and Cauchy inequality, inequality (b) is by Lemma D.6 and Lemma D.8. Next, we calculate the

$$
\begin{aligned}
\left|\mathbf{H}_{e,e'}^{6}(t)\right| &= \left|\sum_{j}\sum_{r=1}^{m}\left(\frac{1}{n_e m}\right)\left(\frac{1}{n_{e'}m}\right)\sum_{i=1}^{n_e}\psi'\ell_i''(t)\hat{y}_i^e(t)j\boldsymbol{\xi}_i^{e\top}\sum_{i'=1}^{n_{e'}}\psi'\ell_{i'}''(t)\hat{y}_{i'}^{e'}(t)j\boldsymbol{\xi}_{i'}^{e'}\right| \\
&\leq \frac{2\beta^2}{mn_e n_{e'}}\left|\sum_{i=1}^{n_e}\|\mathbf{w}_{j,r}(t)\|_2\|\boldsymbol{\xi}_i\|_2\ell_i''(t)\hat{y}_i^e\boldsymbol{\xi}_i^{e\top}\sum_{i'=1}^{n_{e'}}\|\mathbf{w}_{j,r}(t)\|_2\|\boldsymbol{\xi}_{i'}^{e'}\|_2\ell_{i'}''(t)\hat{y}_{i'}^{e'}(t)\boldsymbol{\xi}_{i'}^{e'}\right| \\
&\leq \frac{8\beta^2 dL^4 R^2}{m}.
\end{aligned}
$$

Similarly, the next $\mathbf{H}$ term can be calculated as follows:

$$
\begin{aligned}
\left|\mathbf{H}_{e,e'}^{7}(t)\right| &= \left|\sum_{j}\sum_{r=1}^{m}\left(\frac{1}{n_e m}\right)\left(\frac{1}{n_{e'}m}\right)\sum_{i=1}^{n_e}\psi'\ell_i''(t)\hat{y}_i^e j\boldsymbol{\xi}_i^{e\top}\sum_{i'=1}^{n_{e'}}\psi'\ell_{i'}'(t)jy_{i'}^{e'}\boldsymbol{\xi}_{i'}^{e'}\right| \\
&\leq \frac{2\beta^2}{mn_e n_{e'}}\left|\sum_{i=1}^{n_e}\|\mathbf{w}_{j,r}(t)\|_2\|\boldsymbol{\xi}_i\|_2\ell_i''(t)\hat{y}_i^e(t)\boldsymbol{\xi}_i^{e\top}\sum_{i'=1}^{n_{e'}}\|\mathbf{w}_{j,r}(t)\|_2\|\boldsymbol{\xi}_{i'}^{e'}\|_2\ell_{i'}'(t)\boldsymbol{\xi}_{i'}^{e'}\right| \\
&\leq \frac{4\beta^2\sigma_p^2 dLR^3}{m}.
\end{aligned}
$$

Finally, we have the upper for the last term:

$$
\begin{aligned}
\left|\mathbf{H}_{e,e'}^{8}(t)\right| &= \left|\sum_{j}\sum_{r=1}^{m}\left(\frac{1}{n_e m}\right)\left(\frac{1}{n_{e'}m}\right)\sum_{i=1}^{n_e}\psi'y_i^e\ell_i'(t)j\boldsymbol{\xi}_i^{e\top}\sum_{i'=1}^{n_{e'}}\psi'\ell_{i'}''(t)j\boldsymbol{\xi}_{i'}^e\hat{y}_{i'}^{e'}(t)\right| \\
&\leq \frac{2\beta^2}{mn_e n_{e'}}\left|\sum_{i=1}^{n_e}\|\mathbf{w}_{j,r}(t)\|_2\|\boldsymbol{\xi}_i\|_2\ell_i'(t)\boldsymbol{\xi}_i^{e\top}\sum_{i'=1}^{n_{e'}}\|\mathbf{w}_{j,r}(t)\|_2\|\boldsymbol{\xi}_{i'}^{e'}\|_2\ell_{i'}''(t)\hat{y}_{i'}^{e'}(t)\boldsymbol{\xi}_{i'}^{e'}\right| \\
&\leq \frac{4\beta^2\sigma_p^2 dLR^3}{m}.
\end{aligned}
$$

To summarize, we have that,

$$\left|\mathbf{H}_{e,e'}(t) - \mathbf{H}_{e,e'}^{\infty}\right| \leq \frac{32\beta^2 R(R + \frac{3}{2}\sigma_0 d)}{m} + \frac{128\beta^2(R + \frac{3}{2}\sigma_0^2 d)^2 L^2 R^2}{m} + \frac{128\beta^2 L R^3}{m}$$
$$+ \frac{2\beta^2 R^2 \sigma_p^2 d}{m} + \frac{8\beta^2 \sigma_p^2 d L^2 R^4}{m} + \frac{4\beta^2 \sigma_p^2 d L R^3}{m}$$
$$\leq O\left(\frac{\beta^2 L R}{m}\right).$$

where we have used $\sigma_p = O(d^{-2})$, $R = o(1)$, and $\sigma_0 = O(\sqrt{R/d})$. Furthermore, we show that the perturbation term in Equation (22) is bounded during training. In particular, we show the complete expression:

$$\mathbf{g}_e(t) = \sum_{j=\pm 1}\sum_{r=1}^{m}\left\langle \frac{\partial C_{\text{IRMv1}}^e(\mathbf{W},t)}{\partial \mathbf{w}_{j,r}(t)}, \frac{\partial L(\mathbf{W},t)}{\partial \mathbf{w}_{j,r}(t)}\right\rangle$$

$$= \sum_{j=\pm 1}\sum_{r=1}^{m}\left[\frac{1}{n_e m}\sum_{i=1}^{n_e}\ell_i'(t)\psi'(\langle \mathbf{w}_{j,r}(t), y_i^e \mathbf{v}_i^e\rangle) \cdot j\mathbf{v}_i^e \frac{1}{n_e m}\sum_{i=1}^{n_e}\ell_i'(t)\psi'(\langle \mathbf{w}_{j,r}(t), y_i^e \mathbf{v}_i^e\rangle) \cdot j\mathbf{v}_i^e\right.$$

$$+ \frac{1}{n_e m}\sum_{i=1}^{n_e}\ell_i''\hat{y}_i^e\psi'(\langle \mathbf{w}_{j,r}(t), y_i^e \mathbf{v}_i^e\rangle) \cdot jy_i^e\mathbf{v}_i^e \frac{1}{n_e m}\sum_{i=1}^{n_e}\ell_i'(t)\psi'(\langle \mathbf{w}_{j,r}(t), y_i^e \mathbf{v}_i^e\rangle) \cdot j\mathbf{v}_i^e$$

$$+ \frac{\eta}{n_e m}\sum_{i=1}^{n_e}\ell_i'(t)\psi'(\langle \mathbf{w}_{j,r}(t), \boldsymbol{\xi}_i\rangle) \cdot jy_i^e\boldsymbol{\xi}_i \frac{\eta}{n_e m}\sum_{i=1}^{n_e}\ell_i'(t)\psi'(\langle \mathbf{w}_{j,r}(t), \boldsymbol{\xi}_i\rangle) \cdot jy_i^e\boldsymbol{\xi}_i$$

$$\left. + \frac{\eta}{n_e m}\sum_{i=1}^{n_e}\ell_i'(t)\psi'(\langle \mathbf{w}_{j,r}(t), \boldsymbol{\xi}_i\rangle) \cdot jy_i^e\boldsymbol{\xi}_i \sum_{i=1}^{n_e}\ell_i''\hat{y}_i^e\psi'(\langle \mathbf{w}_{j,r}(t), \boldsymbol{\xi}_i\rangle) \cdot j\boldsymbol{\xi}_i\right]$$

$$\triangleq I_1 + I_2 + I_3 + I_4.$$

Similar to the computation process for matrix $\mathbf{H}$, we adopt a divide and conquer manner:

$$|I_1| \leq \frac{2\beta^2}{m n_e n_e}\left|\sum_{i=1}^{n_e}\|\mathbf{w}_{j,r}(t)\|_2\|\mathbf{v}_i^e\|_2\ell_i'(t)\mathbf{v}_i^{e\top}\sum_{i=1}^{n_e}\|\mathbf{w}_{j,r}(t)\|_2\|\mathbf{v}_i^e\|_2\ell_{i'}'(t)\mathbf{v}_i^e\right|$$

$$\leq \frac{32\beta^2(R + \frac{3}{2}\sigma_0^2 d)^2}{m}.$$

The techniques used are the same when deriving upper bound for matrix $\mathbf{H}$. Next, we have

$$|I_2| \leq \frac{2\beta^2}{m n_e n_e}\left|\sum_{i=1}^{n_e}\|\mathbf{w}_{j,r}(t)\|_2\|\mathbf{v}_i^e\|_2\ell_i''(t)\hat{y}_i^e(t)\mathbf{v}_i^{e\top}\sum_{i=1}^{n_e}\|\mathbf{w}_{j,r}(t)\|_2\|\mathbf{v}_i^e\|_2\ell_{i'}'(t)\mathbf{v}_i^e\right|$$

$$\leq \frac{64\beta^2 R^2 L R}{m}.$$

The last second term can be calculated as follows:

$$|I_3| \leq \frac{2\beta^2}{m n_e n_e}\left|\sum_{i=1}^{n_e}\|\mathbf{w}_{j,r}(t)\|_2\|\boldsymbol{\xi}_i\|_2\ell_i'(t)\boldsymbol{\xi}_i^{e\top}\sum_{i=1}^{n_e}\|\mathbf{w}_{j,r}(t)\|_2\|\boldsymbol{\xi}_i^e\|_2\ell_i'(t)\boldsymbol{\xi}_i^e\right|$$

$$\leq \frac{2\beta^2 R^2 \sigma_p^2 d}{m},$$

Finally, we show the upper bound of last term:

$$|I_4| \leq \frac{2\beta^2}{m n_e n_e}\left|\sum_{i=1}^{n_e}\|\mathbf{w}_{j,r}(t)\|_2\|\boldsymbol{\xi}_i\|_2\ell_i''(t)\hat{y}_i^e(t)\boldsymbol{\xi}_i^{e\top}\sum_{i=1}^{n_e}\|\mathbf{w}_{j,r}(t)\|_2\|\boldsymbol{\xi}_i^e\|_2\ell_{i'}'(t)\boldsymbol{\xi}_i^e\right|$$

$$\leq \frac{4\beta^2 \sigma_p^2 d L R^3}{m}.$$

In a summary, we have the following inequality:

$$|\mathbf{g}_e(t)| \leq \frac{32\beta^2 R^2}{m} + \frac{64\beta^2 LR^3}{m} + \frac{2\beta^2 R^2 \sigma_p^2 d}{m} + \frac{4\beta^2 R^3 \sigma_p^2 dL}{m}$$

$$\leq O\left(\frac{\beta^2 LR^2}{m}\right),$$

where we have used $\sigma_p = O(d^{-2})$ and $R = o(1)$. With all the bounds at hand, we are ready to have the dynamics for $\|\mathbf{c}(t)\|_2^2$

$$\frac{d\|\mathbf{c}(t)\|_2^2}{dt} = -2\lambda \mathbf{c}^\top(t) \mathbf{H}(t)\mathbf{c}(t) - \mathbf{c}(t)\mathbf{g}(t) \leq -\lambda_0 \lambda \|\mathbf{c}(t)\|_2^2, \tag{23}$$

which requires that $\|\mathbf{H}(t) - \mathbf{H}^\infty\|_2 \leq \lambda_0$. This leads to the following inequality:

$$\|\mathbf{H}(t) - \mathbf{H}^\infty\|_2 \leq \|\mathbf{H}(t) - \mathbf{H}^\infty\|_F \leq \sum_{i,j} |\mathbf{H}_{ij}(t) - \mathbf{H}_{ij}^\infty|$$

$$\leq \frac{|\mathcal{E}_{tr}|^2 \beta^2 LR}{m} \leq \lambda_0.$$

which leads to the conclusion for $R$ as follows:

$$R \leq \frac{\lambda_0 m}{|\mathcal{E}_{tr}|^2 \beta^2 L}. \tag{24}$$

Besides, we have the inequality that

$$\|\mathbf{g}\|_2 \leq \frac{\sqrt{|\mathcal{E}_{tr}|}\beta^2 LR}{m} \leq \lambda\lambda_0\|\mathbf{c}(0)\|_2. \tag{25}$$

Combined with Equation (24), we obtain the condition for $\lambda$ as follows:

$$\lambda \geq 1/(\sigma_0\sqrt{|\mathcal{E}_{tr}|}^3). \tag{26}$$

By inequality (23), taking the convergence time $T = \Omega\left(\frac{\log(\sigma_0/\epsilon)}{\eta\lambda\lambda_0}\right)$ we have that:

$$\|\mathbf{c}(T)\|_2 \leq \epsilon.$$

According to the gradient descent for IRMV1 objective function, the evolution of coefficients can be expressed as:

$$\gamma_{j,r}^{inv}(t+1) = \gamma_{j,r}^{inv}(t) - \frac{\eta}{m} \cdot \sum_{e \in \mathcal{E}_{tr}} (1 + 2\lambda C_{\text{IRMv1}}^e(t))\frac{1}{n_e}\sum_{i=1}^{n_e} \ell_i'(t)\psi_i'(t)\text{Rad}(\alpha)_i$$

$$- \frac{\eta\lambda}{m} \cdot \sum_{e \in \mathcal{E}_{tr}} 2C_{\text{IRMv1}}^e \frac{1}{n_e}\sum_{i=1}^{n_e} \ell_i''\psi_i'(t)\hat{y}_i^e \cdot y_i^e\text{Rad}(\alpha)_i,$$

$$\gamma_{j,r}^{spu}(t+1) = \gamma_{j,r}^{spu}(t) - \frac{\eta}{m} \cdot \sum_{e \in \mathcal{E}_{tr}} (1 + 2\lambda C_{\text{IRMv1}}^e(t))\frac{1}{n_e}\sum_{i=1}^{n_e} \ell_i'(t)\psi_i'(t)\text{Rad}(\beta_e)$$

$$- \frac{\eta\lambda}{m} \cdot \sum_{e \in \mathcal{E}_{tr}} 2C_{\text{IRMv1}}^e \frac{1}{n_e}\sum_{i=1}^{n_e} \psi_i'(t)\ell_i''\hat{y}_i^e \cdot y_i^e\text{Rad}(\beta_e)_i.$$

Then we have,

$$|\gamma_{j,r}^{inv}(t+1)| \leq |\gamma_{j,r}^{inv}(t)| + \left|\frac{\eta}{m} \cdot \sum_{e \in \mathcal{E}_{tr}} (1 + 2\lambda C_{\text{IRMv1}}^e(t))\frac{1}{n_e}\sum_{i=1}^{n_e} \psi_i'(t)\ell_i'(t)\text{Rad}(\alpha)_i\right|$$

$$+ \left|\frac{\eta\lambda}{m} \cdot \sum_{e \in \mathcal{E}_{tr}} 2C_{\text{IRMv1}}^e \frac{1}{n_e}\sum_{i=1}^{n_e} \ell_i''\psi_i'(t)\hat{y}_i^e \cdot y_i^e\text{Rad}(\alpha)_i\right|$$

$$\leq |\gamma_{j,r}^{inv}(t)| + C\frac{\eta\sqrt{|\mathcal{E}_{tr}|}\lambda\beta R^2 L}{m}\|\mathbf{c}(t)\|_2.$$

Similarly, we have,

$$|\gamma_{j,r}^{spu}(t+1)| \leq |\gamma_{j,r,2}^{spu}(t)| + C\frac{\eta\sqrt{|\mathcal{E}_{tr}|}\lambda\beta R^2 L}{m}\|\mathbf{c}(t)\|_2.$$

At the time step $T$, the feature learning satisfies that:

$$\gamma_{j,r}^{inv}(T) \leq C\frac{\eta\sqrt{|\mathcal{E}_{tr}|}\lambda\beta R^2 LT}{m}\|\mathbf{c}(0)\|_2; \quad \gamma_{j,r}^{spu}(T) \leq C\frac{\eta\sqrt{|\mathcal{E}_{tr}|}\lambda\beta R^2 LT}{m}\|\mathbf{c}(0)\|_2.$$

To make sure that $\gamma_{j,r}^{inv}(T) = o(1)$ and $\gamma_{j,r}^{spu}(T) = o(1)$, we need the following condition:

$$C\frac{\eta\sqrt{|\mathcal{E}_{tr}|}\lambda\beta R^2 LT}{m}\|\mathbf{c}(0)\|_2 \leq d^{-\frac{1}{2}}, \tag{27}$$

combined with inequality (24) and inequality (26), we have:

$$\sigma_0 \leq \frac{|\mathcal{E}_{tr}|^{7/2}\beta^3 L}{d^{1/2}m^2\lambda_0^2\log(1/\epsilon)}.$$

$\square$

### D.4 Proof for Proposition 4.3

**Proposition D.9** (Restatement of Proposition 4.3). *Consider training the CNN model with the same data as Theorem 4.1, suppose that $\psi(x) = x$, $\gamma_{j,r,1}(t_1) = \gamma_{j,r,1}(t_1 - 1)$, and $\gamma_{j,r,2}(t_1) = \gamma_{j,r,2}(t_1 - 1)$ at the end of ERM pre-train $t_1$ and $\mathcal{E}_{tr} = \{(0.25, 0.1), (0.25, 0.2)\}$. Suppose that $\delta > 0$, and $n > C\log(1/\delta)$, with $C$ being a positive constant, then with a high probability at least $1 - \delta$, we have*

- $\sum_e C_{IRMvI}^e(t_1) = 0$.
- $\gamma_{j,r,1}(t_1 + 1) > \gamma_{j,r,1}(t_1)$.
- $\gamma_{j,r,2}(t_1 + 1) < \gamma_{j,r,2}(t_1)$.

*Proof of Proposition D.9.* According to the gradient descent for IRMV1 objective function, the evolution of coefficients can be expressed as:

$$\gamma_{j,r,1}(t+1) = \gamma_{j,r,1}(t) - \frac{\eta}{m}\cdot\sum_{e\in\mathcal{E}_{tr}}(1 + 2\lambda C_{IRMv1}^e(t))\frac{1}{n_e}\sum_{i=1}^{n_e}\ell_i'(t)\text{Rad}(\alpha)_i$$

$$- \frac{\eta\lambda}{m}\cdot\sum_{e\in\mathcal{E}_{tr}}2C_{IRMv1}^e\frac{1}{n_e}\sum_{i=1}^{n_e}\ell_i''\hat{y}_i^e\cdot y_i^e\text{Rad}(\alpha)_i,$$

$$\gamma_{j,r,2}(t+1) = \gamma_{j,r,2}(t) - \frac{\eta}{m}\cdot\sum_{e\in\mathcal{E}_{tr}}(1 + 2\lambda C_{IRMv1}^e(t))\frac{1}{n_e}\sum_{i=1}^{n_e}\ell_i'(t)\text{Rad}(\beta_e)$$

$$- \frac{\eta\lambda}{m}\cdot\sum_{e\in\mathcal{E}_{tr}}2C_{IRMv1}^e\frac{1}{n_e}\sum_{i=1}^{n_e}\ell_i''\hat{y}_i^e\cdot y_i^e\text{Rad}(\beta_e)_i,$$

where $\ell''(y_i^e\cdot f(\mathbf{W}, \mathbf{x}_i^e)) = \frac{\exp(-y_i^e\cdot f(\mathbf{W}, \mathbf{x}_i))}{(1 + \exp(-y_i^e\cdot f(\mathbf{W}, \mathbf{x}_i)))^2}$.

To simplify the notation, we further define

$$A_1^e = \frac{1}{n_e}\sum_{i=1}^{n_e}\ell_i'\text{Rad}(\alpha)_i$$

and

$$A_2^e = \frac{1}{n_e}\sum_{i=1}^{n_e}\ell_i''\hat{y}_i^e y_i^e\text{Rad}(\alpha)_i$$

. Similarly, we define

$$B_1^e = \frac{1}{n_e} \sum_{i=1}^{n_e} \ell_i' \mathrm{Rad}(\beta_e)_i$$

and

$$B_2^e = \frac{1}{n_e} \sum_{i=1}^{n_e} \ell_i'' \hat{y}_i^e y_i^e \mathrm{Rad}(\beta_e)_i.$$

In the limit of $n \to \infty$, we have:

$$\lim_{n\to\infty} A_1^1(t_1) = -1/(1 + e^{(\gamma_1+\gamma_2)})(1-\alpha)(1-\beta_1) - 1/(1 + e^{\gamma_1-\gamma_2})(1-\alpha)\beta_1$$
$$+ 1/(1 + e^{\gamma_2-\gamma_1})\alpha(1-\beta_1) + 1/(1 + e^{-\gamma_1-\gamma_2})\alpha\beta_1,$$
$$\lim_{n\to\infty} A_1^2(t_1) = -1/(1 + e^{\gamma_1+\gamma_2})(1-\alpha)(1-\beta_2) - 1/(1 + e^{\gamma_1-\gamma_2})(1-\alpha)\beta_2$$
$$+ 1/(1 + e^{-\gamma_1+\gamma_2})\alpha(1-\beta_2) + 1/(1 + e^{-\gamma_1-\gamma_2})\alpha\beta_2,$$
$$\lim_{n\to\infty} B_1^1(t_1) = -1/(1 + e^{\gamma_1+\gamma_2})(1-\alpha)(1-\beta_1) + 1/(1 + e^{\gamma_1-\gamma_2})(1-\alpha)\beta_1$$
$$- 1/(1 + e^{-\gamma_1+\gamma_2})\alpha(1-\beta_1) + 1/(1 + e^{-\gamma_1-\gamma_2})\alpha\beta_1,$$
$$\lim_{n\to\infty} B_1^2(t_1) = -1/(1 + e^{\gamma_1+\gamma_2})(1-\alpha)(1-\beta_2) + 1/(1 + e^{\gamma_1-\gamma_2})(1-\alpha)\beta_2$$
$$- 1/(1 + e^{-\gamma_1+\gamma_2})\alpha(1-\beta_2) + 1/(1 + e^{-\gamma_1-\gamma_2})\alpha\beta_2.$$

and,

$$\lim_{n\to\infty} A_2^1(t_1) = e^{\gamma_1+\gamma_2}/(1 + e^{\gamma_1+\gamma_2})^2(1-\alpha)(1-\beta_1)(\gamma_1+\gamma_2) + e^{\gamma_1-\gamma_2}/(1 + e^{\gamma_1-\gamma_2})^2(1-\alpha)\beta_1(\gamma_1-\gamma_2)$$
$$+ e^{-\gamma_1+\gamma_2}/(1 + e^{-\gamma_1+\gamma_2})^2\alpha(1-\beta_1)(\gamma_1-\gamma_2) + e^{-\gamma_1-\gamma_2}/(1 + e^{-\gamma_1-\gamma_2})^2\alpha\beta_1(\gamma_1+\gamma_2),$$
$$\lim_{n\to\infty} A_2^2(t_1) = e^{\gamma_1+\gamma_2}/(1 + e^{\gamma_1+\gamma_2})^2(1-\alpha)(1-\beta_2)(\gamma_1+\gamma_2) + e^{\gamma_1-\gamma_2}/(1 + e^{\gamma_1-\gamma_2})^2(1-\alpha)\beta_2(\gamma_1-\gamma_2)$$
$$+ e^{-\gamma_1+\gamma_2}/(1 + e^{-\gamma_1+\gamma_2})^2\alpha(1-\beta_2)(\gamma_1-\gamma_2) + e^{-\gamma_1-\gamma_2}/(1 + e^{-\gamma_1-\gamma_2})^2\alpha\beta_2(\gamma_1+\gamma_2),$$
$$\lim_{n\to\infty} B_2^1(t_1) = e^{\gamma_1+\gamma_2}/(1 + e^{\gamma_1+\gamma_2})^2(1-\alpha)(1-\beta_1)(\gamma_1+\gamma_2) + e^{\gamma_1-\gamma_2}/(1 + e^{\gamma_1-\gamma_2})^2(1-\alpha)\beta_1(-\gamma_1+\gamma_2)$$
$$+ e^{-\gamma_1+\gamma_2}/(1 + e^{-\gamma_1+\gamma_2})^2\alpha(1-\beta_1)(-\gamma_1+\gamma_2) + e^{-\gamma_1-\gamma_2}/(1 + e^{-\gamma_1-\gamma_2})^2\alpha\beta_1(\gamma_1+\gamma_2),$$
$$\lim_{n\to\infty} B_2^2(t_1) = e^{\gamma_1+\gamma_2}/(1 + e^{\gamma_1+\gamma_2})^2(1-\alpha)(1-\beta_2)(\gamma_1+\gamma_2) + e^{\gamma_1-\gamma_2}/(1 + e^{\gamma_1-\gamma_2})^2(1-\alpha)\beta_2(-\gamma_1+\gamma_2)$$
$$+ e^{-\gamma_1+\gamma_2}/(1 + e^{-\gamma_1+\gamma_2})^2\alpha(1-\beta_2)(-\gamma_1+\gamma_2) + e^{-\gamma_1-\gamma_2}/(1 + e^{-\gamma_1-\gamma_2})^2\alpha\beta_2(\gamma_1+\gamma_2).$$

Because $\mathrm{Rad}(\alpha)_i$ and $\mathrm{Rad}(\beta)_i$ are random variables, applying Hoeffding's inequality, we have with probability at least $1 - \delta$,

$$\left| A_1^1(t_1) - \lim_{n\to\infty} A_1^1(t_1) \right| \leq \sqrt{\frac{4\log(1/\delta)}{n}}.$$

Similarly, we can apply the concentration bound to other quantities and obtain the same bound.

By the assumption that $\gamma_{j,r,1}(t_1) = \gamma_{j,r,1}(t_1 - 1)$ and $\gamma_{j,r,2}(t_1) = \gamma_{j,r,2}(t_1 - 1)$, we have that $\sum_e A_1^e(t_1) = \sum_e B_1^e(t_1) = 0$:

$$\lim_{n\to\infty} (A_1^1(t_1) + A_1^2(t_1)) = -1/(1 + e^{\gamma_1+\gamma_2})(1-\alpha)(2-\beta_1-\beta_2) - 1/(1 + e^{\gamma_1-\gamma_2})(1-\alpha)(\beta_1+\beta_2)$$
$$+ 1/(1 + e^{-\gamma_1+\gamma_2})\alpha(2-\beta_1-\beta_2) + 1/(1 + e^{-\gamma_1-\gamma_2})\alpha(\beta_1+\beta_2) = 0$$
$$\lim_{n\to\infty} (B_1^1(t_1) + B_1^2(t_1)) = -1/(1 + e^{\gamma_1+\gamma_2})(1-\alpha)(2-\beta_1-\beta_2) + 1/(1 + e^{\gamma_1-\gamma_2})(1-\alpha)(\beta_1+\beta_2)$$
$$+ 1/(1 + e^{-\gamma_1+\gamma_2})\alpha(2-\beta_1-\beta_2) + 1/(1 + e^{-\gamma_1-\gamma_2})\alpha(\beta_1+\beta_2) = 0$$

Solving the above equations, we have,

$$\gamma_1^\infty(t_1) = \frac{1}{2}\log(G_m G_b) \quad \gamma_2^\infty(t_1) = \frac{1}{2}\log(G_m/G_b)$$

where we denote $\gamma_1^\infty(t_1) \triangleq \lim_{n \to \infty} \gamma_1(t_1)$ and $\gamma_2^\infty(t_1) \triangleq \lim_{n \to \infty} \gamma_1(t_2)$, $G_m = ((1 - A) + \sqrt{(A-1)^2 + 4A})/(2A)$ and $G_b = ((1 - B) + \sqrt{(B-1)^2 + 4B})/(2B)$, with $A = \alpha(\beta_1 + \beta_2)/((1 - \alpha)(2 - \beta_1 - \beta_2))$ and $B = \alpha(2 - \beta_1 - \beta_2)/((1 - \alpha) * (\beta_1 + \beta_2))$.

By the convexity of function $f(x) = e^x$, with a constant $C$, we have:

$$|\gamma_1 - \gamma_1^\infty| < \left| e^{\gamma_1} - e^{\gamma_1^\infty} \right| \leq C \left| 1/(1 + e^{\gamma_1}) - 1/(1 + e^{\gamma_1^\infty}) \right| \leq \sqrt{\frac{4 \log(1/\delta)}{n}},$$

$$|\gamma_2 - \gamma_2^\infty| < \left| e^{\gamma_2} - e^{\gamma_2^\infty} \right| \leq C \left| 1/(1 + e^{\gamma_2}) - 1/(1 + e^{\gamma_2^\infty}) \right| \leq \sqrt{\frac{4 \log(1/\delta)}{n}}.$$

Then we know that,

$$C_{\text{IRMv1}}^1 = \frac{1}{n_1} \sum_{i=1}^{n_1} \ell_i' \hat{y}_i^1 y_i^1 = \gamma_1 A_1^1 + \gamma_2 B_1^1$$

$$C_{\text{IRMv1}}^2 = \frac{1}{n_2} \sum_{i=1}^{n_2} \ell_i' \hat{y}_i^2 y_i^2 = \gamma_1 A_1^2 + \gamma_2 B_1^2$$

Therefore, we have that:

$$C_{\text{IRMv1}}^1 + C_{\text{IRMv1}}^2 = 0$$

Then the evolution of coefficients reduces to

$$\gamma_{j,r,1}(t+1) = \gamma_{j,r,1}(t) - \frac{\eta}{m} \cdot \sum_{e \in \mathcal{E}_{\text{tr}}} (1 + 2\lambda C_{\text{IRMv1}}^e(t)) A_1^e(t) - \frac{\eta\lambda}{m} \cdot \sum_{e \in \mathcal{E}_{\text{tr}}} 2C_{\text{IRMv1}}^e A_2^e(t)$$

$$\gamma_{j,r,2}(t+1) = \gamma_{j,r,2}(t) - \frac{\eta}{m} \cdot \sum_{e \in \mathcal{E}_{\text{tr}}} (1 + 2\lambda C_{\text{IRMv1}}^e(t)) B_1^e(t) - \frac{\eta\lambda}{m} \cdot \sum_{e \in \mathcal{E}_{\text{tr}}} 2C_{\text{IRMv1}}^e B_2^e(t)$$

Taking the solution of $\gamma_{j,r,1}(t_1)$, $\gamma_{j,r,2}(t_1)$ and value of $\alpha, \beta_1, \beta_2$, we arrive at the conclusion that with a high a probability at least $1 - \delta$ and $n > C_1 \log(1/\delta)$ with $C_1$ being a positive constant, we have:

$$\gamma_{j,r,1}(t_1 + 1) > \gamma_{j,r,1}(t_1),$$
$$\gamma_{j,r,2}(t_1 + 1) < \gamma_{j,r,2}(t_1).$$

$\square$

### D.5  Proof for Corollary 4.4

**Corollary D.10** (Restatement of Corollary 4.4). *Consider training the CNN model with the data generated from Def. 3.1, suppose that $\psi(x) = x$, $\gamma_{j,r,1}(t_1) = o(1)$, and $\gamma_{j,r,2}(t_1) = \Theta(1)$ at the end of ERM pre-train $t_1$ and $\mathcal{E}_{tr} = \{(0.25, 0.1), (0.25, 0.2)\}$. Suppose that $\delta > 0$, and $n > C \log(1/\delta)$, with $C$ being a positive constant, then with a high probability at least $1 - \delta$, we have*

$$\gamma_{j,r,1}(t_1 + 1) < \gamma_{j,r,1}(t_1).$$

*Proof of Corollary D.10.* Recall that the feature learning update rule:

$$\gamma_{j,r,1}(t+1) = \gamma_{j,r,1}(t) - \frac{\eta}{m} \cdot \sum_{e \in \mathcal{E}_{\text{tr}}} (1 + 2\lambda C_{\text{IRMv1}}^e(t)) \frac{1}{n_e} \sum_{i=1}^{n_e} \ell_i'(t) \text{Rad}(\alpha)_i$$

$$- \frac{\eta\lambda}{m} \cdot \sum_{e \in \mathcal{E}_{\text{tr}}} 2C_{\text{IRMv1}}^e \frac{1}{n_e} \sum_{i=1}^{n_e} \ell_i'' \hat{y}_i^e \cdot y_i^e \text{Rad}(\alpha)_i,$$

$$\gamma_{j,r,2}(t+1) = \gamma_{j,r,2}(t) - \frac{\eta}{m} \cdot \sum_{e \in \mathcal{E}_{\text{tr}}} (1 + 2\lambda C_{\text{IRMv1}}^e(t)) \frac{1}{n_e} \sum_{i=1}^{n_e} \ell_i'(t) \text{Rad}(\beta_e)$$

$$- \frac{\eta\lambda}{m} \cdot \sum_{e \in \mathcal{E}_{\text{tr}}} 2C_{\text{IRMv1}}^e \frac{1}{n_e} \sum_{i=1}^{n_e} \ell_i'' \hat{y}_i^e \cdot y_i^e \text{Rad}(\beta_e)_i,$$

Taking the value of $\gamma_{j,r,1}(t_1)$, $\gamma_{j,r,2}(t_1)$ and, we can conclude that:

$$
\begin{aligned}
\lim_{n\to\infty} A_1^1(t_1) &= -1/(1+e^{\gamma_2})(1-\alpha)(1-\beta_1) - 1/(1+e^{-\gamma_2})(1-\alpha)\beta_1 + \\
&\quad 1/(1+e^{\gamma_2})\alpha(1-\beta_1) + 1/(1+e^{-\gamma_2})\alpha\beta_1 \\
&= 1/(1+e^{\gamma_2})(2\alpha-1)(1-\beta_1) + 1/(1+e^{-\gamma_2})(2\alpha-1)(\beta_1) \\
&= (2\alpha-1)[1/(1+e^{\gamma_2})(1-\beta_2) + 1/(1+e^{-\gamma_2})\beta_1)] \\
\lim_{n\to\infty} A_1^2(t_1) &= 1/(1+e^{\gamma_2})(2\alpha-1)(1-\beta_2) + 1/(1+e^{-\gamma_2})(2\alpha-1)(\beta_2) \\
&= (2\alpha-1)[1/(1+e^{\gamma_2})(1-\beta_2) + 1/(1+e^{-\gamma_2})\beta_2)] \\
\lim_{n\to\infty} B_1^1(t_1) &= -1/(1+e^{\gamma_2})(1-\alpha)(1-\beta_1) + 1/(1+e^{-\gamma_2})(1-\alpha)\beta_1 - \\
&\quad 1/(1+e^{\gamma_2})\alpha(1-\beta_1) + 1/(1+e^{-\gamma_2})\alpha\beta_1 \\
&= -1/(1+e^{\gamma_2})(1-\beta_1) + 1/(1+e^{-\gamma_2})\beta_1 \\
\lim_{n\to\infty} B_1^2(t_1) &= -1/(1+e^{\gamma_2})(1-\alpha)(1-\beta_2) + 1/(1+e^{-\gamma_2})(1-\alpha)\beta_2 - \\
&\quad 1/(1+e^{\gamma_2})\alpha(1-\beta_2) + 1/(1+e^{-\gamma_2})\alpha\beta_2 \\
&= -1/(1+e^{\gamma_2})(1-\beta_2) + 1/(1+e^{-\gamma_2})\beta_2
\end{aligned}
$$

On the other hand,

$$
\begin{aligned}
\lim_{n\to\infty} A_2^1(t_1) &= e^{\gamma_2}/(1+e^{\gamma_2})^2(1-\alpha)(1-\beta_1)(\gamma_2) + e^{-\gamma_2}/(1+e^{-\gamma_2})^2(1-\alpha)\beta_1(-\gamma_2) \\
&\quad + e^{+\gamma_2}/(1+e^{\gamma_2})^2\alpha(1-\beta_1)(-\gamma_2) + e^{-\gamma_2}/(1+e^{-\gamma_2})^2\alpha\beta_1(\gamma_2) \\
&= e^{\gamma_2}/(1+e^{\gamma_2})^2(1-2\alpha)(1-\beta_1) + e^{-\gamma_2}/(1+e^{-\gamma_2})^2(2\alpha-1)\beta_1\gamma_2 \\
\lim_{n\to\infty} A_2^2(t_1) &= e^{\gamma_2}/(1+e^{\gamma_2})^2(1-\alpha)(1-\beta_2)(\gamma_2) + e^{-\gamma_2}/(1+e^{-\gamma_2})^2(1-\alpha)\beta_2(-\gamma_2) \\
&\quad + e^{\gamma_2}/(1+e^{\gamma_2})^2\alpha(1-\beta_2)(-\gamma_2) + e^{-\gamma_2}/(1+e^{-\gamma_2})^2\alpha\beta_2(\gamma_2) \\
&= e^{\gamma_2}/(1+e^{\gamma_2})^2(1-2\alpha)(1-\beta_2) + e^{-\gamma_2}/(1+e^{-\gamma_2})^2(2\alpha-1)\beta_2\gamma_2 \\
\lim_{n\to\infty} B_2^1(t_1) &= e^{\gamma_2}/(1+e^{\gamma_2})^2(1-\alpha)(1-\beta_1)(\gamma_2) + e^{-\gamma_2}/(1+e^{-\gamma_2})^2(1-\alpha)\beta_1(\gamma_2) \\
&\quad + e^{\gamma_2}/(1+e^{\gamma_2})^2\alpha(1-\beta_1)(\gamma_2) + e^{-\gamma_2}/(1+e^{-\gamma_2})^2\alpha\beta_1(\gamma_2), \\
\lim_{n\to\infty} B_2^2(t_1) &= e^{\gamma_2}/(1+e^{\gamma_2})^2(1-\alpha)(1-\beta_2)(\gamma_2) + e^{-\gamma_2}/(1+e^{-\gamma_2})^2(1-\alpha)\beta_2(\gamma_2) \\
&\quad + e^{\gamma_2}/(1+e^{\gamma_2})^2\alpha(1-\beta_2)(\gamma_2) + e^{-\gamma_2}/(1+e^{-\gamma_2})^2\alpha\beta_2(\gamma_2).
\end{aligned}
$$

Finally, taking the value of environment of $(\alpha, \beta_1, \beta_2) = (0.25, 0.1, 0.2)$, we conclude that with a high a probability at least $1 - \delta$ and $n > C_1 \log(1/\delta)$ with $C_1$ being a positive constant, we have:

$$
\gamma_{j,r,1}(t_1 + 1) < \gamma_{j,r,1}(t_1).
$$

$\square$

# E More Details about iFeAT

As mentioned in Sec. 5.2 that, when the featurizer is implemented as a deep net that have a massive amount of parameters, backpropagating through Algorithm 1 can allocate too much memory for propagating with $2K - 1$ batches of data. It is common for many realistic benchmarks such as Camelyon17 and FMoW in wilds benchmark [39] that adopts a DenseNet [31] with 121 layers as the featurizer. To relieve the exceeding computational and memory overhead, we propose a lightweight version of FeAT, denoted as FeAT. Instead of storing all of historical subsets and classifiers, iFeAT iteratively use the augmentation and retention sets and historical classifier from only the last round. In contrast, previous rich feature learning algorithm [59, 83] incurs a high computational and memory overhead as the round grows. For example, in RxRx1, we have to reduce the batch size of Bonsai to allow the proceeding of rounds $\geq 3$.

We elaborate the detailed algorithmic description of iFeAT in Algorithm 2.

---

**Algorithm 2** FeAT: **Fe**ature **A**ugmented **T**raining

---

1: **Input:** Training data $\mathcal{D}_{\mathrm{tr}}$; the maximum augmentation rounds $K$; predictor $f := w \circ \varphi$; length of inner training epochs $e$; termination threshold $p$;
2: Initialize groups $G^a \leftarrow \mathcal{D}_{\mathrm{tr}}, G^r \leftarrow \{\}$;
3: **for** $k \in [1, \ldots, K]$ **do**
4:     Randomly initialize $w_k$;
5:     **for** $j \in [1, \ldots, e]$ **do**
6:         Obtain $\ell_{\mathrm{FeAT}}$ with $G$ via Eq. 7;
7:         Update $w_k, \varphi$ with $\ell_{\mathrm{FeAT}}$;
8:     **end for**
9:     `// Early Stop if` $f_k = w_k \circ \varphi$ `fails to find new features.`
10:     **if** Training accuracy of $f_k$ is smaller than $p$ **then**
11:         Set $K = k - 1$ and terminate the loop;
12:     **end if**
13:     **if** $k > 1$ **then**
14:         `// Hence it doesnot need to maintain all historical classifiers.`
15:         Update $w_k \leftarrow (w_{k-1}, w_k)$;
16:     **end if**
17:     Split $\mathcal{D}_{\mathrm{tr}}$ into groups $\mathcal{D}_k^a, \mathcal{D}_k^r$ according to $f_k$;
18:     `// Hence it doesnot need to maintain all historical subsets.`
19:     Update groups $G^a \leftarrow \{\mathcal{D}_k^a\}, G^r \leftarrow \{\mathcal{D}_k^r\}$;
20: **end for**
21: **return** $f = w \circ \varphi$;

---

# F More Details about the Experiments

In this section, we provide more details and the implementation, evaluation and hyperparameter setups in complementary to the experiments in Sec. 6.

### F.1 More details about COLOREDMNIST experiments

**Datasets.** In the controlled experiments with COLOREDMNIST, we follow the evaluation settings as previous works [4, 16, 83]. In addition to the original COLOREDMNIST with $\mathcal{E}_{\mathrm{tr}} = \{(0.25, 0.1), (0.25, 0.2)\}$ (denoted as COLOREDMNIST-025) where spurious features are better correlated with labels, we also incorporate the modified one (denoted as COLOREDMNIST-01) with $\mathcal{E}_{\mathrm{tr}} = \{(0.1, 0.2), (0.1, 0.25)\}$ where invariant features are better correlated with labels, since both cases can happen at real world.

**Architecture and optimization.** To ensure a fair comparison, we use 4-Layer MLP with a hidden dimension of 256 as the backbone model for all methods, where we take the first 3 layers as the featurizer and the last layer as the classifier, following the common practice [25, 39]. For the optimization of the models, we use the Adam [37] optimizer with a learning rate of $1e - 3$ and a

weight decay of $1e - 3$. We report the mean and standard deviation of the performances of different methods with each configuration of hyperparameters 10 times with the random seeds from 1 to 10.

**Implementation of ERM-NF and OOD objectives.** For the common pre-training protocol with ERM, our implementation follows the previous works [83]. Specifically, we first train the model with $\{0, 50, 100, 150, 200, 250\}$ epochs and then apply the OOD regularization of various objectives with a penalty weight of $\{1e1, 1e2, 1e3, 1e4, 1e5\}$. We adopt the implementations from Zhang et al. [83] for various OOD objectives, including IRMv1 [4],VREx [41],IB-IRM [1],CLOvE [76],IGA [40] and Fishr [57] Besides, we also incorporate the state-of-the-art OOD objective proposed by Chen et al. [16] that is able to resolve both COLOREDMNIST-025 and COLOREDMNIST-01.

**Evaluation of feature learning methods.** For the sake of fairness in comparison, by default, we train all feature learning methods by the same number of epochs and rounds (if applicable). For the implementation Bonsai, we strictly follow the recommended setups provided by Zhang et al. [83], [5] where we train the model with Bonsai by 2 rounds with 50 epochs for round 1, 500 epochs for round 2, and 500 epochs for the synthesize round in COLOREDMNIST-025. While in COLOREDMNIST-01, round 1 contains 150 epochs, round 2 contains 400 epochs and the synthesize round contains 500 epochs. For the implementation of FeAT, we train the model with 2 rounds of FeAT in COLOREDMNIST-025, and 3 rounds of FeAT in COLOREDMNIST-01, where each round contains 150 epochs. While for the retain penalty, we find using a fixed number of 0.01 already achieved sufficiently good performance. ERM only contains 1 round, for which we train the model with 150 epochs in COLOREDMNIST-025 as we empirically find more epochs will incur severe performance degeneration in COLOREDMNIST-025. While in COLOREDMNIST-01, we train the model with ERM by 500 epochs to match up the overall training epochs of FeAT and Bonsai. We provide a detailed distribution of the number of epochs in each round in Table 5. It can be found

Table 5: Number of epochs in each round of various feature learning algorithms.

| CMNIST-025 | ROUND-1 | ROUND-2 | ROUND-3 | SYN. ROUND |
|---|---|---|---|---|
| ERM | 150 | - | - | - |
| BONSAI | 50 | 150 | - | 500 |
| FEAT | 150 | 150 | - | - |
| CMNIST-01 | ROUND-1 | ROUND-2 | ROUND-3 | SYN. ROUND |
| ERM | 500 | - | - | - |
| BONSAI | 150 | 400 | - | 500 |
| FEAT | 150 | 150 | 150 | - |

that, although Bonsai costs $2 - 3$ times of training epochs more than ERM and FeAT, Bonsai does not necessarily find better feature representations for OOD training, as demonstrated in Table. 1. In contrast, FeAT significantly and consistently learns richer features given both COLOREDMNIST-025 and COLOREDMNIST-01 than ERM, which shows the superiority of FeAT.

**The termination check in FeAT.** A key difference between FeAT and previous rich feature learning algorithms is that FeAT is able to perform the automatic termination check and learn the desired features stably. As elaborated in Sec. 5.2, FeAT can terminate automatically by inspecting the retention accuracy. To verify, we list the FeAT performances in various subsets of COLOREDMNIST-025 and COLOREDMNIST-01 at different rounds. We use a termination accuracy of $130\%$, which trades off the exploration (i.e., training accuracy as $80\%$) and the retention (i.e., retention accuracy as $50\%$) properly. As shown in Table 6, in COLOREDMNIST-025 (COLOREDMNIST-01), after FeAT learns sufficiently good features at Round 2 (3), respectively, it is not necessary to proceed with Round 3 (4) as it will destroy the already learned features and lead to degenerated retention performance (i.e., the sum of training and retention accuracies is worse than $130\%$.

---

[5] https://github.com/TjuJianyu/RFC

Table 6: Performances in various sets at different FeAT rounds.

| COLOREDMNIST-025 | ROUND-1 | ROUND-2 | ROUND-3 | |
|---|---|---|---|---|
| TRAINING ACC. | 85.08± 0.14 | 71.87± 0.96 | 84.93± 1.26 | |
| RETENTION ACC. | - | 88.11± 4.28 | 43.82± 0.59 | |
| OOD ACC. | 11.08± 0.30 | 70.64± 0.62 | 10.07± 0.26 | |

| COLOREDMNIST-01 | ROUND-1 | ROUND-2 | ROUND-3 | ROUND-4 |
|---|---|---|---|---|
| TRAINING ACC. | 88.63± 0.15 | 74.25± 1.23 | 86.07± 0.36 | 77.29± 0.24 |
| RETENTION ACC. | - | 85.91± 1.78 | 48.05± 1.39 | 29.09± 1.15 |
| OOD ACC. | 73.50± 0.41 | 17.32± 2.69 | 85.40± 0.54 | 12.48± 2.85 |

## F.2 More details about WILDS experiments

In this section, we provide more details about the WILDS datasets used in the experiments as well as the evaluation setups.

### F.2.1 Dataset description.

To evaluate the feature learning performance given data from realistic scenarios, we select 6 challenging datasets from WILDS [39] benchmark. The datasets contain various realistic distribution shifts, ranging from domain distribution shifts, subpopulation shifts and the their mixed. A summary of the basic information and statistics of the selected WILDS datasets can be found in Table. 7, Table. 8, respectively. In the following, we will give a brief introduction to each of the datasets. More details can be found in the WILDS paper [39].

Table 7: A summary of datasets information from WILDS.

| Dataset | Data ($x$) | Class information | Domains | Metric | Architecture |
|---|---|---|---|---|---|
| AMAZON | Product reviews | Star ratings (5 classes) | 7,676 reviewers | 10-eth percentile acc. | DistillBERT |
| CAMELYON17 | Tissue slides | Tumor (2 classes) | 5 hospitals | Avg. acc. | DenseNet-121 |
| CIVILCOMMENTS | Online comments | Toxicity (2 classes) | 8 demographic groups | Wr. group acc. | DistillBERT |
| FMOW | Satellite images | Land use (62 classes) | 16 years x 5 regions | Wr. group acc. | DenseNet-121 |
| IWILDCAM | Photos | Animal species (186 classes) | 324 locations | Macro F1 | ResNet-50 |
| RXRX1 | Cell images | Genetic treatments (1,139 classes) | 51 experimental batches | Avg. acc | ResNet-50 |

Table 8: A summary of datasets statistics from WILDS.

| Dataset | # Examples | | | # Domains | | |
|---|---|---|---|---|---|---|
| | train | val | test | train | val | test |
| AMAZON | 1,000,124 | 100,050 | 100,050 | 5,008 | 1,334 | 1,334 |
| CAMELYON17 | 302,436 | 34,904 | 85,054 | 3 | 1 | 1 |
| CIVILCOMMENTS | 269,038 | 45,180 | 133,782 | - | - | - |
| FMOW | 76,863 | 19,915 | 22,108 | 11 | 3 | 2 |
| IWILDCAM | 129,809 | 14,961 | 42,791 | 243 | 32 | 48 |
| RXRX1 | 40,612 | 9,854 | 34,432 | 33 | 4 | 14 |

**Amazon.** We follow the WILDS splits and data processing pipeline for the Amazon dataset [51]. It provides $1.4$ million comments collected from $7,676$ Amazon customers. The task is to predict the score (1-5 stars) for each review. The domains $d$ are defined according to the reviewer/customer who wrote the product reviews. The evaluation metric used for the task is 10th percentile of per-user accuracies in the OOD test sets, and the backbone model is a DistilBert [66], following the WILDS protocol [39].

**Camelyon17.** We follow the WILDS splits and data processing pipeline for the Camelyon17 dataset [6]. It provides $450,000$ lymph-node scans from 5 hospitals. The task in Camelyon17 is to take the input of $96 \times 96$ medical images to predict whether there exists a tumor tissue in the image. The domains $d$ refers to the index of the hospital where the image was taken. The training data are sampled from the first 3 hospitals where the OOD validation and test data are sampled from the $4$-th and $5$-th hospital, respectively. We will use the average accuracy as the evaluation metric and a DenseNet-121 [31] as the backbone for the featurizer.

**CivilComments.** We follow the WILDS splits and data processing pipeline for the CivilComments dataset [9]. It provides $450,000$ comments collected from online articles. The task is to classify whether an online comment text is toxic or non-toxic. The domains $d$ are defined according to the demographic features, including male, female, LGBTQ, Christian, Muslim, other religions, Black, and White. CivilComments is used to study the subpopulation shifts, here we will use the worst group/domain accuracy as the evaluation metric. As for the backbone of the featurizer, we will use a DistillBert [66] following WILDS [39].

**FMoW.** We follow the WILDS splits and data processing pipeline for the FMoW dataset [18]. It provides satellite images from 16 years and 5 regions. The task in FMoW is to classify the images into 62 classes of building or land use categories. The domain is split according to the year that the satellite image was collected, as well as the regions in the image which could be Africa, America, Asia, Europe or Oceania. Distribution shifts could happen across different years and regions. The training data contains data collected before 2013, while the validation data contains images collected within 2013 to 2015, and the test data contains images collected after 2015. The evaluation metric for FMoW is the worst region accuracy and the backbone model for the featurizer is a DenseNet-121 [31].

**iWildCam.** We follow the WILDS splits and data processing pipeline for the iWildCam dataset [8]. It is consist of $203,029$ heat or motion-activated photos of animal specifies from 323 different camera traps across different countries around the world. The task of iWildCam is to classify the corresponding animal specifies in the photos. The domains is split according to the locations of the camera traps which could introduce the distribution shifts. We will use the Macro F1 as the evaluation metric and a ResNet-50 [27] as the backbone for the featurizer.

**RxRx1.** We follow the WILDS splits and data processing pipeline for the RxRx1 dataset [72]. The input is an image of cells taken by fluorescent microscopy. The cells can be genetically perturbed by siRNA and the task of RxRx1 is to predict the class of the corresponding siRNA that have treated the cells. There exists $1,139$ genetic treatments and the domain shifts are introduced by the experimental batches. We will use the average accuracy of the OOD experimental batches as the evaluation metric and a ResNet-50 [27] as the backbone for the featurizer.

### F.2.2 Training and evaluation details.

We follow previous works to implement and evaluate different methods used in our experiments [39]. The information of the referred paper and code is listed as in Table. 9.

Table 9: The information of the referred paper and code.

| Paper | Commit | Code |
|---|---|---|
| WILDS [39] | v2.0.0 | https://wilds.stanford.edu/ |
| Fish [69] | 333efa24572d99da0a4107ab9cc4af93a915d2a9 | https://github.com/YugeTen/fish |
| Bonsai [83] | 33b9ecad0ce8b3462793a2da7a9348d053c06ce0 | https://github.com/TjuJianyu/RFC |
| DFR [33, 38] | 6d098440c697a1175de6a24d7a46ddf91786804c | https://github.com/izmailovpavel/spurious_feature_learning |

The general hyperparemter setting inherit from the referred codes and papers, and are as listed in Table 10. We use the same backbone models to implement the featurizer [27, 31, 66]. By default, we repeat the experiments by 3 runs with the random seeds of $0, 1, 2$. While for Camelyon17, we follow the official guide to repeat 10 times with the random seeds from 0 to 9.

Table 10: General hyperparameter settings for the experiments on WILDS.

| Dataset | AMAZON | CAMELYON17 | CIVILCOMMENTS | FMoW | iWILDCAM | RxRx1 |
|---|---|---|---|---|---|---|
| Num. of seeds | 3 | 10 | 3 | 3 | 3 | 3 |
| Learning rate | 2e-6 | 1e-4 | 1e-5 | 1e-4 | 1e-4 | 1e-3 |
| Weight decay | 0 | 0 | 0.01 | 0 | 0 | 1e-5 |
| Scheduler | n/a | n/a | n/a | n/a | n/a | Cosine Warmup |
| Batch size | 64 | 32 | 16 | 32 | 16 | 72 |
| Architecture | DistilBert | DenseNet121 | DistilBert | DenseNet121 | ResNet50 | ResNet50 |
| Optimizer | Adam | SGD | Adam | Adam | Adam | Adam |
| Domains in minibatch | 5 | 3 | 5 | 5 | 10 | 10 |
| Group by | Countries | Hospitals | Demographics× toxicity | Times × regions | Trap locations | Experimental batches |
| Training epochs | 200 | 10 | 5 | 12 | 9 | 90 |

**OOD objective implementations.** We choose 4 representative OOD objectives to evaluate the quality of learned features, including GroupDRO [64], IRMv1 [4], VREx [41] and IRMX [16].

We implement the OOD objectives based on the code provided by Shi et al. [69]. For each OOD objective, by default, we follow the WILDS practice to sweep the penalty weights from the range of $\{1e-2, 1e-1, 1, 1e1, 1e2\}$, and perform the model and hyperparameter selection via the performance in the provided OOD validation set of each dataset. Due to the overwhelming computational overhead required by large datasets and resource constraints, we tune the penalty weight in iWildCam according to the performance with seed 0, which we empirically find yields similar results as full seed tunning. Besides in Amazon, we adopt the penalty weights tuned from CivilComments since the two datasets share a relatively high similarity, which we empirically find yields similar results as full seed tunning, too. On the other hand, it raises more challenges for feature learning algorithms in iWildCam and Amazon.

**Deep Feature Reweighting (DFR) implementations.** For the implementation of DFR [33, 38], we use the code provided in Izmailov et al. [33]. By default, DFR considers the OOD validation as an unbiased dataset and adopts the OOD validation set to learn a new classifier based on the frozen features from the pre-trained featurizer. We follow the same implementation and evaluation protocol when evaluating feature learning quality in FMoW and CivilComments. However, since Camelyon17 does not have the desired OOD validation set, we follow the "cheating" protocol as in Rosenfeld et al. [63] to perform the logistic regression based the train and test sets. Note that when "cheating", the model is not able to access the whole test sets. Instead, the logistic regression is conducted on a random split of the concatenated train and test data. Moreover, for Amazon and iWildCam, we find the original implementation fails to converge possibly due to the complexity of the task, and the relatively poor feature learning quality. Hence we implement a new logistic regression based on PyTorch [54] optimized with SGD, and perform DFR using "cheating" protocol based on the OOD validation set and test set. Besides, we find neither the two aforementioned implementations or dataset choices can lead to DFR convergence in RxRx, which we will leave for future investigations.

**Feature learning algorithm implementations.** We implement all the feature learning methods based on the Fish code framework. For the fairness of comparison, we set all the methods to train the same number of steps or rounds (if applicable) in WILDS datasets. The only exception is in RxRx1, where both Bonsai and FeAT require more steps to converge, since the initialized featurizer has a relatively large distance from the desired featurizer in the task. We did not train the model for much too long epochs as Izmailov et al. [33] find that it only requires $2 - 5$ epochs for deep nets to learn high-quality invariant features. The final model is selected based on the OOD validation accuracy during the training. Besides, we tune the retain penalty in FeAT by searching over $\{1e - 2, 1e - 1, 0.5, 1, 2, 10\}$, and finalize the retain penalty according to the OOD validation performance. We list the detailed training steps and rounds setups, as well as the used retain penalty in FeAT in Table 11.

Table 11: Hyperparameter setups of feature learning algorithms for the experiments on WILDS.

| Dataset | AMAZON | CAMELYON17 | CIVILCOMMENTS | FMOW | iWILDCAM | RxRx1 |
|---|---|---|---|---|---|---|
| Overall steps | 31,000 | 10,000 | 50,445 | 9,600 | 48,000 | 20,000 |
| Approx. epochs | 4 | 10 | 3 | 4 | 10 | 10 |
| Num. of rounds | 3 | 2 | 3 | 2 | 2 | 10 |
| Steps per round | 10,334 | 5,000 | 16,815 | 4,800 | 10 | 10 |
| FeAT Retain penalty | 2.0 | 1e-2 | 1e-2 | 1.0 | 0.5 | 10 |

For ERM, we train the model simply by the overall number of steps, except for RxRx1 where we train the model by $15,000$ steps following previous setups [69]. Bonsai and FeAT directly adopt the setting listed in the Table 11. Besides, Bonsai will adopt one additional round for synthesizing the pre-trained models from different rounds. Although Zhang et al. [83] requires Bonsai to train the two rounds for synthesizing the learned features, we empirically find additional training steps in synthesizing will incur overfitting and worse performance. Moreover, as Bonsai requires propagating $2K - 1$ batches of the data that may exceed the memory limits, we use a smaller batch size when training Bonsai in iWildCam (8) and RxRx1 (56).

### F.3 Software and hardware

We implement our methods with PyTorch [54]. For the software and hardware configurations, we ensure the consistent environments for each datasets. We run all the experiments on Linux servers with NVIDIA V100 graphics cards with CUDA 10.2.

### F.4 Computational analysis

Compared to ERM, the additional computational and memory overhead introduced in FeAT mainly lie in the FeAT training and partitioning. At each training step, FeAT needs $(k-1)$ additional forward and backward propagation, the same as Bonsai, while FeAT only needs $\min(1, k-1)$ additional propagation. Besides, Bonsai additionally requires another round of training with $(K-1)$ additional propagation, given $K$ total rounds.

We calculated the computational overhead: The results aligned with our discussion. Bonsai requires

Table 12: Training and memory overhead of different algorithms.

|  | Camelyon17 Training time | Memory (%) | CivilComments Training time | Memory (%) |
|---|---|---|---|---|
| ERM | 56.21±8.29 mins | 22.56±0.00 | 24.22±0.33 hrs | 36.46±0.00 |
| Bonsai | 214.55±1.13 mins | 51.75±0.01 | 58.47±0.91 hrs | 64.43±0.31 |
| FeAT | 101.14±12.79 mins | 51.92±0.04 | 28.19±1.15 hrs | 56.21±0.48 |

much more time for the additional synthetic round and much more memory when there are 3 or more rounds. In contrast, FeAT achieves the best performance without introducing too much additional computational overhead.

### F.5 Feature learning analysis

We first visualize the feature learning of ERM and FeAT on ColoredMNIST-025, as shown in Fig. 4 It can be found that ERM can learn both invariant and spurious features to predict the label, aligned with our theory.

However, ERM focuses more on spurious features and even forgets certain features with longer training epochs, which could be due to multiple reasons such as the simplicity biases of ERM. Hence predictions based on ERM learned features fail to generalize to OOD examples. In contrast, FeAT effectively captures the meaningful features for all samples and generalizes to OOD examples well.

Table 13: Labels and predictions for the visualized samples.

|  | Label | ERM | Bonsai | FeAT |  | Label | ERM | Bonsai | FeAT |
|---|---|---|---|---|---|---|---|---|---|
| CAMELYON17 | 1 | 1 | 0 | 1 | IWILDCAM | 113 | 68 | 0 | 113 |
|  | 1 | 1 | 0 | 1 |  | 113 | 0 | 0 | 113 |
|  | 1 | 1 | 0 | 1 |  | 36 | 36 | 36 | 36 |
|  | 1 | 0 | 0 | 0 |  | 36 | 36 | 36 | 36 |
| FMOW | 40 | 40 | 40 | 40 | RXRX1 | 1138 | 812 | 812 | 812 |
|  | 40 | 40 | 40 | 40 |  | 1138 | 1133 | 1125 | 1133 |
|  | 40 | 2 | 29 | 29 |  | 35 | 43 | 1119 | 143 |
|  | 40 | 40 | 40 | 40 |  | 35 | 35 | 1054 | 35 |
| CIVILCOMMENTS | toxic | toxic | toxic | toxic | AMAZON | 2 | 3 | 3 | 2 |
|  | toxic | toxic | toxic | toxic |  | 5 | 5 | 5 | 5 |
|  | toxic | toxic | toxic | toxic |  | 3 | 4 | 4 | 4 |
|  | nontoxic | nontoxic | nontoxic | nontoxic |  | 5 | 5 | 5 | 5 |

We also visualize the saliency maps of ERM, Bonsai, and FeAT on all real-world datasets used in our work with https://github.com/pytorch/captum. The visualizations are shown as in Fig. 5 to Fig. 11, for which the labels and the predictions of different algorithms are given in Table. 13. It can be found that, across various tasks and data modalities, FeAT effectively learns more meaningful and diverse features than ERM and Bonsai, which serve as strong evidence for the consistent superiority of FeAT in OOD generalization.

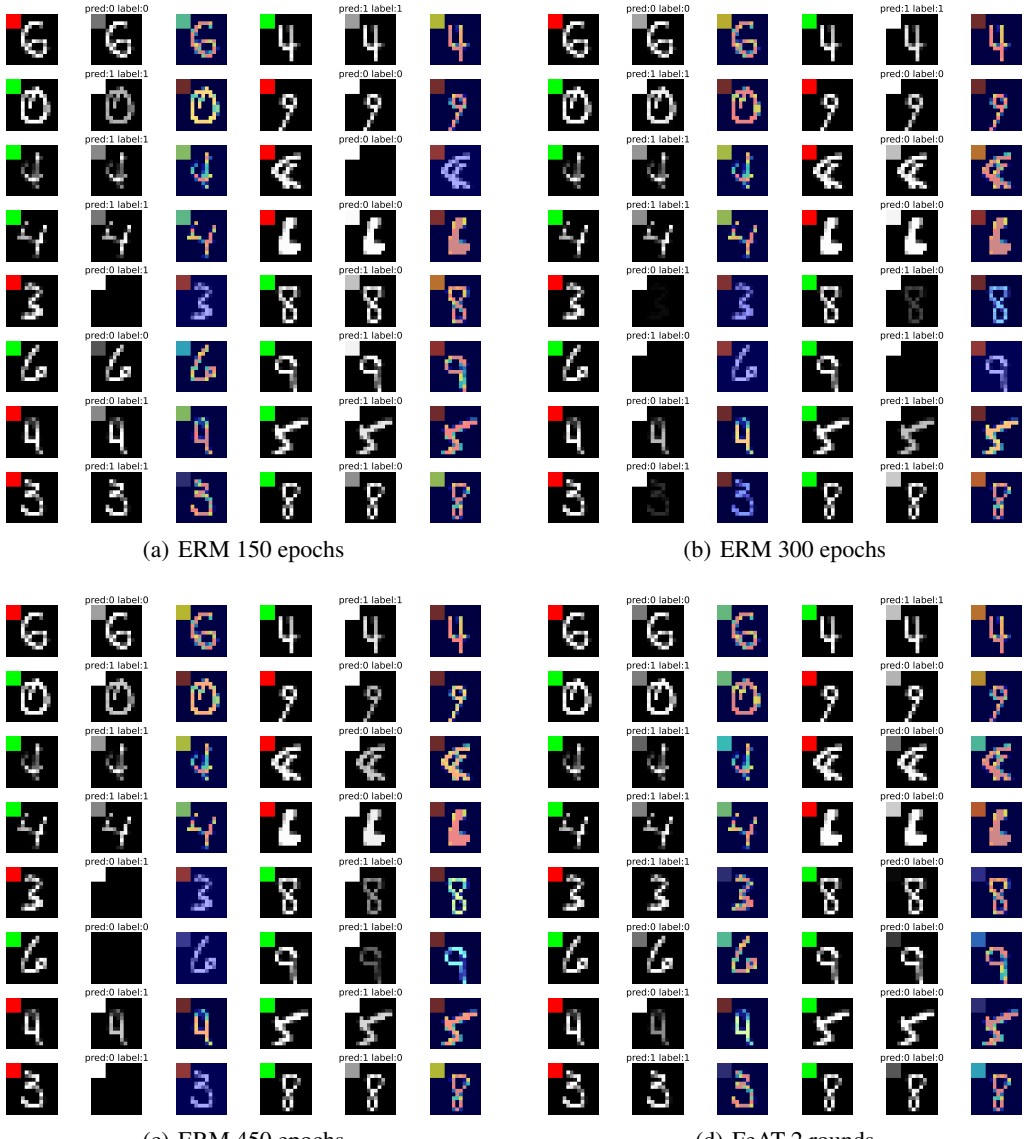

(a) ERM 150 epochs

(b) ERM 300 epochs

(c) ERM 450 epochs

(d) FeAT 2 rounds

Figure 4: GradCAM visualization on COLOREDMNIST-025, where the shortcuts are now concentrated to a colored path at the up left. Three visualizations are drawn for each sample: the original figure, the gray-colored gradcam, and the gradcam. It can be found that ERM can not properly capture the desired features or even forget certain features with longer training epochs. FeAT can stably capture the desired features.

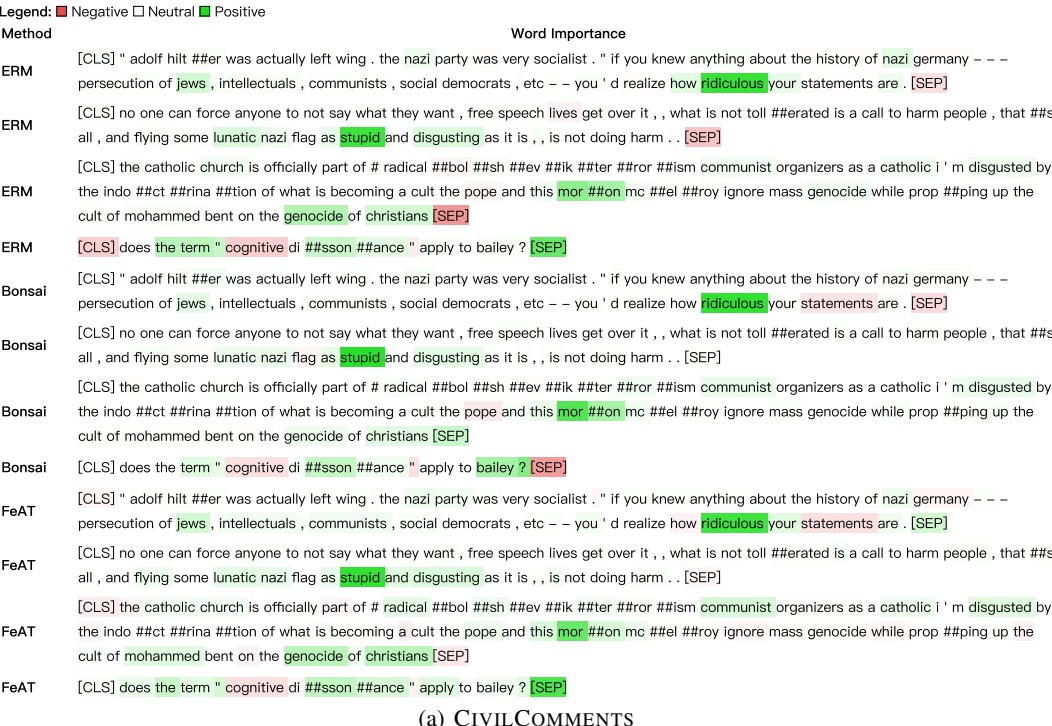

(a) CIVILCOMMENTS

Figure 5: Saliency map of feature learning on WILDS CIVILCOMMENTS benchmark. The green-colored tokens are the learned features that contributed most to the target class, while the red-colored tokens contributed to the other classes. It can be found that FeAT is able to learn more meaningful and diverse features than ERM and Bonsai.

| Method | Word Importance |
|---|---|
| ERM | [CLS] american dream ##z try ##s to be american idol with a political agenda on the bush ad ##mins ##tra ##tion which in itself goes all over the place & never provides entertainment . hugh grant is a actor who pretty much hit his peak well over a decade ago & his take as simon from idol is rather just there . mandy moore is ok nothing to say more than she may be a bit player as a actress at best . this film takes on so much to be a satire , but it ends up with more questions then any answers . too long & over ##bl ##own . william def ##oe was trying to look like dick cheney but ended up looking more like henry fond ##a . dennis qu ##aid was channel ##ing george bush with mixed results . just a ok film . [SEP] |
| ERM | [CLS] this fits my iphone perfectly , and i already have an inc ##ip ##io silicon sleeve that i keep it in permanently . it ' s a little s ##nu ##g with the sleeve , but easy enough to put in and take out . there are 2 belt loops , that i don ' t use , and one belt clip that i do use . the belt clip seems sturdy enough . when i bought this , it was $ 2 . 98 with no s & h . this certainly isn ' t worth a whole lot more than that , but it ' s a great value for the price . i ' d actually be willing to pay up to $ 7 . 00 for it . it ' s not too bulky , at least not more than a holster should be . the magnet keeps it closed . it stays on my belt just fine , and it ' s sturdy enough to keep the iphone safe for a short drop , especially with the silicon sleeve . it ' s not terribly fashionable , but you can ' t beat the price . [SEP] |
| ERM | [CLS] the mind of till ##ie cole must be quite an interesting place . in her third book of the scarred souls series cole once again explores the deepest parts of human de ##pr ##avi ##ty . it ' ll shock and outrage you , it ' ll make you feel so much more for her characters . ra ##va ##ge is told mainly from the points of views of 194 and z ##oya with a couple chapters from lu ##ka . it is not a stand alone book , in my opinion this series should be read in order . i ' ve been a fan of this series from the beginning . the darkness of human nature till ##ie cole continually portrays never cease ##s to engage me . her stories come from a place of raw honesty and empathy . her writing mature ##s with each book . the story of 194 and z ##oya and their roles in this tangled web of a world cole has created was as consuming as the others . however , for me , something was off in their relationship , in the romance . i missed the connection i felt with the previous books main characters . while cole dug deep into their past and pulled out their flaws and vu ##ln ##era ##bilities so well as individuals , an aspect of their chemistry and need for each other just didn ' t quite click . this series consistently moves forward though adding more depth and layers to the ultimate story and goals of all of these characters . and in that aspect , the story really excelled and has me just as intrigued in how it all works out as i ever was and i very much look forward to the next installment of the scarred souls series . ra ##va ##ge left much open to be explored and i ' m confident in cole ' s story telling to take me even deeper into her interesting ##ly de ##pr ##ave ##d world . [SEP] |
| ERM | [CLS] just keeps getting better and better ! if you like alpha men , strong heroine ##s and laughter with a dash of seriousness , don ' t miss this series ! ! [SEP] |

(a) AMAZON-ERM

| Method | Word Importance |
|---|---|
| Bonsai | [CLS] american dream ##z try ##s to be american idol with a political agenda on the bush ad ##mins ##tra ##tion which in itself goes all over the place & never provides entertainment . hugh grant is a actor who pretty much hit his peak well over a decade ago & his take as simon from idol is rather just there . mandy moore is ok nothing to say more than she may be a bit player as a actress at best . this film takes on so much to be a satire , but it ends up with more questions then any answers . too long & over ##bl ##own . william def ##oe was trying to look like dick cheney but ended up looking more like henry fond ##a . dennis qu ##aid was channel ##ing george bush with mixed results . just a ok film . [SEP] |
| Bonsai | [CLS] this fits my iphone perfectly , and i already have an inc ##ip ##io silicon sleeve that i keep it in permanently . it ' s a little s ##nu ##g with the sleeve , but easy enough to put in and take out . there are 2 belt loops , that i don ' t use , and one belt clip that i do use . the belt clip seems sturdy enough . when i bought this , it was $ 2 . 98 with no s & h . this certainly isn ' t worth a whole lot more than that , but it ' s a great value for the price . i ' d actually be willing to pay up to $ 7 . 00 for it . it ' s not too bulky , at least not more than a holster should be . the magnet keeps it closed . it stays on my belt just fine , and it ' s sturdy enough to keep the iphone safe for a short drop , especially with the silicon sleeve . it ' s not terribly fashionable , but you can ' t beat the price . [SEP] |
| Bonsai | [CLS] the mind of till ##ie cole must be quite an interesting place . in her third book of the scarred souls series cole once again explores the deepest parts of human de ##pr ##avi ##ty . it ' ll shock and outrage you , it ' ll make you feel so much more for her characters . ra ##va ##ge is told mainly from the points of views of 194 and z ##oya with a couple chapters from lu ##ka . it is not a stand alone book , in my opinion this series should be read in order . i ' ve been a fan of this series from the beginning . the darkness of human nature till ##ie cole continually portrays never cease ##s to engage me . her stories come from a place of raw honesty and empathy . her writing mature ##s with each book . the story of 194 and z ##oya and their roles in this tangled web of a world cole has created was as consuming as the others . however , for me , something was off in their relationship , in the romance . i missed the connection i felt with the previous books main characters . while cole dug deep into their past and pulled out their flaws and vu ##ln ##era ##bilities so well as individuals , an aspect of their chemistry and need for each other just didn ' t quite click . this series consistently moves forward though adding more depth and layers to the ultimate story and goals of all of these characters . and in that aspect , the story really excelled and has me just as intrigued in how it all works out as i ever was and i very much look forward to the next installment of the scarred souls series . ra ##va ##ge left much open to be explored and i ' m confident in cole ' s story telling to take me even deeper into her interesting ##ly de ##pr ##ave ##d world . [SEP] |
| Bonsai | [CLS] just keeps getting better and better ! if you like alpha men , strong heroine ##s and laughter with a dash of seriousness , don ' t miss this series ! ! [SEP] |

(b) AMAZON-Bonsai

Figure 6: Saliency map of feature learning on WILDS AMAZON benchmark. The green-colored tokens are the learned features that contributed most to the target class, while the red-colored tokens contributed to the other classes. It can be found that FeAT is able to learn more meaningful and diverse features than ERM and Bonsai.

| Method | Word Importance |
|---|---|
| FeAT | [CLS] american dream ##z try ##s to be american idol with a political agenda on the bush ad ##mins ##tra ##tion which in itself goes all over the place & never provides entertainment . hugh grant is a actor who pretty much hit his peak well over a decade ago & his take as simon from idol is rather just there . mandy moore is ok nothing to say more than she may be a bit player as a actress at best . this film takes on so much to be a satire , but it ends up with more questions then any answers . too long & over ##bl ##own . william def ##oe was trying to look like dick cheney but ended up looking more like henry fond ##a . dennis qu ##aid was channel ##ing george bush with mixed results . just a ok film . [SEP] |
| FeAT | [CLS] this fits my iphone perfectly , and i already have an inc ##ip ##io silicon sleeve that i keep it in permanently . it ' s a little s ##nu ##g with the sleeve , but easy enough to put in and take out . there are 2 belt loops , that i don ' t use , and one belt clip that i do use . the belt clip seems sturdy enough . when i bought this , it was $ 2 . 98 with no s & h . this certainly isn ' t worth a whole lot more than that , but it ' s a great value for the price . i ' d actually be willing to pay up to $ 7 . 00 for it . it ' s not too bulky , at least not more than a holster should be . the magnet keeps it closed . it stays on my belt just fine , and it ' s sturdy enough to keep the iphone safe for a short drop , especially with the silicon sleeve . it ' s not terribly fashionable , but you can ' t beat the price . [SEP] |
| FeAT | [CLS] the mind of till ##ie cole must be quite an interesting place . in her third book of the scarred souls series cole once again explores the deepest parts of human de ##pr ##avi ##ty . it ' ll shock and outrage you , it ' ll make you feel so much more for her characters . ra ##va ##ge is told mainly from the points of views of 194 and z ##oya with a couple chapters from lu ##ka . it is not a stand alone book , in my opinion this series should be read in order . i ' ve been a fan of this series from the beginning . the darkness of human nature till ##ie cole continually portrays never cease ##s to engage me . her stories come from a place of raw honesty and empathy . her writing mature ##s with each book . the story of 194 and z ##oya and their roles in this tangled web of a world cole has created was as consuming as the others . however , for me , something was off in their relationship , in the romance . i missed the connection i felt with the previous books main characters . while cole dug deep into their past and pulled out their flaws and vu ##ln ##era ##bilities so well as individuals , an aspect of their chemistry and need for each other just didn ' t quite click . this series consistently moves forward though adding more depth and layers to the ultimate story and goals of all of these characters . and in that aspect , the story really excelled and has me just as intrigued in how it all works out as i ever was and i very much look forward to the next installment of the scarred souls series . ra ##va ##ge left much open to be explored and i ' m confident in cole ' s story telling to take me even deeper into her interesting ##ly de ##pr ##ave ##d world . [SEP] |
| FeAT | [CLS] just keeps getting better and better ! if you like alpha men , strong heroine ##s and laughter with a dash of seriousness , don ' t miss this series ! ! [SEP] |

(a) AMAZON-FeAT

Figure 7: Saliency map of feature learning on WILDS AMAZON benchmark (part 2). The green-colored tokens are the learned features that contributed most to the target class, while the red-colored tokens contributed to the other classes. It can be found that FeAT is able to learn more meaningful and diverse features than ERM and Bonsai.

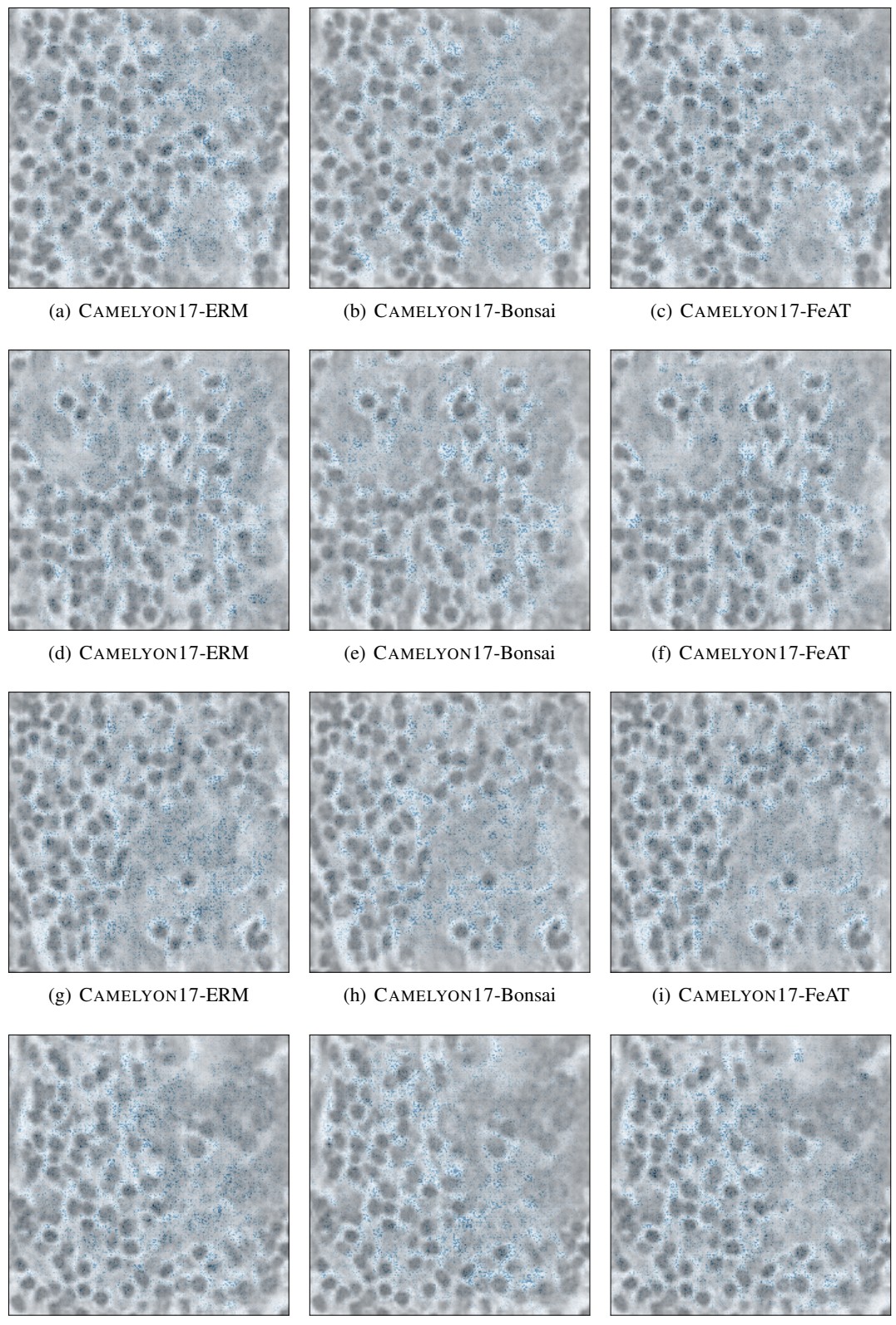

Figure 8: Saliency map of feature learning on CAMELYON17 benchmark. The blue dots are the salient features. A deeper blue color denotes more salient features. It can be found that FeAT is able to learn more meaningful and diverse features than ERM and Bonsai.

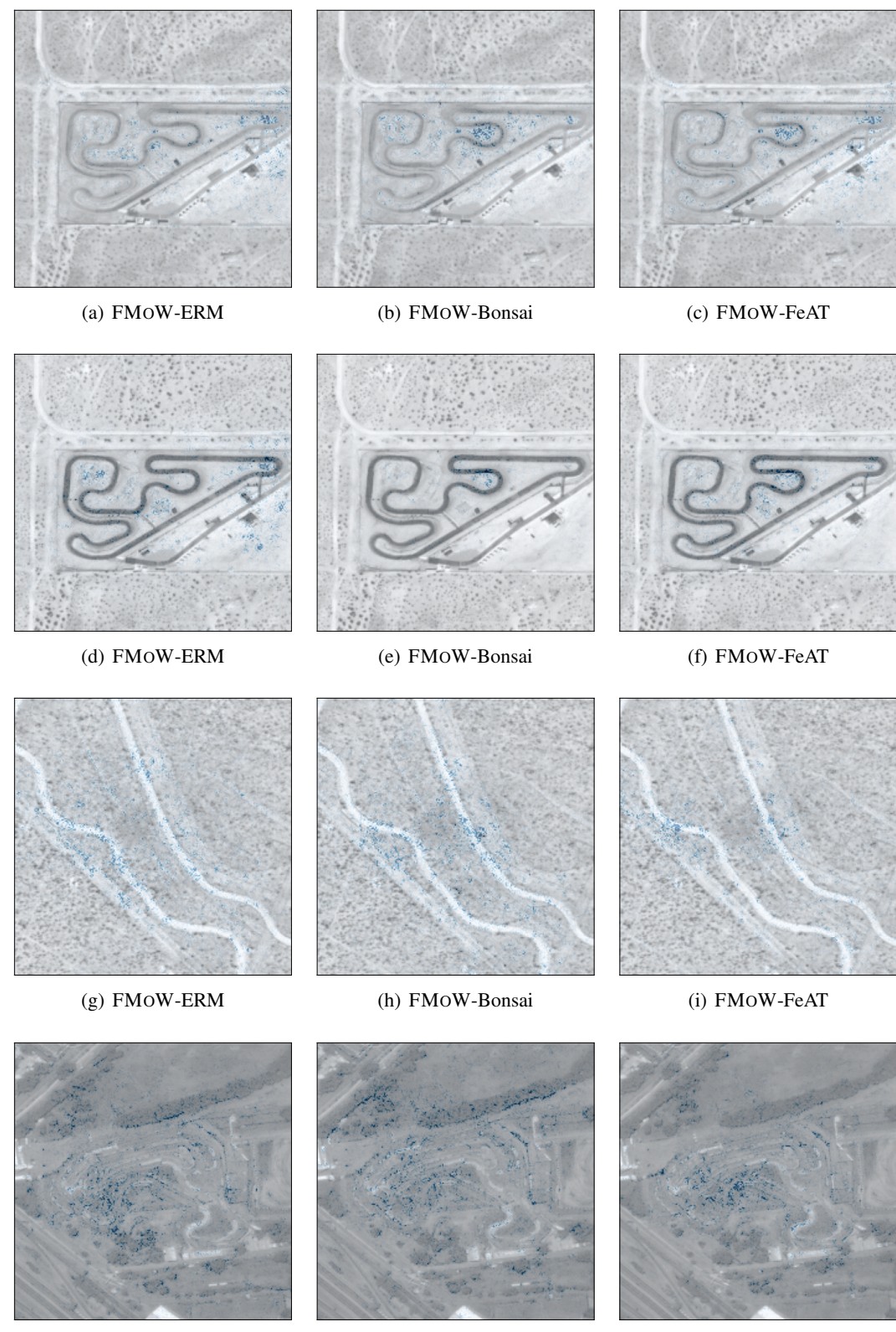

Figure 9: Saliency map of feature learning on FMoW benchmark. The blue dots are the salient features. A deeper blue color denotes more salient features. It can be found that FeAT is able to learn more meaningful and diverse features than ERM and Bonsai.

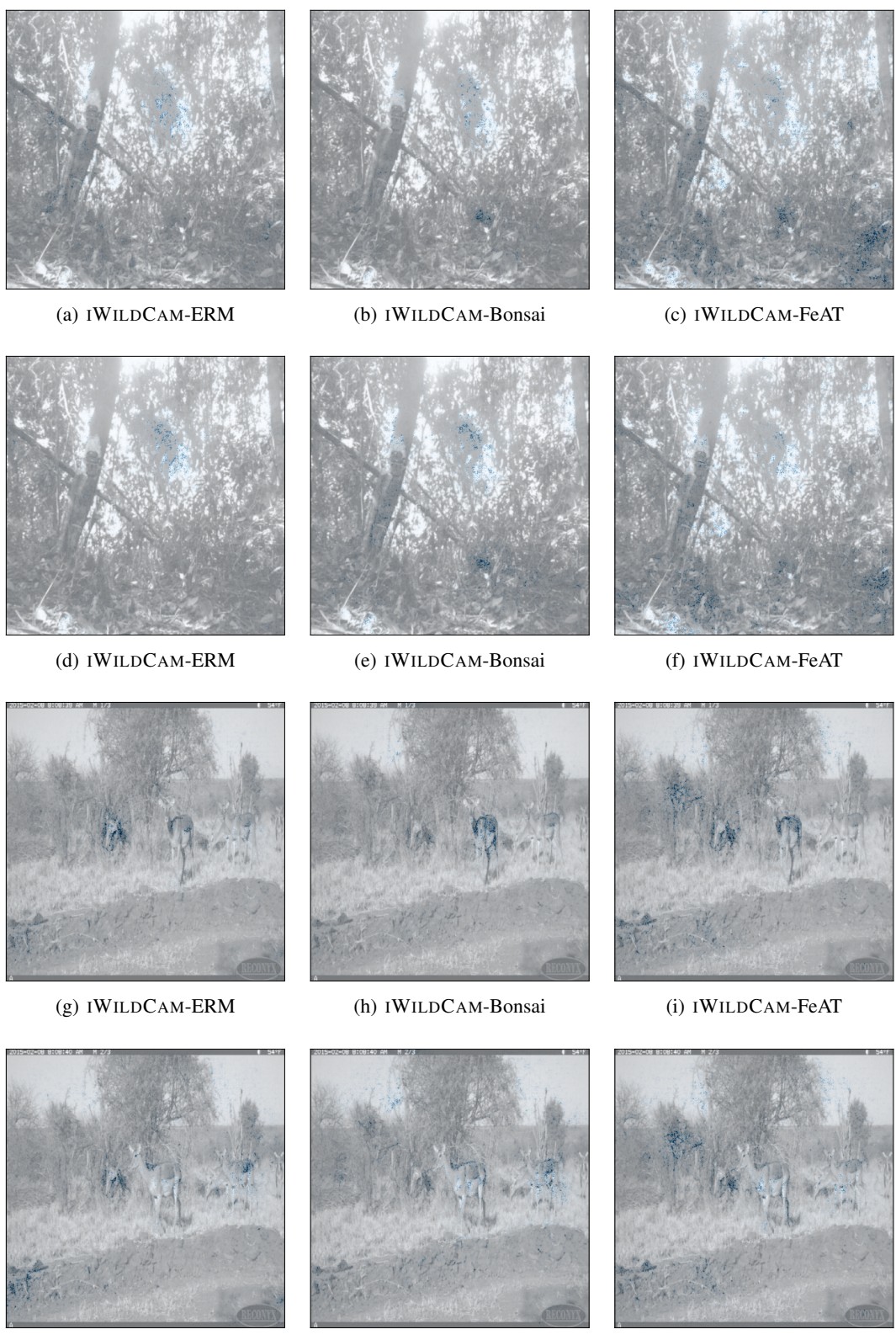

Figure 10: Saliency map of feature learning on IWILDCAM benchmark. The blue dots are the salient features. A deeper blue color denotes more salient features. It can be found that FeAT is able to learn more meaningful and diverse features than ERM and Bonsai.

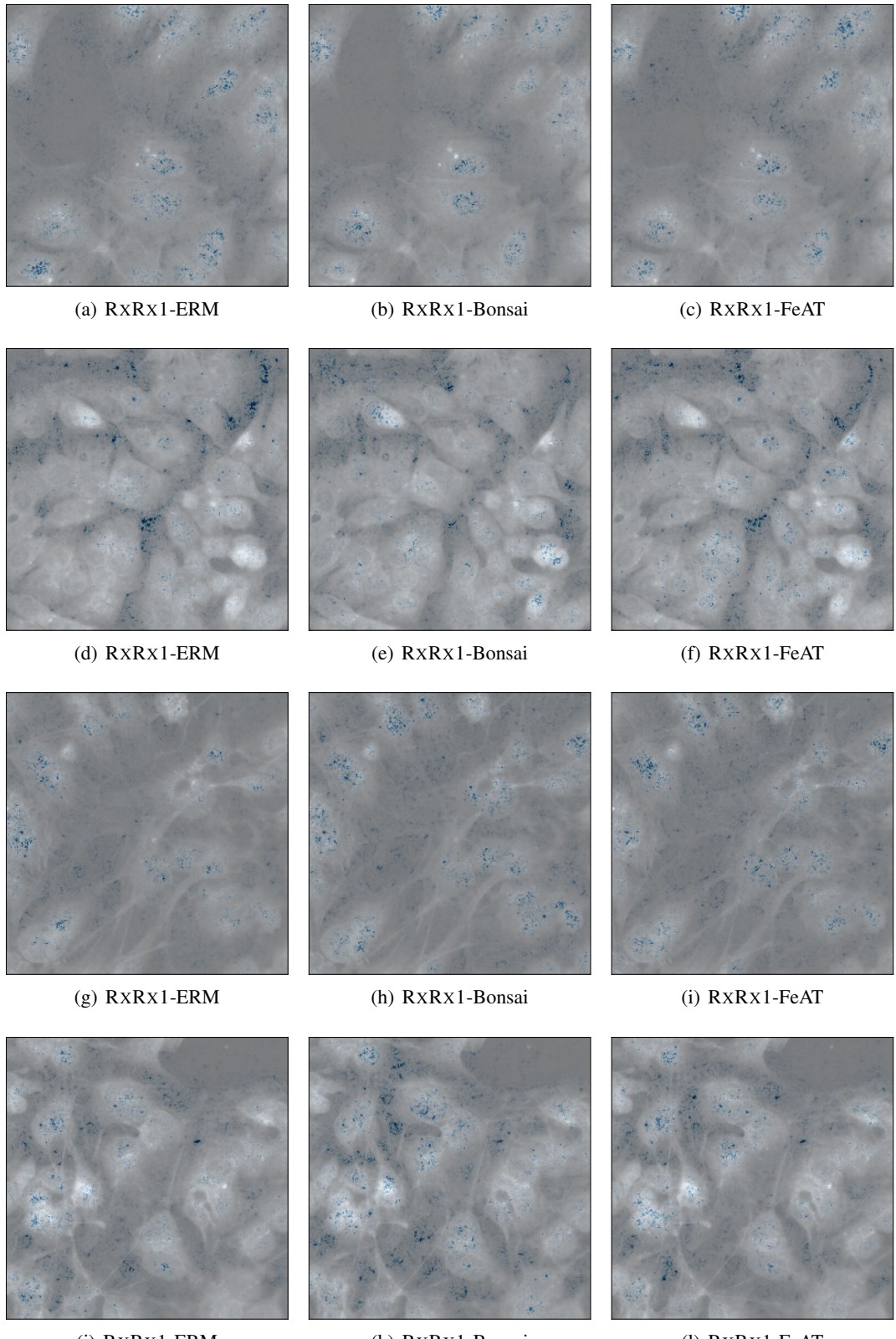

(a) RxRx1-ERM      (b) RxRx1-Bonsai      (c) RxRx1-FeAT

(d) RxRx1-ERM      (e) RxRx1-Bonsai      (f) RxRx1-FeAT

(g) RxRx1-ERM      (h) RxRx1-Bonsai      (i) RxRx1-FeAT

(j) RxRx1-ERM      (k) RxRx1-Bonsai      (l) RxRx1-FeAT

Figure 11: Saliency map of feature learning on RxRx1 benchmark. The blue dots are the salient features. A deeper blue color denotes more salient features. It can be found that FeAT is able to learn more meaningful and diverse features than ERM and Bonsai.

