# OpenReview forum: "Understanding and Improving Feature Learning for Out-of-Distribution Generalization"
_NeurIPS.cc/2023/Conference — NeurIPS 2023 poster_

### Official Review · Reviewer_qcQH · 2023-06-18

**Soundness:** 3 good
**Presentation:** 3 good
**Contribution:** 3 good
**Rating:** 7
**Confidence:** 4

**Summary:**

The paper studies OOD generalization in the presence of spurious correlation. First, it provides a theoretical analysis showing that during ERM training, both spurious and invariant features are learned but at different rates, which, in turn, influences the performance of the following optimization with OOD objectives, such as IRM. Then, it leverages the theoretical analysis to propose a new feature pre-training method to facilitate learning a more diverse set of features. The empirical evaluation shows that the proposed method improves standard ERM pre-training and other feature pre-training baselines in various settings.


**Strengths:**

The paper provides a good combination of theoretical and empirical studies. Both parts are well-motivated, sound and offer a valuable contribution to the field:
- The theoretical results deepen our understanding of how ERM learns different features and how this learning schedule interacts with further OOD training and generalization.
- The proposed method, motivated by the theoretical analysis, is an efficient and effective way of improving OOD performance, as shown on several datasets.

**Weaknesses:**

- Discussions provided after each formal statement helps understand their implications. However, the overall clarity and presentation of the paper could be improved to be more friendly for a more general audience. For example:
    - Fig. 2 – more details can be presented in the caption, e.g., it is unclear what the “Feature Learning” axis means. Are those coefficients introduced in Lemma 3.2?
    - It is not entirely clear what “OOD objectives” mean before L122.
    - The description of the FAT method could be more elaborate. For example, the description of the retention mechanism via saving previous linear layers could be explained more explicitly.
- The theoretical setting and proofs use the linear activation function, effectively rendering the model linear. While the authors mention in the main text that this framework can be extended to ReLU activation functions, it would be good to provide more details in the proof of where and how the authors think the referred works (L119) could be applied to extend their results to non-linear networks.


**Questions:**

- According to the provided theory, pre-training features with ERM for longer seems always beneficial. However, as shown in Fig.1 left and mentioned in experiments (e.g., L306-307), overfitting can occur, and OOD performance decreases with further pre-training. Does your theory also explain this phenomenon?
- It was unclear why the memory cost was too significant to switch to iFAT. Couldn’t the subsets be stored as masks, which would not increase the memory costs considerably, given that K equals 2 or 3 in practice? The additional parameters are the linear layers w_i, which also should not contribute much to the memory cost.


**Limitations:**

- As mentioned in the weaknesses, the presented proofs are limited to linear models. Additional work is needed to extend these results to the case of non-linear CNN.

---

> ### Author Rebuttal · Authors · 2023-08-05
>
> Thank you for your time in reviewing our paper and your positive feedback! We hope our response below would make you more confident in supporting our work.
>
> > W1.1  Axis in Fig.2.
>
> We have revised the caption of Fig.2 to include the details.  As mentioned in Appendix C.1, the invariant and spurious feature learning terms plotted in Fig. 2 are the means of $\langle \mathbf{w}\_{j,r}, j\mathbf{v}\_1 \rangle$ and $\langle \mathbf{w}\_{j,r}, j\mathbf{v}\_2 \rangle$ for $j\in \lbrace\pm 1\rbrace, r\in[m]$, respectively.
>
> > W1.2  OOD objective.
>
> We have revised our draft to make it clearer: we refer OOD objective as the additional penalty terms developed to regularize ERM to capture the invariant features, as introduced in line 29.
>
> > W1.3 Retention mechanism.
>
> We have revised our work to explicitly describe the retention mechanism: FAT needs to retain the already learned features by minimizing the empirical risk at $G^r$, for which we store and use the historical classifiers $w_i$ with the current featurizer to evaluate the feature retention degree.
>
> > W2 Extending results to non-linear.
>
> Here we showcase how our two key theorems can be extended to non-linear settings:
>   - For the ERM feature learning  (Thm 4.1), we have updated our draft to include the analysis for non-linear activation functions. The result is that for smooth or piecewise linear activation functions (ReLU, Leaky ReLU, Softplus, etc.), suppose we run $T = O(d\log{d})$ iterations of GD for the ERM objective (in the early stage of training), the feature learning terms (e.g., $\sigma(\langle \mathbf{w}\_{j,r}, j\mathbf{v}\_1 \rangle)$) will be proportional to the empirical distributions of the signal strengths (e.g., $\textup{Rad}(\alpha)$) up to an error of $O(d^{-\Omega(\zeta)})$ with $\zeta\in (0, \frac{1}{4})$ being a fixed constant. The proof ideas are summarized as follows: We adapt the results in [A] (Thm 3.2), which shows that in the early stage of GD training, for a network that is sufficiently wide (NTK region), its output can be well approximated by a linear model, and the approximation error scales as $O(d^{-\Omega(\zeta)})$.  Based on this approximation, we extended our analysis in the linear case to allow non-linear activation functions. We observe a weaker dominance of the spurious feature in the non-linear case both theoretically and empirically.
>
>   - For IRM feature learning (Thm 4.2), we have updated our draft to include the analysis for non-linear activation functions. To account for non-linear activation, we introduce an additional assumption, which requires that non-linear activation is Lipschitz and Smooth: activation function is smooth, $\psi'(0) \le \beta$, $|\psi'(x)-\psi'(x')|<\beta|x - x'|$ and Lipschitz $|\psi(0)| < L $, $|\psi(x)-\psi(x')|< L |x - x'|$. Based on the additional assumption, we can extend our analysis in the linear case to allow non-linear activation functions.
>
>
>
> > Q1 ERM pre-training for longer.
>
> We need to clarify that **our theory does not imply that longer ERM pre-training would benefit the feature learning** while suggesting the ERM learns both invariant and spurious feature learning till convergence. In other words, ERM feature learning will **saturate** after a certain number of steps, which aligns with the empirical evidence in Fig. 10 of [25].
>
> The performance decrease of ERM feature learning in Fig. 1b implies one of the ERM feature learning drawbacks, which may be because of the “simplicity bias” that ERM tends to learn simple functions [50]. Our additional analysis in the rebuttal pdf also suggests that ERM can **forget** certain useful features.
>
> There are more other factors that could influence feature learning, such as network architectures and optimization algorithms. As the first work that theoretically characterizes feature learning of ERM and OOD objectives, incorporating all of those factors could unnecessarily improve the difficulty of the analysis. Nevertheless, as discussed in Appendix A, we believe it is a promising future extension of our theory.
>
> > Q2 memory cost
>
> We need to clarify that FAT needs to store all $D^a_i$, $D^r_i$ and $w_i$ yielded in previous rounds. The storage of these items does not cause much memory issue, while **training using FAT objective (Eq.7) with all previous subsets can lead to OOM for a large network**, as Eq. 7 increases the batch size by a factor of $k$ for the $k$-th round. Note that typically, the batches are also sampled from each environment as shown in Table 9. Let the number of sampled domains be $d$, and the batch size from each domain be $b$, then each round will additionally introduce $2d\times b$ samples in each minibatch, which is typically more than 200 and leads to OOM issue for a V100 GPU with around 32G GPU memory.
>
> We also show the running time and memory cost of different methods in the "global" response.
>
> **References**
>
> [25] On feature learning in the presence of spurious correlations, NeurIPS'22.
>
> [50] The pitfalls of simplicity bias in neural networks, NeurIPS'20.
>
> [A] The surprising simplicity of the early-time learning dynamics of neural networks, NeurIPS’20.
>
> We are happy to answer any outstanding questions, and we’d appreciate it if you could jointly consider our responses when making the final evaluation of our work.

---

> > ### Comment · Reviewer_qcQH · 2023-08-15
> >
> > I thank the authors for their response clarifying my comments and questions. I increase my confidence and the presentation score assuming the mentioned clarifications will make it to the camera-ready version.

---

> > > ### Author Response · Authors · 2023-08-16
> > > **Thank you**
> > >
> > > Thank you for checking our rebuttal. We're glad that our responses are able to address your comments and questions. If you have any other comments, feel free to let us know. Thank you again for your valuable comments and suggestions.

---

### Official Review · Reviewer_Jcjd · 2023-07-02

**Soundness:** 3 good
**Presentation:** 1 poor
**Contribution:** 2 fair
**Rating:** 5
**Confidence:** 4

**Summary:**

The paper consists of two parts. The first part provides a theoretical analysis of the training dynamics for a simple model and data distribution under ERM and IRM. Specifically, the authors explore the questions of feature learning, when one of the features changes its correlation to the target between environments (spurious feature) and the other does not (core feature). The second part proposes a training method for extracting reach features, and shows promising results across an array of benchmarks.

**Strengths:**

**S1.** The paper provides an extensive theoretical analysis of gradient descent dynamics under ERM and IRMv1 for a specific model and data distribution.

**S2.** The paper proposes a novel training method (FAT) which shows strong performance in terms of feature learning

**Weaknesses:**

### W1: Theory

The paper places a lot of emphasis on the theoretical analysis, which occupies pages 3-6. The authors use the theoretical results to provide intuition about feature learning in neural networks, and training dynamics of ERM and IRM.

Unfortunately, the model used is extremely simple. While the authors argue that it is a convolutional neural network, it is in fact **a linear model**. Indeed, the authors define the model as
$f(W, x) = F_{+1}(W_{+1}, x) - F_{-1}(W_{-1}, x)$, where
$F_j(W_j, x) = \frac 1 m \sum_{r=1}^m [w_{j, r}^T x_1 + w_{j, r}^T x_2]$, where I omitted the activation $\sigma$, which the authors set to be the identity mapping.

We can then rewrite the model as a linear model
$f(W, x) = \frac 1 m \sum_{r=1}^m [w_{+1, r} - w_{-1, r}]^T (x_1 + x_2) = \tilde w^T \tilde x$, where $\tilde w = \frac 1 m \sum_{r=1}^m [w_{+1, r} - w_{-1, r}] $ is the effective weight vector and $\tilde x = x_1 + x_2$ is the effective feature vector.
In other words, the "CNN" model used by the authors is simply a reparameterized linear model.
Moreover, the reparameterization does not really change the gradient dynamics in a non-trivial way, as
$\frac {\partial L}{\partial w_{j, r}} = \frac {\partial L}{\partial \tilde w} \frac {\partial \tilde w} {\partial w_{j, r}} = \frac {j}{m} \nabla_{\tilde w} L$.
In other words, all of the weights $w_{j, r}$ are updated with $\pm$ the gradient of the linear model divided by $m$.
So **understanding the training dynamics in the "CNN" model is equivalent to understanding the training dynamics of a linear model**.

To summarize, the authors study the training dynamics of linear model with a logistic loss.
The authors do not clarify this connection in the paper, which in my opinion is misleading. This omission also unnecessarily complicates the presentation.

I also believe there are several important implications of the fact that the authors analyze the training dynamics of a linear model:
- It is not clear what is even meant by feature learning. Specifically, the authors refer to how much weight the model assigns to each of the input features. However, this is more similar to last layer / classifier training in the context of [1] that the authors reference, rather than feature learning.
- Consequently, it's unclear what conclusions can even be made from the experiments about _feature learning_ in _neural networks_. The connection here seems far-fetched.
- The setup for the IRM is quite strange. Specifically, the feature that the model outputs in this context is the logit (a number) predicted by the model, and the classifier is just fixed to be $1$. As a result, the authors get a logistic regression model with some additional gradient penalty. I am not an expert on IRM, but it is not clear how relevant this model is to _feature learning_ in neural networks with IRM.

Please correct me if I am wrong in the reasoning above!

### W2: Theory $\leftrightarrow$ Methodology

The connection between the theory and the proposed method (FAT) is not clear to me.
It seems like the main conclusion from the theory comes down to the idea that we need to learn diverse features, which ERM by itself might not do.
However, as I mentioned above, the relevance of the theory to feature learning in neural networks is questionable.

In particular, the theory does not connect as far as I can tell to any of the details of the method.
The method could very easily be presented without the theory.

### W3: Presentation

Because so much emphasis is placed on the theoretical results (which are in my opinion of limited relevance), the authors have to describe the method and the experiments in a limited space.
The presentation of the method is not very clear.
In particular, many of the design decisions are not explained.
For example, why do we need to reinitialize the weights of the classifier in each "round"?
The datasets $\mathcal{D_i^a}$ are never defined in the text, but used in line 262.
Overall, it is quite hard to follow the description of the algorithm, and the intuition behind it.

Moreover, the iFAT method which is actually used in all of the experiments is not even described in the paper.

I would recommend to deemphasize the theory, and use most of the space in the paper to clearly describe the method, present detailed experimental results and ablations on the various design choices.

### W4: Performance

Overall, FAT seems to consistently provide good results in Table 2. However, it's worth noting that the improvements over ERM appear to be fairly small (<1%) except for Camelyon17.

**References**

[1] [_Last Layer Re-Training is Sufficient for Robustness to Spurious Correlations_](https://openreview.net/forum?id=Zb6c8A-Fghk);
P. Kirichenko, P. Izmailov, A. G. Wilson;
ICLR 2023

**Questions:**

**Q1.** In Eq. 5, why is the second term divided by $n_e^2$ and not $n_e$?

**Q2.** You mention in line 279 that FAT comes with additional memory requirements, especially if the feature extractor has many parameters. Why is that? What exactly do you need to store?

**Q3.** How exactly do you run DFR on CivilComments and Camelyon17? The results, even for ERM+DFR, appear to be surprisingly good, and in particular better than what's currently reported in the [WILDS leaderboard](https://wilds.stanford.edu/leaderboard/)?

**Limitations:**

The limitations are adequately addressed in my opinion, except for the issues raised in the weakness section.

---

> ### Author Rebuttal · Authors · 2023-08-09
>
> Thank you for your time in reviewing our work. Please see our detailed responses to your comments and suggestions below where we use references in our draft due to the token limit.
>
> > W1.1 The linear activation in the CNN model
>
> We respectfully disagree with the point:
> - First, we need to clarify that, as the first work analyzing the ERM and IRMv1 feature learning under distribution shifts, the results with IRM regularization (which is **non-linear**) we obtained are not obvious under **non-convexity** [a] despite the linear activation. In fact, The assumption of linear activation is widely used and standard in the literature of theory analysis in OOD generalization [4,32,40,61,64]. Linear CNN is also widely studied by the community [b,c,d].
>
> - Besides, **we also show that our key theoretical results could be generalized to the non-linear setting**. Please find the details in [our response to Reviewer qcQH](https://openreview.net/forum?id=eozEoAtjG8&noteId=KmRvdQIZ2C) due to the character limits.
>
> - We adopt the linearity activation function primarily for the sake of simplicity and clarity when studying IRMv1. It's worth noting that the IRMv1 involves high-order derivatives, which can be a significant challenge to learning theory.
>
> - Going beyond the linearity of activations, the experiments in Fig. 1b show that, our theoretical results align with the empirical discoveries in more complex settings.
>
> > W1.2 The meaning of feature learning
>
> We need to clarify that feature learning in neural networks refers to how the weights of a neural network **evolve to extract different features (Eq.6)**, especially when trained from scratch [2,11,51,58]. **The objectives are not limited to ERM**.
>
> The last layer training method is applied to the **fixed trained features** and is limited to ERM, which serves as an indirect measure of feature learning for deep and complex networks where explicit quantities in Eq. 6 are not accessible.
>
> Another key difference is that our analysis studies training a neural network based on **dataset containing spurious correlations (aligned with the setting in OOD generalization literature)**, while last layer training requires an **unbiased dataset** (i.e., without spurious correlations) to (indirectly) examine the invariant feature learning.
>
> > W1.3 The setup for IRM
>
> The use of scalar in IRMv1 is because of the complicated formulation of the original IRM framework which involves a bilevel optimization (more details can be found in [13]). The gradient penalty in IRMv1 is **a practical variant to regularize** the feature learning in deep networks to focus on invariant features, and has gained lots of success[4].
>
> To avoid misunderstanding, we removed the sentence  “defined classifier w as the scalar 1” in the draft. We understand that Reviewer Jcjd may not be familiar with the literature of OOD generalization and IRM, nevertheless, we are happy to provide more details for any specific points unclear to Reviewer Jcjd.
>
> > W2 Theory and method.
>
> As clarified, our theory precisely analyzes the feature learning of a CNN when optimized via ERM and IRMv1, and characterizes the functionalities of ERM and OOD objectives. Specifically, the second part of our theory implies that IRMv1 solely can not learn features but requires high quality feature representations for OOD generalization, hence the pre-training stage needs to learn high quality features. As pre-training with ERM may not learn all useful features for OOD generalization, our algorithm is thus motivated to strengthen the pre-training stage for better OOD generalization.
>
> **Without our theory, it’s unclear which stage needs to be improved and what objectives need to be used for improving OOD generalization.**
>
> > W3 Presentation
>
> As shown in Eq. 7, we need to initialize (instead of “reinitialize”) a new classifier in each round to learn new features from the augmentation set while keeping the historical classifiers to retain the features already learned in each previous round.
>
> We complemented the introduction of $D^a_i, D^r_i$ in line 262: $D_i^a$ and $D_i^r$ are the corresponding augmentation and retention set elicited at $i$-th round.
>
> We also gave specific details in the revised paper, about the differences between iFAT and FAT, that iFAT stores only $D_{k-1}^a, D_{k-1}^r, w_{k-1}$ at the $k$-th round.
>
> > W4 Performance
>
> We need to clarify that a large improvement in these real-world datasets is extremely more challenging than other ML tasks, as one may see in the Wilds leaderboard. The consistent improvements of FAT can serve as strong evidence for its superiority.
>
> > Q1 Eq 5
>
> $L_e$ is the averaged ERM risk among $n_e$ samples in environment $e$. The IRMv1 penalty (Eq.4) takes the square of the gradients wrt. $L_e$ and thus $1/n_e$ in $L_e$ is $1/n_e^2$ in Eq. 5.
>
> > Q2 Memory cost
>
> FAT needs to store all $D^a_i$, $D^r_i$ and $w_i$ from previous rounds. The storage of these items does not cost much memory, while training using FAT objective (Eq.7) with all previous subsets can lead to OOM for a large network, as Eq. 7 increases the batch size by a factor of $k$ for the $k$-th round.
>
> > Q3 DFR results
>
> Note that the results are aligned with DFR [28] and its seminal works [25,46]. We strictly follow their protocol with details provided in line 325 and Appendix E.2.2, that we use an unbiased subset to train the last layer with frozen ERM learned features. The results demonstrate ERM already learns invariant features, aligned with our Thm 4.1.
>
> **References**
>
> [a] Deep linear networks with arbitrary loss: All local minima are global, ICML’18.
>
> [b] Implicit bias of gradient descent on linear convolutional networks, NeurIPS’18.
>
> [c] Representation costs of linear neural networks: Analysis and design, NeurIPS’21.
>
> [d] Inductive bias of multi-channel linear convolutional networks with bounded weight norm, COLT’22.

---

> ### Author Response · Authors · 2023-08-17
> **A summary of rebuttal in response to Reviewer Jcjd**
>
> Dear Reviewer Jcjd,
>
> We want to thank you again for reviewing our paper. We have responded to each of the weaknesses/questions you raised in the review. In summary,
>
> - We have clarified the misconception in your reasoning about the linear activation function, and showed that our results could be generalized to the non-linear activation function;
> - We have shown how our theoretical results could motivate the proposed method;
> - We have improved the paper presentation following your suggestions.
>
> We would appreciate it if you could take a look at our responses and let us know if any of your remaining concerns are not addressed, and we would try our best to address them.

---

> ### Author Response · Authors · 2023-08-19
> **A gentle reminder for the closing rebuttal window**
>
> Dear Reviewer Jcjd,
>
> We would like to remind you that the rebuttal will be closed very soon. To allow us sufficient time to discuss with you your concerns about our work, we would appreciate it if you could take some time to read our rebuttal and give us some feedback. Thank you very much.

---

> > ### Comment · Reviewer_Jcjd · 2023-08-19
> > **Updated the score**
> >
> > Dear authors, thank you for the rebuttal. Based on the clarifications and the new results, I have updated the score to a borderline accept. I believe some of my concerns still hold, such as the limitations of the theory, and limited empirical improvements, but I am not opposed to accepting the paper.

---

> > > ### Author Response · Authors · 2023-08-20
> > > **Thank you for checking our rebuttal**
> > >
> > > Dear Reviewer Jcjd,
> > >
> > > Thank you very much for checking our rebuttal. We are glad that our clarifications and new results changed your opinion about our work. We understand there are still limitations in our theory and method, but as the first work to characterize the feature learning of ERM and OOD objectives and their interactions under distribution shifts, we believe our work could facilitate the understanding of feature learning under distribution shifts and lay the foundation for future work on representation learning. Future extensions could be generalizing our theory to more complex networks and objectives, and improving the data partitioning and feature learning in our method to better tackle the challenging real-world OOD generalization problem.

---

### Official Review · Reviewer_NB5i · 2023-07-04

**Soundness:** 3 good
**Presentation:** 2 fair
**Contribution:** 3 good
**Rating:** 7
**Confidence:** 4

**Summary:**

This work aims to understand and compare feature learning in ERM and certain OOD generalization objectives. Additionally, it proposes an approach to enhance feature learning for improved OOD generalization.

First, the authors examine data consisting of invariant and spurious features. They theoretically show that when training a two-layer CNN with ERM, it learns both the features. However, when the spurious correlation in stronger, the spurious features are learned faster. Additionally, they explore the IRMv1 objective and illustrate that fine-tuning the ERM-trained model using the IRMv1 objective does not result in learning of new features.

To improve feature learning compared to ERM, the authors propose a technique called feature augmented training (FAT). At each training step, they partition the data into two sets based on the correctness of predictions. The set with accurate predictions is added to an augmentation set, while the set with incorrect predictions is added to a retention set. Both sets expand with each training round. The training objective involves applying Distributionally Robust Optimization (DRO) on the augmentation set to learn new features, combined with ERM on the retention set to retain the learned features.

The authors demonstrate that using OOD objectives on models trained with FAT leads to improvement in OOD performance on two variants of the colored MNIST dataset and six datasets from the WILDS benchmark.

**Strengths:**

1. This work aims to improve feature learning for better OOD generalization, which has been identified as a limitation of existing OOD objectives in recent research [1]. While the authors acknowledge that the idea to do this is not new [2], the proposed approach FAT is novel and demonstrates effectiveness in enhancing OOD performance. FAT incorporates a termination step to halt further training once the model has acquired sufficiently rich features, and the authors also propose an efficient version, iFAT, to make the approach practical for implementation.

2. The theoretical results presented in this work provide intriguing insights into feature learning with ERM and the observation [1] that OOD generalization objectives do not improve feature learning.

3. In their comparisons, the authors extensively evaluate ERM and Bonsai [2]—the primary contenders for feature learning—as well as various OOD objectives applied on top of these methods, across multiple datasets (as shown in Table 2). The results demonstrate that FAT enhances OOD performance.

**Weaknesses:**

1. The claim that ERM does not learn sufficiently good feature representations, while FAT improves them, requires stronger support and evidence.

2. Some sections of the writing require clarifications to improve understanding and readability.

**Questions:**

1. Improvement in feature learning compared to ERM:

- In Section 6, it is mentioned that FAT, Bonsai [2] and ERM are generally trained for the same number of overall epochs. However, given that ERM is computationally less expensive than the other two methods, it would be valuable to investigate whether training ERM for a longer duration can further enhance the learned feature representations, or if the representations stop improving after a certain number of training epochs. This analysis would provide insight into the extent to which FAT and Bonsai truly improve feature learning, especially considering that they do not consistently lead to significant improvements in OOD performance in many cases (as observed in Table 2).

- To better understand and compare feature learning in ERM and FAT, it would be beneficial to include additional analyses, such as visualizing the saliency maps using GradCAM.

2. Clarity:

- In the caption for Fig. 1, the description of FAT is unclear and it does not adequately explain the illustration of FAT in Fig. 1(a). Although the experimental results in Figure 1(b) provide informative insights, it would be more comprehensive to include details about the dataset and the number of training epochs used for FAT.

- The description of the method in the introduction (lines 55-58) is unclear and difficult to comprehend without referencing Algorithm 1.

- Concerning the algorithm, the rationale behind Line 16 (returning the average of all linear classifiers) is unclear and could benefit from further explanation.

3. Minor:

- Please address any typos, e.g., in the Fig. 1 caption, and inconsistencies in notation, e.g., in Equation (5), Lemma 3.2, etc.

- It may be worth considering renaming the method from FAT to FeAT, as it better reflects the focus of the approach on improved feature learning.

References:

[1] P. Kirichenko, P. Izmailov, and A. G. Wilson. Last layer re-training is sufficient for robustness to spurious correlations. arXiv preprint arXiv:2204.02937, 2022.

[2] J. Zhang, D. Lopez-Paz, and L. Bottou. Rich feature construction for the optimization-generalization dilemma. In International Conference on Machine Learning, pages 26397–26411, 529 2022.

**Limitations:**


[-] Reproducibility: The authors have not provided the code for implementing their approach. While the Appendix contains most of the necessary implementation details, sharing the code would greatly facilitate the verification of their findings.

[+] The authors acknowledge the limitation that feature learning is influenced by various factors, and this work specifically aims to investigate feature learning in the context of spurious and invariant features.

---

> ### Author Rebuttal · Authors · 2023-08-09
>
> Thanks for your support and constructive comments! Please see our detailed responses to your comments and suggestions below where we use reference numbers in our manuscript due to the character limit:
>
> > Q1.1 The effects of longer ERM training epoch.
>
> Thank you for the insightful question. **In fact, the representations of ERM will stop improving after a certain number of ERM training epochs, shown in Fig. 10 of [25], or even decrease when being fed to OOD training for some datasets (Fig.1b)**, which highlights a drawback of ERM feature learning. Note that for deep models, the feature learning will quickly saturate within one epoch [25], within the training epochs we use (in Table 10). Therefore, the consistent improvements by FAT are non-trivial.
>
> > Q1.2 GradCAM
>
> We followed your suggestion and plotted the saliency maps to better understand the feature learning of different algorithms. The figures are attached to the rebuttal pdf and to our latest draft as well.
>
> We first visualize the feature learning of ERM and FAT on ColoredMNIST-025. It can be found that
> - ERM can learn both invariant and spurious features to predict the label, aligned with our theory.
> - However, ERM focuses more on spurious features and even **forgets** certain features with longer training epochs, which could be due to multiple reasons such as the simplicity biases of ERM[a]. Hence predictions based on ERM learned features fail to generalize to OOD examples.
> - In contrast, FAT effectively captures the meaningful features for all samples and generalizes to OOD examples well.
>
> We also visualize the saliency maps of ERM, Bonsai and FAT on all real-world datasets used in our work. It can be found that, across various tasks and data modalities, FAT effectively learns more meaningful and diverse features than ERM and Bonsai, which serve as strong evidence for the consistent superiority of FAT in OOD generalization.
>
> > Q2.1 Caption of Fig.1,
>
> We included more details when introducing FAT:
>
> FAT iteratively checks and divides $D_{tr}$ into augmentation $D^a$ and retention sets $D^r$ that contain features not learned and already learned by the current model at the round, respectively. Then FAT augments the model with new features while retaining the already learned features, which leads to richer features for OOD training and better OOD performance.
>
> We also included the dataset and epochs/rounds used in FAT when introducing the experiments of Fig.1.
>
> > Q2.2 Introduction
>
> We revised the introduction with a more intuitive explanation:
>
> In each round, FAT separates the train set into two subsets according to whether the underlying features in each set are already learned (Retention set $D^r$) or not (Augmentation set $D^a$), by examining whether the model yields correct ($D^r$) or incorrect ($D^a$) predictions for samples from the subsets, respectively. Intuitively, $D^a$ and $D^r$ will contain distinct features that are separated in different rounds. Then, FAT performs distributionally robust optimization (DRO) on all subsets, which *augments* the model to learn new features by minimizing ERM losses on all $D^a$ and *retains* the already learned features by retaining ERM losses on all $D^r$.
>
> > Q2.3 Explanation of the algorithm return
>
> We revised the paper to add the explanation. The rationale is that the average of historical classifiers could be a good initial point as they already capitalize all the learned features in each round.
>
> > Q3.1 Typos
>
> We have double checked and corrected all the typos and inconsistencies.
>
> > Q3.2 A new name
>
> Thanks for this wonderful suggestion! We switched the name to FeAT in the revised draft! Nevertheless, to avoid confusion to other reviewers, we will use FAT for consistency during rebuttal.
>
> > L1 Reproducibilty
>
> As claimed “ We will provide a link to an anonymous repository of our code during the discussion phase.” during submission, we have provided the code in an anonymized link to AC, following the rebuttal guideline that we are not allowed to disclosure any external link in the rebuttal.
>
> **References**
>
> [a] The pitfalls of simplicity bias in neural networks, NeurIPS'20.
>
>
> Please let us know if you have any further questions. We’d be grateful if you could take the above responses into consideration when making the final evaluation of our work.

---

> > ### Comment · Reviewer_NB5i · 2023-08-11
> >
> > Thank you for the detailed responses. Many of my concerns have been addressed. Additional comments/questions:
> > 1. Regarding saliency maps shown in the PDF: Thank you for including these visualizations for several datasets.
> > - For Fig. 1, what is the color scale for grey-colored GradCAM and GradCAM visualizations? It is hard to interpret the visualizations without the color scale.
> > - For Fig. 2, it would be useful to discuss what features are useful and task-relevant for the image datasets. I can see differences in the saliency maps for images from FMoW and iWildCam datasets across the three methods, whereas the differences for images from Camelyon17 and RxRx1 datasets are not very clear.
> > - It would be useful to include more examples for datasets from the WILDS benchmark in the revised version. I also suggest including the predicted and ground truth labels for these examples for better understanding. E.g., in the current example from CivilComments, while FAT relies less on some demographic attributes compared to ERM, all methods seem to rely highly on the word *ridiculous*, which is relevant to the task.
> > - There are some typos in the captions that should be corrected.
> >
> > 2. Regarding description of the algorithm: I still think that the description of the method in the Fig. 1 caption and introduction of the paper needs further clarity. Consider using separate sentences for the description and the intuition to make it easier to understand. In the current explanation, it is not clear that the augmentation and retention sets grow at every step. Some phrases like *augmenting the model* and *retaining ERM losses* can also be refined.

---

> > > ### Author Response · Authors · 2023-08-12
> > > **Response to Reviewer NB5i (1/2)**
> > >
> > > Thank you for the comments! We are glad to hear that many of your concerns are addressed in our previous response. Let us address your left questions:
> > >
> > > > Q1.1 Color scales:
> > >
> > > Both of them have a color scale of [0,255], which is scaled up from the originally normalized data in [0,1].
> > > - The grey-colored GradCAM only contains GradCAM information: a whiter color indicates a higher gradcam value.
> > > - The GradCAM converts the grey-colored GradCAM to RGB: a warmer color indicates a higher gradcam value.
> > >
> > > Correspondingly, when a region in the grey-colored GradCAM visualization has a whiter color, it has a warmer color in the GradCAM visualization.
> > >
> > > We will include the aforementioned details in our caption of the GradCAM visualizations.
> > >
> > > > Q1.2 What features are useful and task-relevant in Fig.2.
> > >
> > > We agree that it is important to include more details about what the task-relevant features are. Regrettably, the authors are not experts in pathology (tumor tissue detection in Camelyon17) nor biology (genetic treatment effects detection in RxRx1), and could only provide some intuitive discussion about the learned features in Camelyon17 and RxRx1:
> > > - Camelyon17:
> > >   - Relevant features: Typically, a pathologist will look into the sizes, shapes and arrangements of the cells and their nucleus to identify the tumor issues. The small nodes indicate normal lymph cells while the cells with large cell nucleus could be macrophage cell that imply the potential infection and the spread of cancer.
> > >   - Analysis of the example: At the up left part of the center image, there exist two large cell nucleus, which are identified by ERM and FAT, but failed to be recognized by Bonsai.
> > > - RxRx1:
> > >   - Relevant features: The raw data is the image of ﬂuorescent microscopy images of human cells and one typically needs to look into the characteristics of the cells to identify which genetic treatment is used, including the morphology and the distribution of the cells.
> > >   - Analysis of the example: For the white regions in the image that imply some morphology cell features, it can be found that Bonsai and ERM can fail to capture full cell features at the up right part.
> > >
> > > From the rich feature learning perspective, we need to clarify that **the model is expected to learn all predictive features, no matter whether they are spurious or not**. Typically, both the ERM and Bonsai learned features are part of the predictive features. From the shown examples, it can be found that:
> > > - ERM can fail to capture part of Bonsai learned features;
> > > - Bonsai can fail to capture part of ERM learned features;
> > > - FAT typically captures both ERM and Bonsai learned features;
> > >
> > > We believe the visualizations could be strong evidence confirming the better capability in learning rich features of FAT, compared to its counterparts.
> > >
> > > > Q1.3 More examples and better understandings.
> > >
> > > Yes, due to the page limit, we could only showcase one example randomly selected from each Wilds dataset, and we will visualize more examples, including the original images, the labels, and the predictions in the revised version.
> > >
> > > The predictions and labels for the current examples are:
> > > |              | Camelyon17 | FMoW | iWildCam | RxRx1 | CivilComments | Amazon |
> > > |--------------|------------|------|----------|-------|---------------|--------|
> > > | Ground Truth | 1          | 40   | 36       | 1138  | toxic         | 2      |
> > > | ERM          | 1          | 40   | 36       | 812   | toxic         | 3      |
> > > | Bonsai       | 0          | 40   | 36       | 812   | toxic         | 3      |
> > > | FAT          | 1          | 40   | 36       | 812   | toxic         | 2      |
> > >
> > > Note that **the primary objective of feature learning is to capture all predictive features**, while making correct predictions is not the primary objective as we could apply OOD methods such as IRMv1 to identify the invariant features.
> > >
> > > For the example of CivilComments, all algorithms make correct predictions, while both ERM and Bonsai could fail to capture sensitive features such as “persecution of jews , intellectuals , communists , social democrats , etc” in the shown example. In contrast, FAT could fully capture the feature.
> > >
> > > The same phenomenon is also observed in Amazon example, where ERM and Bonsai could fail to fully capture some relevant features such as “never provides entertainment”, “rather”, “nothing to say”, “too long” and etc..
> > >
> > > > Q1.4 Typos in the captions.
> > >
> > > We appreciate the carefulness of Reviewer NB5i. We double checked and corrected typos such as “ Saliency map of feature learning”, “the learned features that contributed most”.

---

> > > ### Author Response · Authors · 2023-08-12
> > > **Response to Reviewer NB5i (2/2)**
> > >
> > >
> > > > Q2.1 Caption of Fig.1 and the introduction of the paper.
> > >
> > > We further refined the descriptions according to your suggestions, where we use bold fonts to highlight the modified texts:
> > > - Caption of Fig.1: Iteratively, FAT divides $D\_{tr}$ into augmentation $D^a$ and retention sets $D^r$ that contain features not learned and already learned by the current model at the round, respectively. **In each round, FAT augments the model with new features contained in the growing augmentation sets while retaining the already learned features contained in the retention sets, which will lead the model to learn richer features for OOD training and obtain a better OOD performance**.
> > > - Introduction: In each round, FAT separates the train set into two subsets according to whether the underlying features in each set are already learned (Retention set $D^r$) or not (Augmentation set $D^a$), by examining whether the model yields correct ($D^r$) or incorrect ($D^a$) predictions for samples from the subsets, respectively. Intuitively, $D^a$ and $D^r$ will contain distinct features that are separated in different rounds. Then, FAT performs distributionally robust optimization (DRO) on all subsets, which *augments* the model to learn new features by **minimizing the maximal ERM loss on all $D^a$** and ***retains* the already learned features by minimizing ERM losses on all $D^r$.** **Along with the growth of the augmentation and retention sets, FAT is able to learn richer features for OOD training and obtain a better OOD performance**.
> > >
> > > We hope our revised version will be clearer and more intuitive to understand. Nevertheless, we are happy to take suggestions to further improve it.
> > >
> > > Please let us know if you have any further questions. We’d be grateful if you could take the above responses into consideration when making the final evaluation of our work.

---

> > > > ### Comment · Reviewer_NB5i · 2023-08-12
> > > >
> > > > I appreciate the authors' efforts in providing further details and discussion regarding the visualizations. I have increased my score based on the responses.

---

> > > > > ### Author Response · Authors · 2023-08-13
> > > > > **Thank you for your constructive comments**
> > > > >
> > > > > We would like to thank you again for your time and efforts in reviewing our paper, as well as for your continuous support of our work. Your insightful comments have helped to improve the paper a lot!
> > > > >
> > > > > Sincerely,
> > > > > The Anonymous Authors.

---

### Official Review · Reviewer_KJZ8 · 2023-07-06

**Soundness:** 3 good
**Presentation:** 2 fair
**Contribution:** 3 good
**Rating:** 5
**Confidence:** 5

**Summary:**

This paper explores the relationshp between the ERM training and OOD generalization in feature learning. The authors analyze the corresponding learned features by ERM and OOD objectives. To answer the question, they conduct the investigation of feature learning in a two-layer CNN network training with ERM and IRMv1. They adopt the data models proposed by [2, 11], and include features with different correlation degrees to the labels to simulate invariant and spurious features like [26]. The theoretocal results extend  [51] from data augmentation to ERM learning process. They find that ERM fails since it learns the spurious features more quickly than invariant features, when spurious correlations are stronger than invariant correlations. However,  invariant feature learning also happens with RRM so long as the invariant feature has a non-trivial correlation strength with the labels. Moreovoer, they find that IRMv1 requires sufficiently well-learned features for OOD generalization. Compared with the former workshop version, this conference submission adds the so-called  Feature Augmented Training (FAT) to learn features for OOD generalization. The proposed method iteratively augments the model to learn new features while retaining the already learned features. In each round, the retention and augmentation operations are performed  distributionally robust optimization on different subsets of the training data that capture distinct features. The experimental results verify the promising OOD generalization performance of the proposed method.

**Strengths:**

1. Interesting theoretocal findings.

2. Promosing empirical results.



**Weaknesses:**

1. The symbol system seems hard to follow. Some superscripts and subscripts are redendunt: L_S in Eq.(2), \ell‘ in Eq.(5), o_d in Thm 4.2. Please double-check these symbols and make them clear.

2. Some symbols are not well-defined, such as the \cdot in the predictor f, little o, big O, \Omega, \Theta.

3. The activation function and the variance share the same symbol.

4. Thm 4.1 (informal) does not provide the quantitative results to describe how much the incerment of spurious feature is larger than that of the invariant feature at any iteration. Althrough this is a informal statement, the quantitative results is important.

5. It is better to discuss the computational complexity of the proposed method and compare with the other competitors.

**Questions:**

1. It is better to claim Lma 3.2 in the supplementary material.

2. The details of Fig.2 are imoportant. It should be presented in the main body.

**Limitations:**

Yes.

---

> ### Author Rebuttal · Authors · 2023-08-09
>
> Thank you for your time and your positive feedback! Please see our detailed responses to your comments and suggestions below.
>
> > W1 Some superscripts and subscripts are redendunt: L_S in Eq.(2), \ell‘ in Eq.(5), o_d in Thm 4.2. Please double-check these symbols and make them clear.
>
> Thanks for your suggestions. We have revised our draft accordingly:
> - We changed $L_S$ to $L$ in Eq.(2);
> - We changed $\ell_i’$ to $\ell\_i’^e$ in Eq. (5) referring to the first order derivative of $L_e$ with respect to the $i$-th sample at $e$-th environment.
> - By using $o_d$, we mean that the value is small with respect to the dimension $d$. This means that the requirements of the value don't grow with $d$, at least asymptotically.
>
> We also double checked the other notations and simplified the superscripts and subscripts when unnecessary. Besides, we provided a table of key notations at the Appendix for readers easier to follow our notations.
>
> > W2 Some symbols are not well-defined, such as the \cdot in the predictor f, little o, big O, \Omega, \Theta.
>
> We have complemented our full introduction of notations in the Appendix A and provide pointers in the main text. Below is the specific definition for the notations mentioned in the review:
> - For $\cdot$ in the predictor $f$, we refer to the product with a vector or a scalar. The use of $\cdot$ in $f$ (Eq. (4)) is because of the relaxation of IRMv1 that relaxes $w$ to be a scalar.
> - The other notations are used to compare two sequences $\{ a_n \}$ and $\{b_n \}$. Specifically, we employ standard asymptotic notations such as $O(\cdot)$, $o(\cdot)$, $\Omega(\cdot)$, and $\Theta(\cdot)$ to describe their limiting behavior. Specifically, we write $a_n =O(b_n) $ if there exists a positive real number $C_1$ and a positive integer $N$ such that $|a_n| \le C_1 |b_n| $ for all $n \ge N$. Similarly, we write $a_n = \Omega (b_n)$ if there exists $C_2 > 0$ and $N >0$ such that $|a_n| > C_2 |b_n |$ for all $n \ge N$. We say $a_n = \Theta(b_n)$ if $a_n = O(b_n)$ and $a_n = \Omega(b_n)$. Besides, if $\lim_{n \rightarrow \infty} |a_n/b_n| =0 $, we express this as $a_n = o(b_n)$. We use $\widetilde{O}(\cdot)$, $\widetilde{\Omega}(\cdot)$, and $\widetilde{\Theta}(\cdot)$ to hide logarithmic factors in these notations respectively. Moreover, we denote $a_n = \textrm{poly} (b_n)$ if $a_n = O((b_n)^p)$ for some positive constant $p$ and $a_n = \textrm{polylog}(b_n)$ if $a_n = \textrm{poly}( \log(b_n))$.
>
> > W3 The activation function and the variance share the same symbol.
>
> We have revised our draft and changed the activation function symbol to $\psi$.
>
> > W4 Quantitative results in Thm 4.1 (informal).
>
> The quantitative results of how much the increment of the spurious feature is larger than that of the invariant feature can be found at Eq. (15) in Appendix C.2. This difference is primarily determined by the empirical distributions of $\textup{Rad}(\alpha)$ and $\textup{Rad}(\beta_e)$, reflected in the quantities $\overline{C}_{j\ell} = \sum_e {\frac{1}{n_e} |\lbrace i\mid \textup{Rad}(\alpha)_i = j, \textup{Rad}(\beta_e)_i = \ell, i\in \mathcal{E}_e\rbrace| }$  for $j\in\lbrace \pm 1\rbrace,\ell\in \lbrace\pm 1\rbrace$. This (positive) difference decreases monotonically from Eq. (17) towards 0. We have added more quantitative results in Thm 4.1.
>
> > W5 Computatial complexity.
>
> We have revised the draft to include a discussion of computational complexity in Sec. 5.2:
>
> Compared to ERM, the additional computational and memory overhead introduced in FAT mainly lie in the FAT training and partitioning. At each training step, FAT needs $(k-1)$ additional forward and backward propagation, the same as Bonsai, while FAT only needs $\min(1,k-1)$ additional propagation. Besides, Bonsai additionally require another round of training with $(K-1)$ additional propagation, given $K$ total rounds.
>
> We calculated the computational overhead:
> |        | Camelyon17            |                | CivilComments      |                |
> |--------|-----------------------|----------------|--------------------|----------------|
> |        | Training time         | Memory (%)     | Training time      | Memory (%)     |
> | ERM    | 56.21$\pm$8.29 mins   | 22.56$\pm$0.00 | 24.22$\pm$0.33 hrs | 36.46$\pm$0.00 |
> | Bonsai | 214.55$\pm$1.13 mins  | 51.75$\pm$0.01 | 58.47$\pm$0.91 hrs | 64.43$\pm$0.31 |
> | FAT    | 101.14$\pm$12.79 mins | 51.92$\pm$0.04 | 28.19$\pm$1.15 hrs | 56.21$\pm$0.48 |
>
> The results aligned with our discussion. Bonsai requires much more time for the additional synthetic round and much more memory when there are 3 or more rounds. In contrast, FAT achieves the best  OOD performance without introducing additional much too computational overhead.
>
>
> > Q1 Claim Lma 3.2 to supplementary part.
>
> Lma 3.2 introduces key concepts of feature learning in neural networks by decomposing weights into two parts. While we agree moving Lma 3.2 will improve clarity, it could make readers who are not familiar with feature learning literature hard to follow our results (e.g., Reviewer Jcjd).
>
> > Q2 The details of Fig.2 are important.
>
> We included the details in our updated draft: the invariant and spurious feature learning terms plotted in Fig. 2 are the mean of $\langle \mathbf{w}\_{j,r}, j\mathbf{v}\_1 \rangle$ and $\langle \mathbf{w}\_{j,r}, j\mathbf{v}\_2 \rangle$ for $j\in \lbrace\pm 1\rbrace, r\in[m]$, respectively.
>
> We’d appreciate it if you could take the above responses into consideration when making the final evaluation of our work. Please let us know if there are any outstanding questions.

---

> ### Author Response · Authors · 2023-08-17
> **A summary of rebuttal in response to Reviewer KJZ8**
>
> Dear Reviewer KJZ8,
>
> We want to thank you again for reviewing our paper. We have responded to each of the weaknesses/questions you raised in the review. In summary,
> - We have revised our paper to improve the readability of our symbol system and details of Fig.2 following your suggestion;
> - We have discussed the quantitative results of Thm 4.1 and compared the computational complexity of our method with other competitors.
>
> We would appreciate it if you could take a look at our responses and let us know if any of your remaining concerns are not addressed, and we would try our best to address them.

---

> ### Author Response · Authors · 2023-08-19
> **A gentle reminder for the closing rebuttal window**
>
> Dear Reviewer KJZ8,
>
> We would like to remind you that the rebuttal will be closed very soon. To allow us sufficient time to discuss with you your concerns about our work, we would appreciate it if you could take some time to read our rebuttal and give us some feedback. Thank you very much.

---

### Author Rebuttal · Authors · 2023-08-09

Dear reviewers,


We thank the reviewers for their many helpful comments and suggestions.

Most reviewers agree that our theoretical findings are interesting, important and useful (KJZ8, NB5i, qcQH). The insights we obtained deepen the understandings of feature learning under distribution shifts (qcQH) and serve as a solid motivation for learning rich features for better OOD generalization (NB5i, qcQH). All reviewers agree that the proposed solution FAT is novel and the empirical improvements are strong and promising.

Regarding the reviewers’ concerns, we believe they can be addressed and we have revised our draft according to the reviewers’ valuable suggestions. In the following, we give our responses to the reviewers’ concerns and suggestions.

1. Regarding the linear setting (Jcjd, qcQH), we’d like to clarify that, because this is the **first** work to theoretically analyze the feature learning of ERM and OOD objectives and their interactions in OOD generalization:
  - We chose a **minimal setting** where we can observe all the necessary phenomena and which enables us to study complicated OOD objectives (e.g., IRMv1) that involve high-order derivatives. **The results with IRM regularization we obtained are not obvious under non-convexity, despite the linear activation**.
  - Nevertheless, **we also show that our key theoretical results could be generalized to the non-linear setting**. Please find the details in [our response to Reviewer qcQH](https://openreview.net/forum?id=eozEoAtjG8&noteId=KmRvdQIZ2C) due to the character limits.
  - Going beyond the linearity of activations, the experiments in Fig. 1b show that, our theoretical results align with the empirical discoveries in more complex settings.

2. Regarding more analysis of feature learning (NB5i, qcQH), we plotted the saliency maps to better understand the feature learning of different algorithms. The figures are attached to the rebuttal pdf and to our latest draft as well.
  - We first visualize the feature learning of ERM and FAT on ColoredMNIST-025. It can be found that ERM can learn both invariant and spurious features to predict the label, aligned with our theory.
  - However, ERM focuses more on spurious features and even forgets certain features with longer training epochs, which could be due to multiple reasons such as the simplicity biases of ERM. Hence predictions based on ERM learned features fail to generalize to OOD examples. In contrast, FAT effectively captures the meaningful features for all samples and generalizes to OOD examples well.
  - We also visualize the saliency maps of ERM, Bonsai and FAT on all real-world datasets used in our work. It can be found that, across various tasks and data modalities, FAT effectively learns more meaningful and diverse features than ERM and Bonsai, which serve as strong evidence for the consistent superiority of FAT in OOD generalization.


3. Regarding the computational complexity (KJZ8) and memory cost (Jcjd, qcQH), compared to ERM, the additional computational and memory overhead introduced in FAT mainly lie in the FAT training and partitioning.

- At each training step, FAT needs $(k-1)$ additional forward and backward propagation, the same as Bonsai, while FAT only needs $\min(1,k-1)$ additional propagation. Besides, Bonsai additionally requires another round of training with $(K-1)$ additional propagation, given $K$ total rounds.
- Although FAT needs to store all $D^a_i$, $D^r_i$ and $w_i$ yielded in previous rounds, the storage of these items does not cause much a memory issue, while **training using the FAT objective (Eq.7) with all previous subsets can lead to OOM for a large network**, as Eq. 7 increases the batch size by a factor of $k$ for the $k$-th round. Note that typically, the batches are also sampled from each environment as shown in Table 9. Let the number of sampled domains be $d$, and the batch size from each domain be $b$, then each round will additionally introduce $2d\times b$ samples in each minibatch, which is typically more than 200 and results in OOM for a V100 GPU with around 32G GPU memory.

- We calculated the computational and memory overhead in the table below. The results aligned with our discussion. Bonsai requires much more time for the additional synthetic round and much more memory when there are 3 or more rounds. In contrast, **FAT achieves the best  OOD performance without introducing too much additional computational overhead**.

|        | Camelyon17            |                | CivilComments      |                |
|--------|-----------------------|----------------|--------------------|----------------|
|        | Training time         | Memory (%)     | Training time      | Memory (%)     |
| ERM    | 56.21$\pm$8.29 mins   | 22.56$\pm$0.00 | 24.22$\pm$0.33 hrs | 36.46$\pm$0.00 |
| Bonsai | 214.55$\pm$1.13 mins  | 51.75$\pm$0.01 | 58.47$\pm$0.91 hrs | 64.43$\pm$0.31 |
| FAT    | 101.14$\pm$12.79 mins | 51.92$\pm$0.04 | 28.19$\pm$1.15 hrs | 56.21$\pm$0.48 |


Besides, we have revised our paper to correct the typos and inconsistencies, provide more intuitive explanations of several key concepts and designs, and add a table of notations for easy reference.

In addition, we have provided a link to our code for reproducing the results in our paper to the AC.

Please let us know if you have any further questions. Thanks.

---

### Decision · Program_Chairs · 2023-09-21

**Decision:**

Accept (poster)

**Comment:**

The work advances the understandings of feature learning under distribution shifts and it provides a solid foundation for learning rich features for better OOD generalization with strong empirical studies. Most reviewers are positive about the novelty of the work both theoretically and empirically.

In the final version, we suggest authors incorporate all the suggestions raised by the reviewers, especially to improve the readability of the paper raised by several reviewers.